# Region-specific and state-dependent action of striatal GABAergic interneurons

Elodie Fino[1,2,3,4], Marie Vandecasteele[1,2], Sylvie Perez[1,2], Frédéric Saudou[3,4,5] & Laurent Venance [1,2]

Striatum processes a wide range of functions including goal-directed behavior and habit formation, respectively encoded by the dorsomedial striatum (DMS) and dorsolateral striatum (DLS). GABAergic feedforward inhibition is known to control the integration of cortical information by striatal projection neurons (SPNs). Here we questioned whether this control is specific between distinct striatal functional territories. Using opto-activation and opto-inhibition of identified GABAergic interneurons, we found that different circuits are engaged in DLS and DMS, both ex vivo and in vivo: while parvalbumin interneurons efficiently control SPNs in DLS, somatostatin interneurons control SPNs in DMS. Moreover, both parvalbumin and somatostatin interneurons use a dual hyperpolarizing/depolarizing effect to control cortical input integration depending on SPN activity state: GABAergic interneurons potently inhibit spiking SPNs while in resting SPNs, they favor cortical activity summation via a depolarizing effect. Our findings establish that striatal GABAergic interneurons exert efficient territory-specific and state-dependent control of SPN activity and functional output.

[1] Center for Interdisciplinary Research in Biology (CIRB), College de France, CNRS UMR7241, INSERM U1050, Paris 75005, France. [2] Université Pierre et Marie Curie, ED 158, Paris Sciences et Lettres, Paris 75005, France. [3] INSERM U1216, Grenoble 38000, France. [4] Grenoble Institute of Neuroscience, Université Grenoble Alpes, Grenoble 38000, France. [5] CHU Grenoble Alpes, Grenoble 38000, France. Correspondence and requests for materials should be addressed to E.F. (email: elodie.fino@univ-grenoble-alpes.fr)

Cerebral cortex and basal ganglia are tightly interconnected structures involved in goal-directed behavior and procedural learning[1–3]. Striatum, the main input nucleus of basal ganglia, receives massive convergent glutamatergic inputs from the whole cortex and distinct inputs from the different cortical areas form distinct functional territories within the striatum[4–7]. Two major functional territories are the dorsomedial striatum (DMS), responsible for cognitive function and goal-directed behavior, and the dorsolateral striatum (DLS), which corresponds to the sensorimotor territory and is involved in habit formation[3,8]. The two territories also interact with each other since in the same behavioral task involving procedural learning, DMS and DLS neurons are both activated, but preferentially at different phases of the task and at different stages of the learning course[9,10]. Both territories then relay the information toward the output structures of basal ganglia (internal part of the globus pallidus and the substantia nigra *pars reticulata* (SNr)).

DMS and DLS are functionally distinct, although the composition and the properties of their microcircuits appear similar. Since striatum has no evident anatomical boundaries, functional differences of the distinct striatal regions could arise from their distinct incoming cortical inputs. The composition of the striatal circuits could also define specific functional regions. Striatal neuronal circuits are composed of a majority of striatal projection neurons (SPNs), and a variety of GABAergic interneurons, which are also efficiently recruited by cortical afferents[11–14] and exert a strong feedforward inhibition on SPNs[15–17]. The role of striatal interneurons is highlighted by the consequences of global alteration in GABAergic circuits, which alters synaptic plasticity[18,19] and leads to severe motor deficits that are particularly exemplified in the context of dystonia or Tourette Syndrome[20]. The two most extensively described interneuron subtypes in striatum are the parvalbumin (PV)-expressing cells (fast-spiking interneurons) and the somatostatin/neuropeptide Y/ nitric oxide synthase (SOM/NPY/NOS)-expressing cells (persistent and low-threshold spiking cells).

Here we questioned whether PV and SOM interneurons could play a role in the distinct properties of DMS and DLS. Using in vivo multi-channel recordings associated with optogenetics, we found that opto-inhibition of PV or SOM cells in DMS or DLS differentially control SNr activity. We explored this functional dichotomy within the striatum and found that PV cells control the activity of SPNs in DLS while SOM cells control SPNs in DMS. This dichotomy is based on a marked heterogeneity in the anatomical distribution, connectivity and electrophysiological properties of PV and SOM cells in DLS and DMS. Interestingly, our results show that the territory specificity of GABAergic microcircuits translates to the trans-striatal transfer of information of cortical inputs to the nigral output of the striatum. We also described that both PV and SOM interneurons mediate a dual hyperpolarizing/depolarizing control of SPNs that depends on SPN activity state, with the depolarizing effect favoring cortical integration. Our findings therefore demonstrate that the selective feedforward control of cortical inputs by GABAergic interneurons is specific to the striatal functional territories and to the SPN activity state.

## Results

### SOM and PV cells in DMS and DLS differentially affect SNr spontaneous activity.
SPNs act as coincidence detectors of coherent cortical activity, extract pertinent information from background noise and relay signals towards the main basal ganglia output structure, the SNr. We used SNr spontaneous activity as a readout of striatal output modulation by striatal interneurons. We first examined the effect of an opto-inhibition of SOM and PV interneurons in DMS or DLS onto SNr spontaneous activity (Fig. 1a). To do so, we recorded extracellular activity of SNr units in vivo in urethane-anesthetized *SOM::Arch3* and *PV::Arch3* mice that selectively express Arch3 in SOM and PV cells, respectively (Fig. 1a, b and Supplementary Fig. 1). SNr units were identified by their high spontaneous spiking frequency (median (interquartile range (IQR)): 18.7 (10.3) Hz, $n = 239$ units) and their regularity (coefficient of variation of the inter-spike intervals (CV-ISI), median (IQR): 0.41 (0.22)) (Fig. 1c). In control conditions (i.e., without opto-inhibition), no difference was found between the distribution of the spontaneous firing rates of SNr units recorded in *SOM::Arch3* vs. *PV::Arch3* mice ($n = 130$ units from 17 *SOM::Arch3* mice, $n = 109$ units from 13 *PV::Arch3* mice, $p = 0.2044$, Fig. 1c). Using optic fibers implanted in the DMS or DLS, we tested the effect of opto-inhibition of SOM and PV cells in the two striatal territories (Fig. 1a, b). We found that all conditions (SOM and PV in DMS and DLS) could efficiently increase SNr activity (Fig. 1d–f and Supplementary Fig. 2). However, we found a selective contribution of SOM and PV opto-inhibition to SNr activity depending on the targeted striatal territory (Fig. 1d, e). First, the proportion of modulated SNr units was significantly higher when opto-inhibiting PV interneurons than SOM interneurons in both DLS and DMS (50.0% of significantly modulated units in PV-DLS, $n = 68$ vs. 8.7% in SOM-DLS, $n = 69$, $p = 0.0001$; 32.9%, $n = 79$ in PV-DMS vs. 14.7%, $n = 109$ in SOM-DMS, $p = 0.0079$), suggesting a stronger weight of PV interneurons compared to SOM interneurons. Second, we found a stronger impact of PV opto-inhibition in DLS than in DMS ($p = 0.0422$) (Fig. 1d–f), suggesting a territory specificity in the modulation of SNr activity. These effects were confirmed at the population level (Fig. 1f and Supplementary Fig. 2), where silencing PV or SOM striatal interneurons induced an overall increase in SNr activity in all conditions (SOM-DMS: median (IQR) change +1.7 (6.1)%, $p = 0.0001$; PV-DMS: +4.1 (8.3)%, $p < 0.0001$: SOM-DLS: +1.0 (5.5)%, $p = 0.0196$; PV-DLS: +9.1 (15.4)%, $p < 0.0001$). The comparison of effects of opto-inhibition between striatal interneurons and territories on the SNr firing rate confirmed the stronger effect of silencing PV compared to SOM interneurons in both territories (DMS: $p = 0.0208$; DLS: $p < 0.0001$), and the stronger effect of PV interneurons in the DLS compared to the DMS ($p = 0.0087$) (Fig. 1f). These differential effects did not result from an anatomical bias in the DMS or DLS connectivity of the targeted SNr region since we did not observe a segregation of significantly modulated units or an effect depending on the recording location (Supplementary Fig. 2).

Together, these in vivo data suggest that SOM and PV interneurons of the dorsal striatum exert differential effects on the spontaneous activity of the SNr. In addition, these effects depend on their location in striatal territories (DMS vs. DLS).

### Selective inhibitory weight on spiking SPNs in DMS and DLS.
To investigate the cause of this differential effect, we dissected the effects of SOM and PV interneuron opto-inhibition onto the SPNs in DMS and DLS. To this aim, we characterized the effect of striatal GABAergic circuits locally in the striatum by performing ex vivo experiments using brain slices preserving layer 5 cortical connections from cingulate cortex to DMS or from somatosensory cortex to DLS (see Methods). We first confirmed that cortical layer 5 pyramidal cells directly contact both types of striatal interneurons as previously described[11–14,21] (Fig. 2). We found that PV cells are tightly locked to the timing of cortical stimulations and display reproducible evoked responses, while SOM cells show more variability in their responses to cortical inputs in

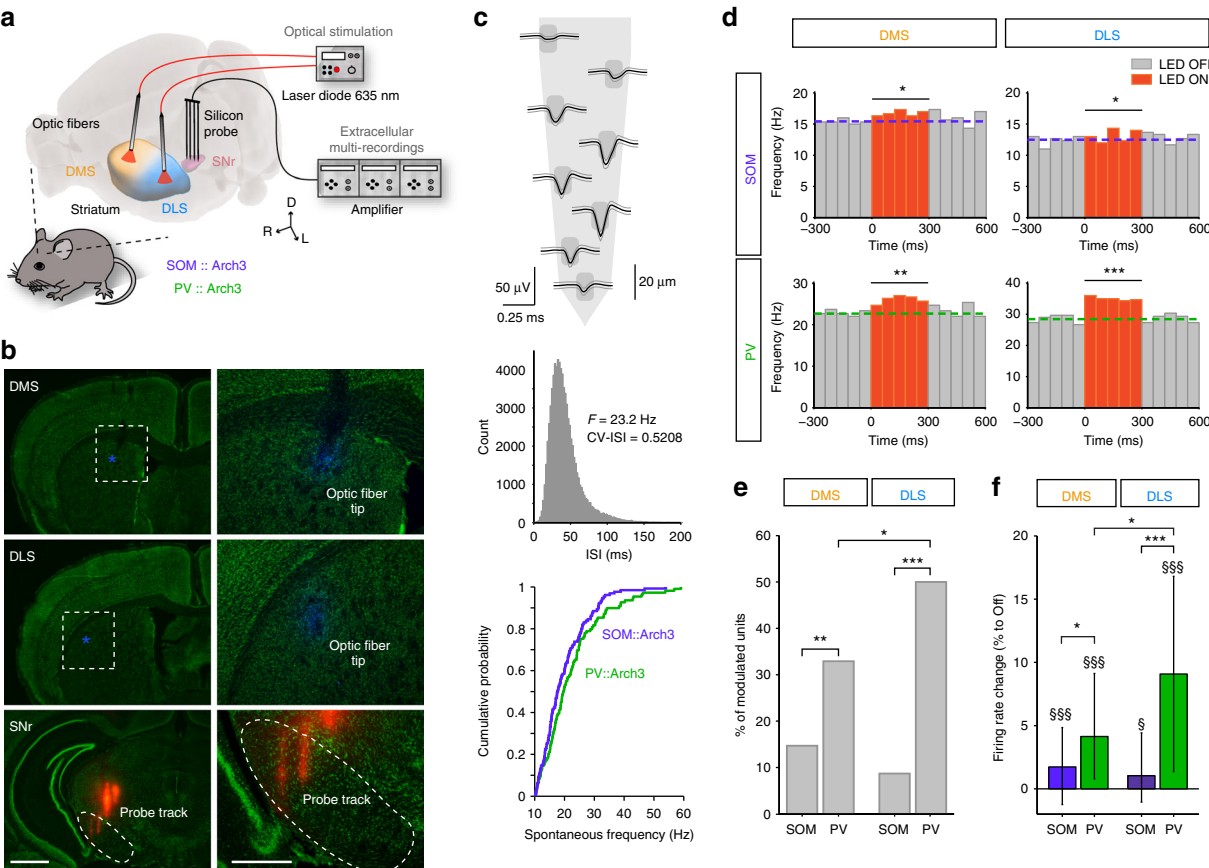

**Fig. 1** Differential modulation of basal ganglia output activity by SOM and PV cells of DMS and DLS. **a** Experimental set up: in vivo multi-channel extracellular recordings of SNr unit activity using 4-shank 32-site silicon probe while PV or SOM interneurons are opto-inhibited using two optic fibers implanted in the DMS and DLS. **b** Post-hoc histological confirmation of the location of the recording probe (shanks painted with DiI before the implantation, red) in the SNr (delineated) and the two optic fibers (tips painted with DAPI, blue) in DMS and DLS, observed in Nissl-stained (green) coronal slices. Scale bars: 1 mm (left), 500 μm (right). **c** Top: Mean ± SD (black and gray lines) waveform of a clustered SNr unit represented atop the corresponding recording sites of the probe (sketch of 1 shank, the 8 recording sites in dark gray). Middle: Distribution of the interspike interval (ISI) of the same unit, displaying the high frequency and low CV-ISI typical of SNr units. Bottom: the distribution of SNr spontaneous firing rates is similar in *SOM::Arch3* mice ($n = 130$ units from 17 mice) and *PV::Arch3* mice ($n = 109$ units from 13 mice) (Kolmogorov–Smirnov test, $p = 0.2044$). **d** Spontaneous firing activity of representative SNr units significantly activated by opto-inhibition (300 ms, 10 mW, 50 trials) of SOM (top) or PV (bottom) light in DMS (left) or DLS (right). The colored dashed lines represent the median of the 300 ms-OFF period before the stimulation. **e** Proportion of SNr units displaying a significant modulation of their firing in response to opto-inhibition of PV or SOM interneurons in DMS or DLS (total $n = 109$ units for *SOM::Arch3* in DMS, $n = 69$ in DLS, $n = 79$ units for *PV::Arch3* mice in DMS, $n = 68$ in DLS). **f** Median (IQR) change in firing rate of all recorded SNr units in response to opto-inhibition of PV (green) and SOM (purple) interneurons in the DMS and DLS. §§§Indicates a significant effect across the population of recorded SNr units for PV-DLS ($^{$}p < 0.05$, $^{$$$}p < 0.001$, Wilcoxon signed rank test corrected for multiple comparisons). In **e** and **f**, results in the 4 conditions were compared using a generalized linear (**e**) or a linear (**f**) model (see Methods) followed by post-hoc tests, $^*p < 0.05$, $^{**}p < 0.01$, $^{***}p < 0.001$. For the 3D images in **a**, brain and brain structures are captured from the Allen Institute for Brain Science's Mouse Brain Atlas[74] (© 2004 Allen Institute for Brain Science, Allen Mouse Brain Atlas available from: mouse.brain-map.org/) using Brain Explorer®2[75]

both territories (Fig. 2a–c). We built input/output curves for SOM and PV interneurons and for SPN evoked responses by gradually increasing the stimulation of cortical afferents (Fig. 2d–f). In the two territories, 100% of PV and SOM interneurons were efficiently activated by cortical inputs and, in both DMS and DLS, SOM and PV cells are recruited before SPNs since they discharged for lower cortical activation than SPNs ($F_{2, 20} = 20.19$, $p < 0.0001$ for DMS and $F_{2, 18} = 11.17$, $p = 0.0025$ for DLS) (Fig. 2d–f). These properties place both SOM and PV cells in a strategic position to mediate efficient feedforward inhibition in both DMS and DLS.

Based on our in vivo observations, we next investigated whether PV and SOM cells would mediate selective feedforward inhibition in DMS and DLS. We therefore quantified the weight of the feedforward inhibition mediated by SOM and PV cells on

the integration of cortical inputs by SPNs in DMS and DLS. To do so, we opto-inhibited SOM or PV cells during cortical activity using either transgenic *SOM::Arch3* and *PV::Arch3* (in which most PV or SOM interneurons in the brain express Arch3; Fig. 3a and Supplementary Figs. 1 and 3) or virally with AAV-Flex-Arch-tdtomato injected in DLS or DMS in *SOM-Cre* or *PV-Cre* mice. The viral approach allowed us to restrict the expression of Arch3 to only striatal interneurons either in DMS or DLS (Fig. 3a and Supplementary Fig. 4) to exclude any eventual external sources of PV or SOM inputs (from cortex or globus pallidus for example) and to study only the effect of local microcircuits. We analyzed SPN spiking activity evoked by a single cortical stimulation during opto-inhibition of interneurons and compared the effect of SOM and PV cells in DMS and DLS. We found that SOM cell opto-inhibition significantly increased the spiking

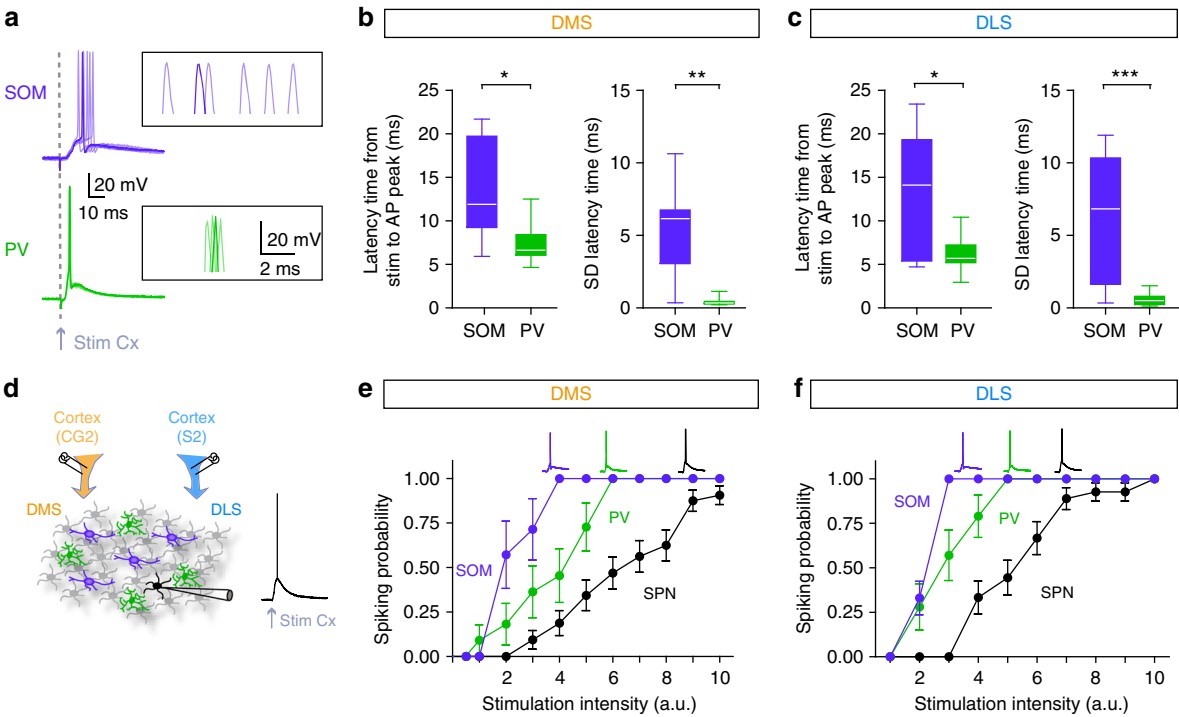

**Fig. 2** SOM and PV cells are differentially recruited by cortical inputs in DMS and DLS. **a** Six superimposed APs evoked in either SOM or PV cells for a given cortical stimulation intensity. **b**, **c** Whisker boxes representing the latency, and the standard deviation (SD) of the latency, between the stimulation artifact and the peak of the AP in PV cells (green) and SOM cells (purple), in DMS and DLS. In both territories, PV interneurons display much shorter latency and SD than SOM cells, (DMS, $p = 0.0148$ for latency and $p = 0.0019$ for SD; DLS, $p = 0.0334$ for latency and $p = 0.0004$ for SD). PV cells display a short latency (6.38 ± 0.38 ms, $n = 30$) and a remarkable small SD of the latency (0.53 ± 0.07 ms, $n = 30$), and this with no significant difference between DMS and DLS ($p = 0.1127$ for latency and $p = 0.4641$ for SD latency), accounting for a very time-locked PV cells recruitment. In contrast, APs evoked in SOM cells by cortical stimulation exhibited longer latency (13.31 ± 1.77 ms, $n = 17$) and a higher latency SD (5.75 ± 0.93 ms, $n = 17$), with no difference in DMS and DLS ($p = 0.8884$ for latency and $p = 0.7430$ for SD latency, Mann–Whitney unpaired $t$-test). **d** Schematic representation of the experimental set up: electrical stimulations were applied in the cortex (CG2 for DMS slices and S2 for DLS slices) and evoked APs were recorded in SPNs, PV and SOM cells. **e**, **f** Input/output relationship (mean ± SEM) is plotted for different neuronal subtypes, SPN (black), PV cells (green) and SOM cells (purple) in DMS (**e**) and DLS (**f**). Insets represent APs evoked in the different cell types after cortical stimulation. In both DLS and DMS, PV and SOM cells discharge for much lower cortical activation than SPNs ($F_{2, 20} = 20.19$, $p < 0.0001$ for DMS, SPN $n = 33$, PV $n = 11$, SOM $n = 8$ and $F_{2, 18} = 11.17$, $p = 0.0025$ for DLS, SPN $n = 27$, PV $n = 14$, SOM $n = 8$, two-way ANOVA); *$p < 0.05$, **$p < 0.01$, ***$p < 0.001$

probability of SPN-DMS ($p = 0.005$, $n = 10$), whereas the spiking probability of SPN-DLS remained unaffected ($p = 0.1571$, $n = 9$). The picture was reversed for PV cells since their opto-inhibition led to an increase in spiking probability of SPNs in DLS ($p = 0.0087$, $n = 14$) but not in DMS ($p = 0.4040$, $n = 8$) (Fig. 3b, c). These results were confirmed with the opposite strategy, by activating SOM and PV interneurons while stimulating cortical afferents using *SOM::ChR2* or *PV::ChR2* mice (Fig. 3d and Supplementary Figs 5 and 6) or virally expressed with AAV-DIO-ChR2-mCherry injected in *SOM-Cre* or *PV-Cre* mice (Fig. 3d and Supplementary Figs 4, 5 and 6). We found that SOM opto-activation decreased spiking probability in DMS ($p = 0.0002$, $n = 7$) but not in DLS ($p = 0.9114$, $n = 6$), while PV cell opto-activation decreased the spiking probability of SPNs in DLS ($p = 0.0059$, $n = 9$) but not in DMS ($p = 0.9816$, $n = 9$) (Fig. 3e, f). These results show that PV and SOM interneurons efficiently modulate cortically evoked SPN firing in selective territories, with a stronger impact of SOM cells in DMS and of PV cells in DLS. Interestingly, we obtained similar results with transgenic mice or virally expressed opsins, which shows that only local PV and SOM striatal interneurons have a role in the effects we observed.

To further explore this differential effect, we investigated the local inhibitory control of the two GABAergic interneuron subtypes on SPN firing in DMS and DLS (Fig. 4a). Using *SOM::ChR2* or *PV::ChR2* mice (or virally expressed with AAV-DIO-

ChR2-mCherry in *SOM-Cre* and *PV-Cre* mice, Supplementary Fig. 4), we compared the inhibitory weight of SOM and PV cells on SPN firing (induced by suprathreshold depolarizing current steps) in DMS and DLS (Fig. 4b, c; Supplementary Fig. 6). We mimicked brief (a single spike induced by 5 ms light pulse, Fig. 4b) or long (bursts of spikes induced by 300 ms light pulse, Fig. 4b, c) activation of GABAergic circuits. We found that SPN firing frequency was significantly decreased by brief opto-stimulation of both SOM and PV cells in DMS and DLS ($F_{3, 54} = 9.151$, $p < 0.0001$) but with different magnitudes depending on the striatal territories. In DMS, SOM cell activation exerted stronger inhibition than PV cell activation (SOM opto-stimulation, normalized frequency before vs. during opto-stimulation was 0.75 ± 0.04, $n = 16$, PV opto-stimulation, normalized frequency was 0.85 ± 0.03, $n = 16$, $p = 0.0258$). In DLS, PV cells induced a stronger decrease in frequency in SPNs than SOM cells (PV: 0.61 ± 0.04, $n = 12$, SOM: 0.78 ± 0.03, $n = 14$, $p = 0.0014$) (Fig. 4b). Using the viral infection strategy, we observed similar results, indicating that only local GABAergic circuits are responsible for the inhibitory weight specificity (Supplementary Fig. 7). These findings were confirmed using longer duration (300 ms) opto-stimulation of both PV and SOM cells in DMS and DLS ($F_{3, 54} = 9.108$, $p < 0.0001$). SOM-DMS cells exerted a stronger inhibition onto SPN firing frequency than PV-DMS cells (frequency ratio before vs. during opto-stimulation,

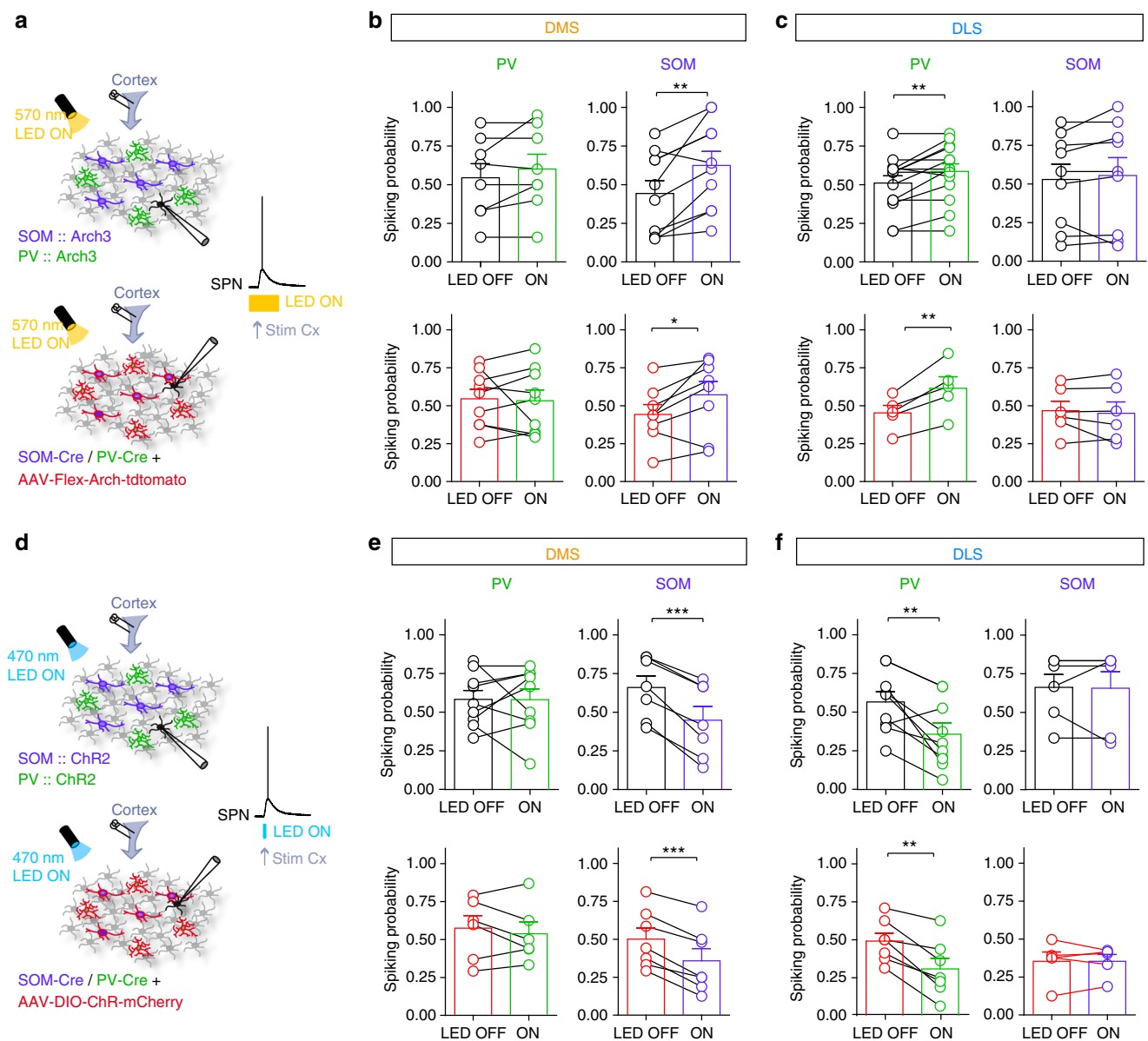

**Fig. 3** Selective modulation of cortically-evoked spiking activity by SOM and PV cells. **a** Experimental set up: electrical stimulations were applied in the cortex and evoked APs were recorded in SPNs, in control or with specific opto-inhibition of PV cells or SOM cells in transgenic mice *PV::Arch3* and *SOM:: Arch3* (top) or *PV-Cre* and *SOM-Cre* mice virally infected with AAV-Flex-Arch-tdtomato (bottom) (Arch3 activation for 300 ms with a 570 nm LED). **b**, **c** The spiking probability of SPNs was compared between interleaved control and opto-inhibition conditions in DMS (**b**) and DLS (**c**) using transgenic mice (top, black) or virally infected interneurons (bottom). Individual experiments and averaged (mean ± SEM) normalized spiking probability are represented. Opto-inhibition of SOM cells leads to a significant increase in spiking probability in DMS ($n = 15$, $p = 0.0063$ for transgenic and $n = 8$, $p = 0.033$ for virus experiments) and no difference in DLS ($n = 10$, $p = 0.2567$ for transgenic and $n = 6$, $p = 0.55479$ for viruses). For PV cells, the effect is opposite, opto-inhibition of PV cells significantly increase the spiking probability in DLS ($n = 16$, $p = 0.0288$ for transgenic and $n = 5$, $p = 0.0063$ for viruses), while their opto-inhibition has no effect in DMS ($n = 7$, $p = 0.1854$ for transgenic and $n = 9$, $p = 0.8007$ for viruses). **d** Experimental set up: electrical stimulations were applied in the cortex and evoked APs were recorded in SPNs, in control or with specific opto-activation of PV cells or SOM cells with ChR2 activation in transgenic mice *PV::ChR2* and *SOM::ChR2* (top) or *PV-Cre* and *SOM-Cre* mice virally infected with AAV-Flex-ChR-mCherry (bottom) (for 5 ms using a 470 nm excitation LED). **e**, **f** The spiking probability of SPNs was compared between the control and opto-activation conditions in DMS (**e**) and DLS (**f**) using transgenic mice (top, black) or virally infected interneurons (bottom, red). Individual experiments and averaged normalized spiking probability are represented. SOM opto-activation leads to a significant decrease in spiking probability in DMS ($n = 7$, $p = 0.0002$ for transgenic mice and $n = 7$, $p = 0.0002$ for viral experiments) and no difference in DLS ($n = 6$, $p = 0.9114$ for transgenic and $n = 5$, $p = 0.9980$ for viruses). Opto-activation of PV cells significantly decrease the spiking probability in DLS ($n = 9$, $p = 0.0059$ for transgenic and $n = 7$, $p = 0.0023$ for viruses) but not in DMS ($n = 9$, $p = 0.9816$ for transgenic and $n = 6$, $p = 0.4518$ for viruses). Paired tests, *$p < 0.05$, **$p < 0.01$, ***$p < 0.001$

SOM, $0.65 \pm 0.03$, $n = 14$ and PV, $0.75 \pm 0.02$, $n = 16$; $p = 0.0129$). In contrast, SOM-DLS cells had a weaker weight than PV-DLS cells (PV, $0.54 \pm 0.05$, $n = 12$, SOM, $0.66 \pm 0.02$, $n = 14$; $p = 0.0011$) (Fig. 4c).

These data demonstrate that SOM cells have a stronger inhibitory weight in DMS while PV cells control more efficiently SPN firing rate in DLS. Remarkably, in both DMS and DLS, the inhibitory effect of PV cells was independent on the initial SPN

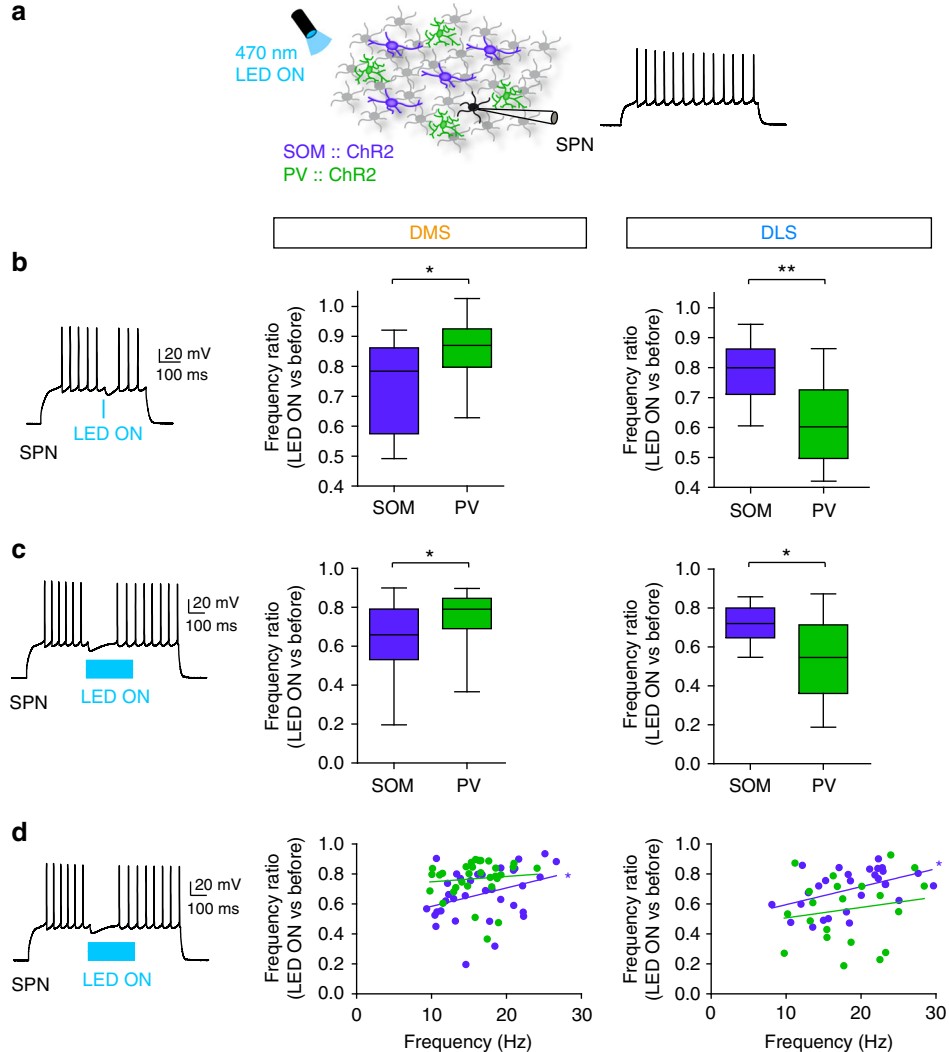

**Fig. 4** Differential local inhibitory weight of SOM and PV cells on firing SPNs. **a** In the DMS and DLS, SPNs are recorded while either PV cells or SOM cells are opto-activated with ChR2 with a 470 nm excitation LED. Effect of PV or SOM cells on SPNs was evaluated in active SPNs, i.e., while inducing spiking activity in SPNs by depolarizing current steps (500 ms in **b** or 1 s in **c** and **d**). **b** Left: Representative trace of discharge frequency inhibition by a brief (5 ms) opto-activation of PV-DLS cells. Right: Frequency inhibition ratio (frequency during LED ON vs. before LED ON, mean ± SEM) in SPNs induced by 5 ms light pulse. Frequency decrease was significantly higher for SOM cell (purple) activation in DMS territory ($p = 0.0258$, $n = 16$ SOM cells and $n = 16$ PV cells), while this decrease was significantly higher for PV cells (green) activation in DLS territory ($p = 0.0014$, $n = 14$ SOM cells and $n = 12$ PV cells). **c** Left: Representative trace of discharge frequency inhibition by long opto-activation of PV-DLS cells. Right: Frequency inhibition ratio (frequency during LED ON vs. before LED ON) induced by 300 ms light pulses. The ratio is significantly higher for SOM-DMS than PV-DMS cells ($p = 0.0129$, $n = 14$ SOM cells and $n = 16$ PV cells) and in PV-DLS cells has a stronger weight than SOM-DLS cells ($p = 0.0011$, $n = 14$ SOM cells and $n = 12$ PV cells). **d** Correlation of PV (green) and SOM (purple) cell inhibitory weight (discharge frequency ratio) and initial SPN discharge frequency. There is a significant correlation for SOM cells in both DMS ($r^2 = 0.3481$, Pearson's correlation, $p = 0.0437$, $n = 14$) and DLS ($r^2 = 0.4648$, $p = 0.0127$, $n = 14$), and there was no significant correlation for PV cells ($r^2 = 0.097$ for DMS and $r^2 = 0.1739$ for DLS); $*p < 0.05$, $**p < 0.01$

firing rate ($p = 0.5973$ for DMS and $p = 0.4509$ for DLS), whereas the effect of SOM cells was negatively correlated to the SPN initial firing rate ($p = 0.0437$ for DMS and $p = 0.0127$ for DLS) (Fig. 4d). These results show that PV cells have a constant inhibitory weight regardless of the level of SPN spiking activity, whereas feedforward inhibition exerted by SOM cells decreases with increasing SPN spiking activity.

**GABAergic microcircuits have distinct properties in DMS and DLS.** We next investigated whether the difference in local inhibitory weight of SOM and PV interneurons in DMS and DLS reflected anatomical and/or functional specificities of these microcircuits in both territories. Because enrichment of one population in a specific region could account for a stronger

inhibitory weight on SPNs, we first examined the anatomical distribution of SOM and PV interneurons in DMS and DLS (Fig. 5a, b). Immunostaining for somatostatin and parvalbumin revealed a heterogeneous distribution of interneurons ($p < 0.0001$, $F_{3, 24} = 22.79$). SOM cells were equally distributed in DMS and DLS ($179.9 \pm 16.2$ in DMS and $175.7 \pm 14.8$ in DLS, $p = 0.8510$, $n = 9$ mice) (Fig. 5a) and their density was similar to PV cells in DMS ($p = 0.7253$ for DMS and $p = 0.5715$ for DLS). In contrast, PV cells were particularly enriched in the DLS when compared to the DMS ($+160\%$, $296.0 \pm 23.2$ cells per $0.1$ mm$^3$ in DLS vs. $187.9 \pm 14.2$ in DMS, $p = 0.0018$, $n = 7$ mice) (Fig. 5b). This heterogeneous distribution was observed in 1–2-month-old animals (Fig. 5a, b) and was maintained in adult animals as well (Supplementary Fig. 8).

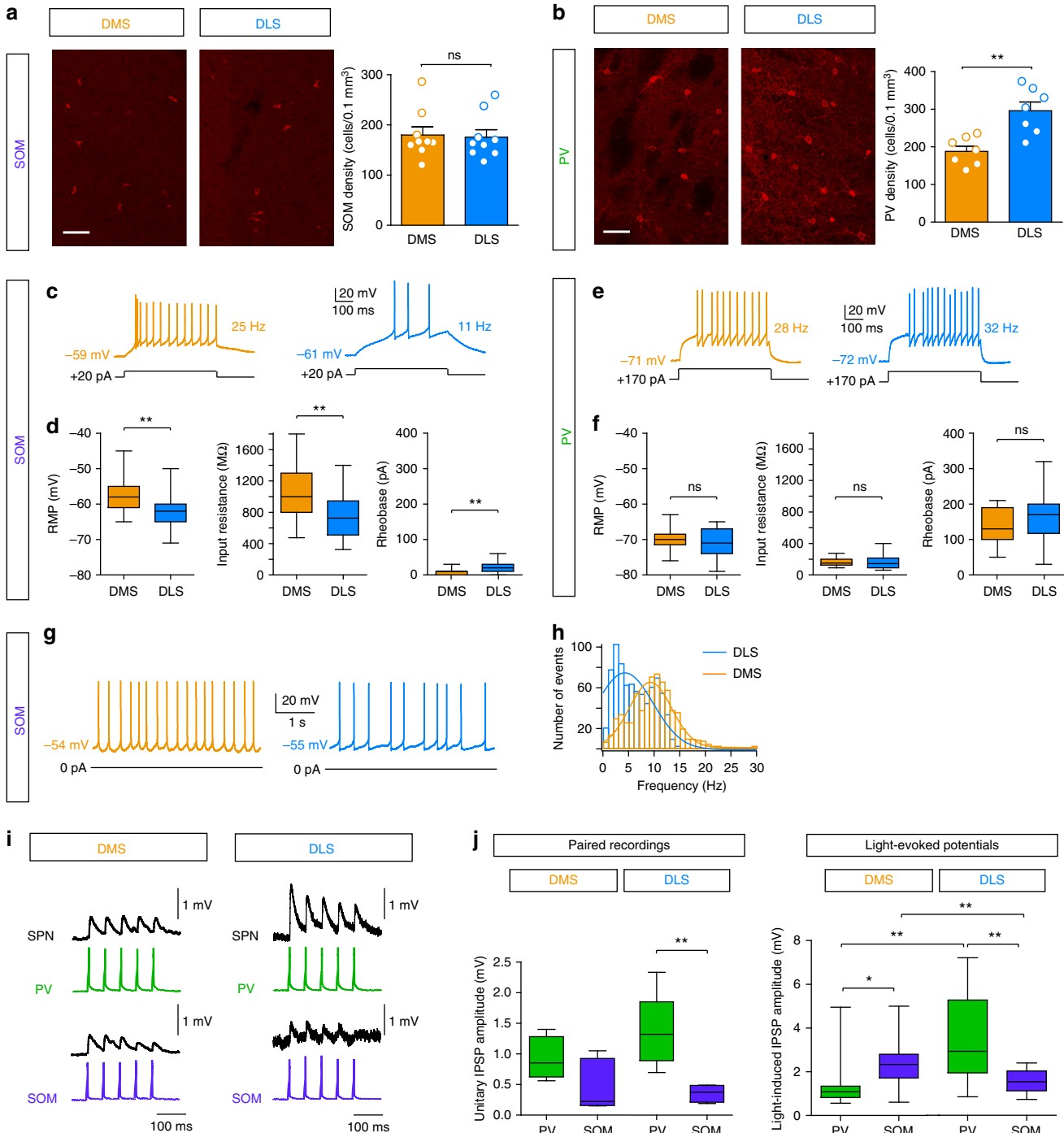

**Fig. 5** Heterogeneous distribution, electrical properties and connectivity of SOM and PV cells in DMS and DLS. **a**, **b** Confocal microscopy images showing a representative overview of SOM (**a**) and PV (**b**) interneuron expression in DMS and DLS (scale bars: 50 μm), identified following immunostainings. On the right panels, quantification of SOM (**a**) and PV (**b**) interneurons in DMS and DLS is represented by the cell density (number of cells per 0.1 mm$^3$) of SOM and PV interneurons in each territory ($n = 9$ mice for SOM interneurons and $n = 7$ mice for PV interneurons). **c** Representative responses of SOM cells to identical current steps ($+20$ pA) in DLS and DMS. **d** Membrane and electrical properties of SOM cells in DMS (orange, $n = 20$) and DLS (blue, $n = 27$) (mean ± SEM).
**e** Representative responses of PV cells to identical current steps ($+170$ pA) in DLS and DMS. **f** Membrane and electrical properties of PV cells in DMS (orange, $n = 17$) and DLS (blue, $n = 30$). There is no significant difference in rheobase ($p = 0.2761$), RMP ($p = 0.6030$) and input resistance ($p = 0.3260$) of PV cells in DMS or in DLS. **g** Representative spontaneous activity of SOM cells in DMS and DLS. **h** Distribution and Gaussian fits of the discharge frequency in DLS (100 action potentials per cell, $n = 8$) and DMS ($n = 6$): the frequency is significantly higher in DMS ($p < 0.0001$). **i** Representative connections between of PV-SPN and SOM-SPN connections in DMS and DLS in response to 20 Hz single AP trains evoked in PV and SOM cells with current injections. **j** Amplitude of unitary IPSPs in a SPN (single AP with paired patch-clamp recordings) in DMS and DLS. For PV-SPN connections the unitary IPSP was median (interquartile range (IQR)): 0.9 (0.6) mV ($n = 5$) in DMS and 1.3 (0.7) mV ($n = 6$) in DLS and for SOM-SPN connections 0.3 (0.8) mV ($n = 6$) in DMS and 0.4 (0.2) mV ($n = 4$) in DLS. They were not statistically different in DMS ($p = 0.0648$) but much stronger from PV in DLS ($p = 0.0094$). Concerning light-induced IPSP amplitudes, SOM cell opto-activation induced significantly stronger IPSPs in DMS ($p = 0.0158$) and PV cells in DLS ($p = 0.0019$) (SOM-DMS: 2.3 (1.1) mV, $n = 19$, PV-DMS: 1.1 (0.5) mV, $n = 16$, SOM-DLS: 1.5 (1.1) mV, $n = 14$, and PV-DLS: 2.9 (3.2) mV, $n = 15$). *$p < 0.05$, **$p < 0.01$, ns not significant

Because a heterogeneous anatomical distribution could not fully account for the functional dichotomy between DMS and DLS (the distribution of SOM interneurons is homogeneous, Fig. 5a), we next examined whether SOM and PV cells exhibited different electrophysiological properties in DMS and DLS. For this purpose, we performed whole-cell recordings of SOM and PV cells targeted thanks to the green fluorescent protein/yellow fluorescent protein (GFP/YFP) of the Arch/ChR2 constructs. All recorded fluorescent neurons exhibited typical basic properties and spiking activity of SOM or PV interneurons (Supplementary Table 1). We first cross-compared the electrophysiological properties of SOM and PV cells. We observed that, in both DMS and DLS, SOM interneurons display more depolarized resting membrane potential (RMP) ($p < 0.0001$ for both DMS and DLS), higher input resistance (Ri) ($p < 0.0001$ for both DMS and DLS) and lower rheobase ($p < 0.0001$ for both DMS and DLS) than PV cells, suggesting that SOM cells are more excitable than PV cells. We then compared the membrane properties of SOM and PV cells according to their location in DMS and DLS. We found no difference in RMP ($n = 30$ in DLS and $n = 18$ in DMS, $p = 0.6030$), Ri ($p = 0.3260$) and rheobase ($p = 0.2761$) between PV-DMS and PV-DLS (Fig. 5e, f). In contrast, we observed a marked difference in the properties of SOM cells since those located in DMS exhibit a higher excitability than those located in DLS: lower rheobase ($n = 27$ in DLS and $n = 20$ in DMS, $p = 0.0011$), depolarized RMP ($p = 0.0059$) and higher Ri ($p = 0.0049$) (Fig. 5c, d). Although PV cells were not spontaneously active (in ex vivo conditions), spontaneous activity was recorded in half of SOM cells in both territories (14/22 cells in DMS and DLS). We observed that the spontaneous firing frequency of active SOM cells was significantly higher in the DMS (DMS 11.7 ± 0.3 Hz, $n = 6$ vs. DLS 6.3 ± 0.1 Hz, $n = 8$, $p < 0.0001$) (Fig. 5g, h), which is in accordance with the higher excitability of SOM cells observed in DMS (Fig. 5c, d).

Another explanation could be that the unitary synaptic weight of PV-SPN and SOM-SPN connections are not the same in DMS and DLS. To go deeper in the origin of the differential effect of PV and SOM cells, we thus performed paired patch-clamp recordings to measure the unitary synaptic weight of PV-SPN and SOM-SPN connections. We observed that unitary inhibitory postsynaptic potential (IPSP) amplitudes induced with a single presynaptic action potential (AP) were not significantly different between PV and SOM in DMS but significantly higher for PV-SPN connections in DLS ($p = 0094$) (Fig. 5i, j). The comparison with light-induced IPSPs indicate that about 3 to 4 interneurons are recruited by the opto-activation in our conditions for SOM-DLS and PV-DLS, 1 to 2 interneurons for PV-DMS and 8 to 10 interneurons for SOM-DMS. Therefore, when comparing PV and SOM cells in each territory, there is a higher amplitude for SOM cell opto-activation in DMS and PV cell opto-activation in DLS ($p = 0.0158$ for DMS and $p = 0.0019$ for DLS). These data show that, for each GABAergic population, there is no significant difference in the weight of their unitary connections in DMS and DLS. The differences of global inhibitory weight would thus result from the recruitment of more PV cells in DLS (due to their higher density in DLS) and more SOM cells in DMS (due to their higher excitability in DMS).

Altogether, these results show a marked heterogeneity in the GABAergic striatal microcircuits at the level of distribution, connectivity and electrophysiological properties, which could account for the territory functional specificity.

**SOM and PV cells mediate a dual effect on SPNs in DMS and DLS**. Due to their intrinsic properties, SPNs fire upon strong and correlated cortical activity while they remain mainly silent and operate in a large range of subthreshold activity in resting states[22–25]. We therefore questioned whether SOM and PV cells could show selective modulation of subthreshold SPNs between DMS and DLS, as we found with spiking SPNs. Using *SOM::ChR2* and *PV::ChR2* mice, we established the current–voltage relationship of light-induced IPSCs in SPNs. We applied a brief stimulation (5 ms duration) to evoke single APs in SOM and PV interneurons (Supplementary Fig. 6). In active states (depolarized potentials, ~−40 mV), we observed stronger light-induced currents for SOM opto-activation in DMS ($p = 0.0354$, $n = 12$ PV cells and $n = 17$ SOM cells) and for PV opto-activation in DLS ($p = 0.0232$, $n = 8$ PV cells and $n = 11$ SOM cells) (Fig. 6a), which is consistent with the differential inhibitory effect we previously observed (Figs. 3, 4). In resting states (at hyperpolarized potentials, ~−80 mV), we observed larger IPSCs after opto-activation of SOM-DMS and PV-DLS cells compared to PV-DMS and SOM-DLS cells ($p = 0.0057$ for DMS, $n = 12$ PV cells and $n = 17$ SOM cells and $p = 0.0057$ for DLS, $n = 8$ PV cells and $n = 11$ SOM cells). Therefore, PV and SOM cells exert differential effect regardless of the SPN activity.

Interestingly, at hyperpolarized states, GABAergic interneuron activation resulted in depolarizing currents (Figs. 5i and 6a–c). This is explained by the fact that (i) SPN membrane fluctuations widely cross $E_{Cl}-$ threshold (physiological $E_{Cl}$ ~−60 mV[18,26]) and (ii) GABA is hyperpolarizing for $E_h > −60$ mV and depolarizing for $E_h < −60$ mV. To investigate whether the depolarizing GABA mediated by PV or SOM interneurons also displays differential effect in DMS and DLS as with inhibitory GABA, we applied brief opto-stimulations and measured light-induced IPSPs in resting SPNs (maintained at ~−80 mV). In accordance with IPSC results (Fig. 6a), we observed significant differences between the different groups ($F_{3, 60} = 7.187$, $p = 0.0003$, one-way analysis of variance (ANOVA)). In DMS, SOM-induced IPSP amplitude recorded in SPNs was larger than PV-induced IPSP (SOM activation 2.56 ± 0.28 mV, $n = 19$ and PV activation 1.48 ± 0.31 mV, $n = 16$; $p = 0.0022$). In contrast, in DLS, PV cells had a stronger effect than SOM cells (PV activation 3.48 ± 0.52 mV, $n = 15$ and SOM activation 1.54 ± 0.13 mV, $n = 14$; $p = 0.0043$) (Fig. 6d).

These differential effects were confirmed using trains of light-induced APs in GABAergic interneurons (at 10 or 50 Hz, Supplementary Fig. 6). Because temporal patterns of presynaptic activity strongly influence synaptic transmission, with specific short-term dynamics, we characterized the short-term dynamics recorded in SPNs induced by trains of opto-activation of either PV or SOM cells. PV cells displayed a marked short-term depression in SPNs in DLS ($p = 0.0018$ and $p < 0.0001$ respectively) but not in DMS ($p = 0.0842$ for 10 Hz and $p = 0.3706$ for 50 Hz) (Fig. 6e). Conversely, SOM cells led to short-term depression for 10 and 50 Hz stimulation in DMS ($p = 0.0290$ and $p < 0.0001$, respectively) and only for 50 Hz trains in DLS ($p = 0.1337$ and $p < 0.0001$ for 10 and 50 Hz). The observed short-term depression suggests that the probability of release of connections from SOM-DMS to SPN and from PV-DLS to SPN is higher than SOM-DLS to SPN and PV-DMS to SPN. This observation is in line with the larger responses in SPNs evoked by opto-stimulation of SOM-DMS cells and PV-DLS cells (Fig. 4).

Altogether, these results further confirm the selective role of SOM and PV interneurons in the control of subthreshold SPN activity in DMS and DLS. They also point out that when SPNs are at resting states, GABAergic microcircuits can induce a strong depolarizing effect, which is also mainly mediated by either PV or SOM cells depending on the striatal territory.

**Differential shaping of subthreshold input integration and summation in DMS and DLS**. The depolarizing properties of

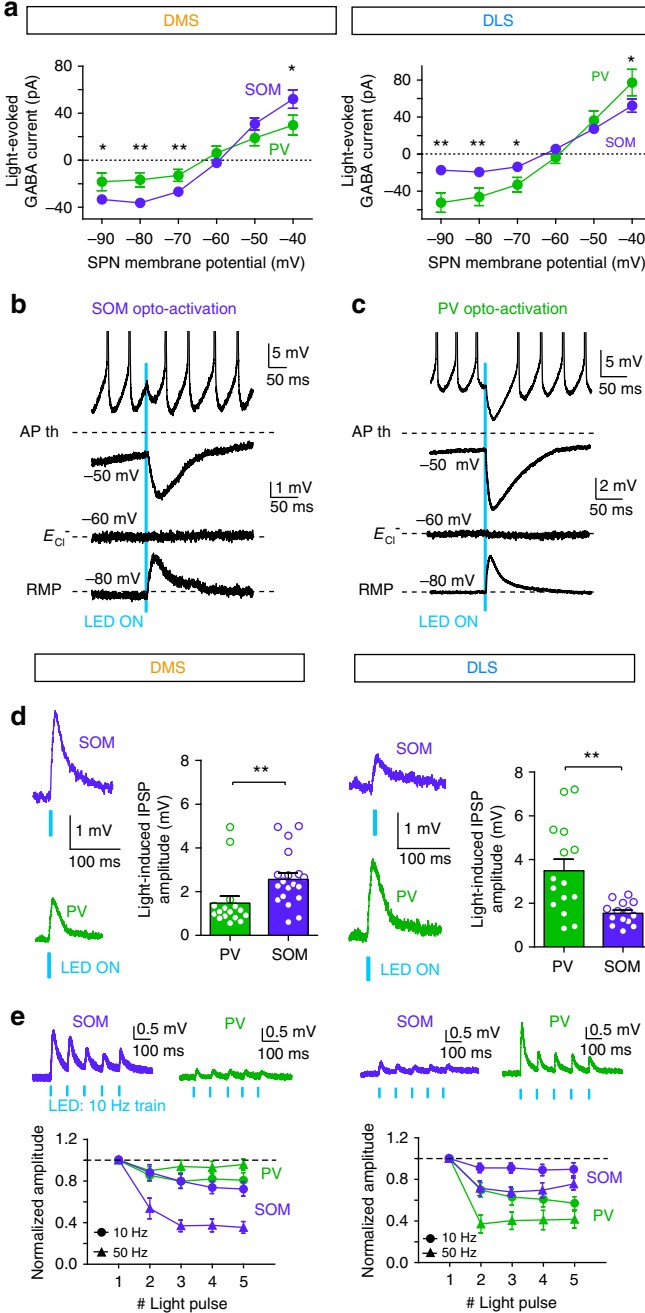

**Fig. 6** Dual effect of SOM and PV cells depending on the state of SPNs. **a** Current/voltage relationship of the light-evoked responses recorded in DMS and DLS after opto-activation of PV (green) or SOM (purple) cells. There are significant differences in the light-induced GABA currents between PV and SOM cells activation at −90, −80, −70 and −40 mV in DMS and DLS. **b**, **c** Representative traces of light-induced PSPs in SPNs evoked by short opto-activation (5 ms light pulses) of either SOM (**b**) or PV cells (**c**). Responses are recorded in SPNs held at different potentials (−80, −60 and −50 mV) or in spiking SPN (APs are truncated for clarity of the figure). **d**, **e** In DMS and DLS, synaptic weight of SOM or PV cells on SPNs were evaluated in SPNs maintained at their RMP (~−80 mV). Amplitudes of IPSPs induced by a single pulse (**d**) or trains of light pulses (**e**) were evaluated. **d** In DMS, single light pulse induces IPSPs with significantly ($p = 0.0022$) larger amplitude in SPNs after SOM cell opto-activation (purple traces, $n = 19$ cells) than after PV cell opto-activation (green traces, $n = 16$ cells). In DLS, opto-activation of PV cells ($n = 14$) induces significantly ($p = 0.0043$) larger IPSPs in SPNs than SOM cell opto-activation ($n = 14$). **e** Representative responses recorded in SPNs after opto-activation of SOM cells (purple traces) and PV (green traces) with 10 Hz trains of 5 ms light pulses. Short-term dynamics of IPSPs induced by trains of light pulses at 10 or 50 Hz after SOM cell (purple traces, $n = 7$ cells in DMS and $n = 6$ in DLS) or PV cell (green traces, $n = 9$ cells in DMS and $n = 6$ in DLS) activation. SOM-induced IPSPs display significantly stronger short-term depression at 10 ($F_{1, 50} = 8.94$, $p = 0.0126$, two-way ANOVA) and 50 Hz ($F_{1, 50} = 44.22$, $p < 0.0001$) in DMS compared to DLS. On the contrary, PV cells induce stronger short-term depression in DLS compared to DMS ($F_{1, 80} = 11.97$, $p = 0.0001$ for 10 Hz and $F_{1, 70} = 41.94$, $p < 0.0001$ for 50 Hz); *$p < 0.05$, **$p < 0.01$

This could be explained by the disynaptic nature of the feedforward inhibition onto SPNs (cortex-interneuron-SPN) that would delay the arrival of the GABA input until the decay phase of the monosynaptic cortico-SPN EPSP. These results show that SOM-DMS and PV-DLS efficiently shape single cortically evoked EPSPs in subthreshold SPNs by slowing down the EPSP decay phase and thus suggest they could affect the integration of subthreshold cortical inputs by SPNs.

We thus investigated whether the shaping of single EPSPs by SOM and PV interneurons could play a role in the summation of cortically induced subthreshold inputs in SPNs in DMS and DLS. To do so, different regimes of cortical activity inducing subthreshold activity in SPNs were mimicked with trains of cortical stimulations at various frequencies (5, 10, 20, 50 and 100 Hz). We compared the short-term dynamics of cortically evoked EPSPs in SPNs maintained at $E_m$ ~−80 mV either in control conditions or under selective opto-inhibition of SOM or PV cells in *SOM::Arch3* and *PV::Arch3* mice (Fig. 7 and Supplementary Fig. 10) or viral expression of AAV-Flex-ArchT-tdtomato in *SOM-Cre* and *PV-Cre* mice (Supplementary Fig. 11). In control conditions (no light), temporal summation was efficient for high frequencies (at 50 and 100 Hz) but not for medium frequencies (10 and 20 Hz). Synaptic depression could even be recorded for low-frequency activation (5 Hz) (both in DMS and DLS, Fig. 7f and Supplementary Fig. 10). Selective opto-inhibition of SOM or PV cells decreased EPSP summation in SPNs in DMS and DLS, respectively (Fig. 7d–g): opto-inhibition of SOM cells induced a decrease of temporal summation in DMS ($F_{1, 120} = 2.05$, $p < 0.0001$, $n = 25$ SPNs) but not in DLS ($F_{1, 80} = 0.11$, $p = 0.1536$, $n = 17$ SPNs), while opto-inhibition of PV cells following a 50 Hz train led to a decrease in the temporal summation of EPSP amplitude in the DLS ($F_{1, 100} = 2.10$, $p < 0.0001$, $n = 24$ SPNs) but not in DMS ($F_{1, 65} = 0.14$, $p = 0.1338$, $n = 14$ SPNs) (Fig. 7d, e). Similar results were obtained for different frequencies (5, 10, 20 and 100 Hz) of cortical stimulations (Fig. 7f, g and Supplementary Fig. 10). In addition, similar results were obtained with expression

GABA could have critical functional consequences on the integration of subthreshold inputs in SPNs at resting states. To assess whether SOM and PV interneurons could influence subthreshold integration in resting SPNs selectively in DMS and DLS, we investigated whether SOM and PV cells could shape cortically evoked EPSP in SPNs in response to a single cortical stimulation (Fig. 7a). Opto-inhibition of SOM and PV cells significantly decreased EPSP decay time (Fig. 7b, c) and this effect was differentially regulated by SOM and PV interneurons in DMS and DLS, respectively: opto-inhibition of SOM cells decreased decay time in the DMS ($p = 0.001$, $n = 20$) but not in the DLS ($p = 0.1007$, $n = 11$), whereas opto-inhibition of PV cells decreased EPSP decay in the DLS ($p = 0.001$, $n = 19$) but not in the DMS ($p = 0.9564$, $n = 16$) (Fig. 7b, c). No effect could be observed on the rise time, slope and amplitude of EPSPs after interneuron opto-inhibition, neither in DMS nor in DLS (Supplementary Fig. 9).

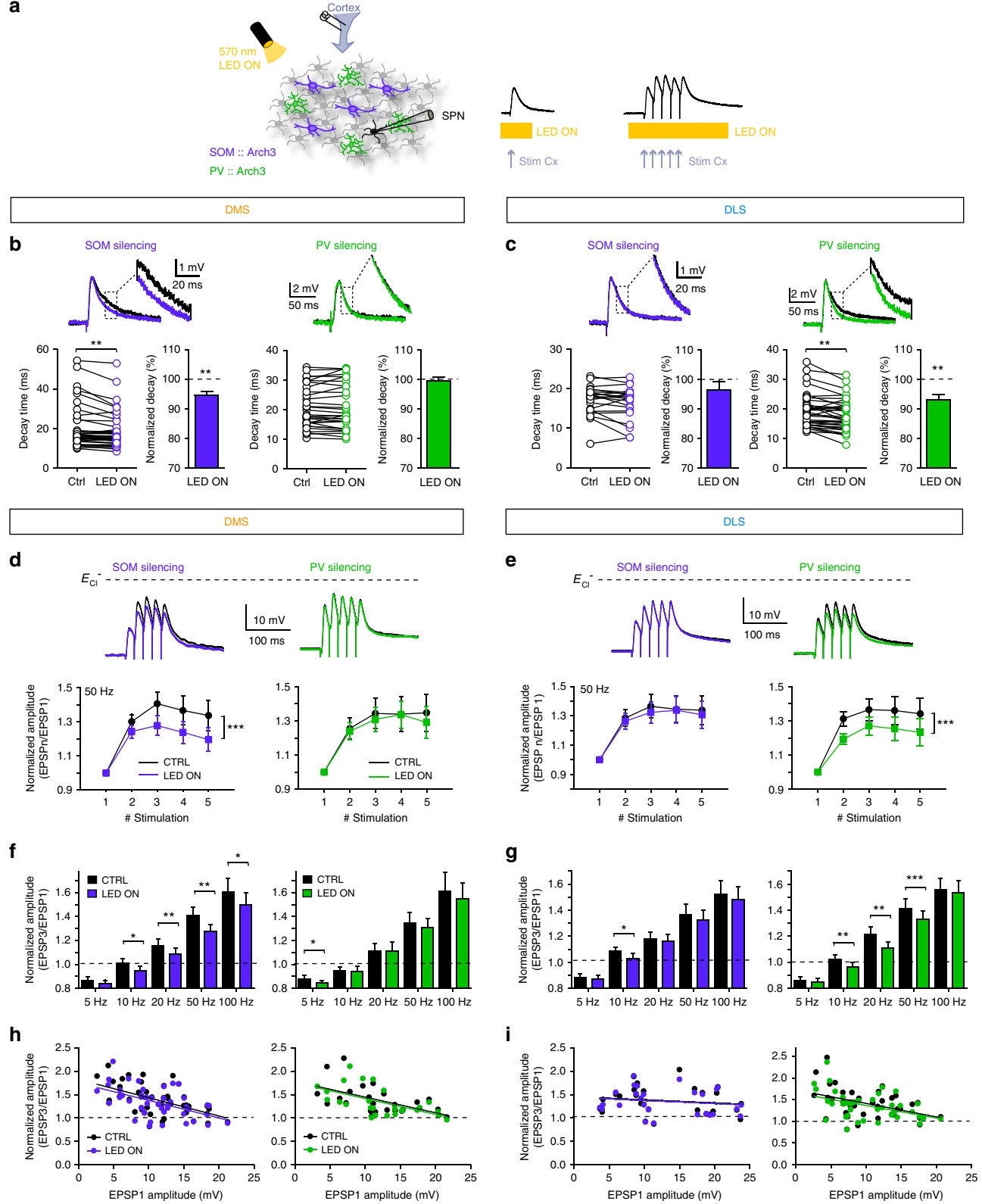

of Arch in SOM and PV cells restricted locally to DMS or DLS using viral strategy (Supplementary Fig. 11). Interestingly, silencing of SOM-DMS and PV-DLS cells had a stronger effect for medium frequencies (10, 20 and 50 Hz) and no or limited effect for low (5 Hz) and high frequencies (100 Hz) (Fig. 7), suggesting that the depolarizing effect of GABAergic

microcircuits preferentially favors the subthreshold summation of such medium frequencies of cortical activity.

Finally, we asked whether the number of cortical cells activated and the resulting variation in EPSP amplitude in SPNs could influence the effect of interneuron silencing on summation. In control conditions, the amplitude of EPSP1 significantly

**Fig. 7** SOM and PV cells favor subthreshold cortical activity summation. **a** In the DMS and DLS, cortically evoked subthreshold EPSPs or trains of EPSPs were recorded in SPNs in control or with opto-inhibition of SOM or PV cells. **b**, **c** Representative EPSPs recorded in DMS (**b**) and DLS (**c**) in control conditions (black) or with opto-inhibition of SOM cells (purple) or PV cells (green). Individual experiments (2 measurements for each neuron) and normalized (mean ± SEM) decay time of cortically evoked EPSPs in SPNs in control condition compared to opto-inhibition of SOM and PV cells. There is a significant decrease of EPSP decay time with opto-inhibition of SOM-DMS cells ($p = 0.001$, $n = 20$, paired-test) and PV-DLS cells ($p = 0.001$, $n = 19$). **d**, **e** Top: Representatives 50 Hz trains of EPSPs recorded in SPNs in control or with selective opto-inhibition of interneurons. Bottom: Temporal summations of EPSPs in SPNs after 50 Hz cortical electrical stimulation in control conditions (black) or with opto-inhibition of SOM (purple) or PV (green) cells. **d** In DMS, opto-inhibition of SOM cells induces a significant decrease in summation of EPSPs ($p < 0.0001$, $n = 24$, two-way ANOVA), whereas PV cell opto-inhibition has no effect ($p = 0.1338$, $n = 14$). **e** In DLS, PV cell opto-inhibition induces a significant decrease in summation of EPSPs ($p < 0.0001$, $n = 25$) and SOM cell opto-inhibition has no effect ($p = 0.1536$, $n = 17$). **f**, **g** Summary of the effects of interneuron silencing on temporal summation: ratio of the third EPSP compared to the first one for the different frequencies. **f** In DMS, SOM cell opto-inhibition strongly affects the summation of EPSPs in the trains for most activation frequencies (10, 20, 50 and 100 Hz), whereas there is no effect of PV cell opto-inhibition. **g** On the contrary, in DLS, PV cells strongly affect the summation of EPSPs in the trains for 10, 20, 50 and 100 Hz while silencing of SOM cells had no effect, except for 10 Hz activation. **h**, **i** Correlation of the normalized amplitude (EPSP3/EPSP1) and the amplitude of the first EPSP of the 50 Hz train, in DMS and DLS and for SOM and PV silencing. For all the conditions, there is a significant correlation, which is similar in control conditions or after interneuron silencing; $*p < 0.05$, $**p < 0.01$, $***p < 0.001$

influenced the summation (for 50 and 100 Hz) since it is more effective for smaller EPSPs but has no effect on the EPSP trains for other frequencies (5, 10 and 20 Hz) (Fig. 7h–i) in both DMS and DLS. The opto-inhibition of SOM or PV cells did not change this correlation regardless of the frequency (Fig. 7h–i). Thus, interneuron silencing would have the same effect for all tested amplitudes of the subthreshold EPSPs.

Altogether, these results show a physiological role for depolarizing GABA in the control of synaptic integration by SPNs that is selectively mediated by SOM and PV interneurons depending on the striatal region.

**In vivo selective modulation of cortically evoked SNr activity by striatal GABAergic circuits.** We finally investigated the outcome of the differential weight of SOM and PV interneurons in DMS and DLS, on the transfer of information between the (cortical) input and the (nigral) output of the striatum. In *SOM::Arch3* and *PV::Arch3* mice, we stimulated the cortical area (cingulate or somatosensory cortex) projecting to DMS and DLS, respectively, and recorded the evoked response of SNr units with and without opto-inhibition of SOM or PV cells in DMS or DLS (Fig. 8a–c). We observed a typical pattern of responses to cortical stimulation, consisting of three consecutive phases due to the activation of three cortico-SNr pathways[27–29]: (phase 1) an early excitation, corresponding to the activation of the "hyperdirect" non-trans-striatal pathway (cortico-STN-SNr), followed by (phase 2) an inhibition, corresponding to the "direct" trans-striatal pathway (cortico-striato-SNr), and (phase 3) a late excitation, reflecting the "indirect" trans-striatal pathway (cortico-striato-pallido-STN-SNr). This typical "triphasic" response was observed in SNr units, as well as other combinations of these three phases, with a similar proportion across conditions of the occurrence of "full" triphasic responses (37%, $n = 27$ units from 4 mice SOM-DMS; 48%, $n = 31$ from 4 mice PV-DMS; 35%, $n = 20$ from 4 mice SOM-DLS and 30%, $n = 20$ from 3 mice PV-DLS, $p = 0.5918$, Fisher's exact test). Similar proportions of occurrence were also observed for each individual phase when considered independently (Supplementary Fig. 12). For each SNr unit, we calculated the deviation from baseline activity (50 ms before stimulation, in the absence or presence of light) caused by the cortical stimulation, and measured the area of each phase (when present). Interestingly, the area of the inhibition phase (phase 2) was significantly reduced only when opto-inhibiting SOM interneurons in DMS ($-74 \pm 35$%, $p = 0.0481$, $n = 17$ units, paired t-test), and not in any of the 3 other conditions (PV-DMS: $p = 0.4971$, $n = 18$; SOM-DLS: $p = 0.4995$, $n = 18$; PV-DLS: $p = 0.4543$, $n = 15$) (Fig. 8d, e; Supplementary Table 2). Conversely, the area of the late excitation (phase 3) was significantly reduced

only when opto-inhibiting PV interneurons in DLS ($-859 \pm 308$%, $p = 0.0237$, $n = 9$ units, paired t-test), and not in any of the other conditions (PV-DMS: $p = 0.2174$, $n = 19$; SOM-DMS: $p = 0.3714$, $n = 19$; SOM-DLS: $p = 0.6949$ $n = 13$) (Fig. 8d, e). The duration of the inhibition phase and the peak of the late excitation phase displayed significant changes consistent with the area of each phase in SOM-DMS and PV-DLS conditions (Supplementary Fig. 12; Supplementary Table 2). In all conditions, the area of the early excitation (phase 1) was not affected by the opto-inhibition of either interneuron type in either territory (Supplementary Fig. 12; Supplementary Table 2) in line with the fact that this phase results from the activation of the non-trans-striatal pathway.

Altogether, these results indicate that the stronger effect of SOM interneurons in DMS and of PV interneurons in DLS translates into specific effect of these striatal interneurons/territory combinations on the trans-striatal transfer of information between the cortical input and the nigral output of the striatum.

## Discussion

In the present study, we describe a marked specificity in GABAergic circuit properties and in the control of SPN activity and their downstream consequences on SNr activity depending on the functional striatal territory (DMS vs. DLS). These results demonstrate a strong heterogeneity in the composition and function of GABAergic microcircuits in DMS and DLS. We show that SOM cells regulate SPN activity more efficiently in the DMS, while PV cells have a stronger weight in the DLS. In addition, we show that GABAergic interneurons regulate SPN activity in a dual manner: hyperpolarizing for suprathreshold SPN activity and depolarizing for subthreshold SPN activity. The depolarizing effect efficiently controls the integration of cortical input in the subthreshold range of SPN activity. Because the expression of the opsins is in about half of the SOM and PV interneurons (both for transgenic and virally expressed), the observed effect might be even stronger when 100% of GABAergic microcircuits are recruited. In addition, similar results obtained in transgenic mice and virally expressed opsins show that the differential effect of SOM and PV cells is strictly due to local striatal microcircuits and not to external SOM and PV projections recently described from the cortex or the globus pallidus[30–32].

In our study, we observed that the differential effect of PV cells might be due to their heterogeneous distribution throughout the striatum. The observed distribution in mice is consistent with previous studies showing a rostro-caudal gradient of PV cells in rats, monkeys and humans[33–36]. In addition, we observed here that PV cells have similar membrane and spiking properties in

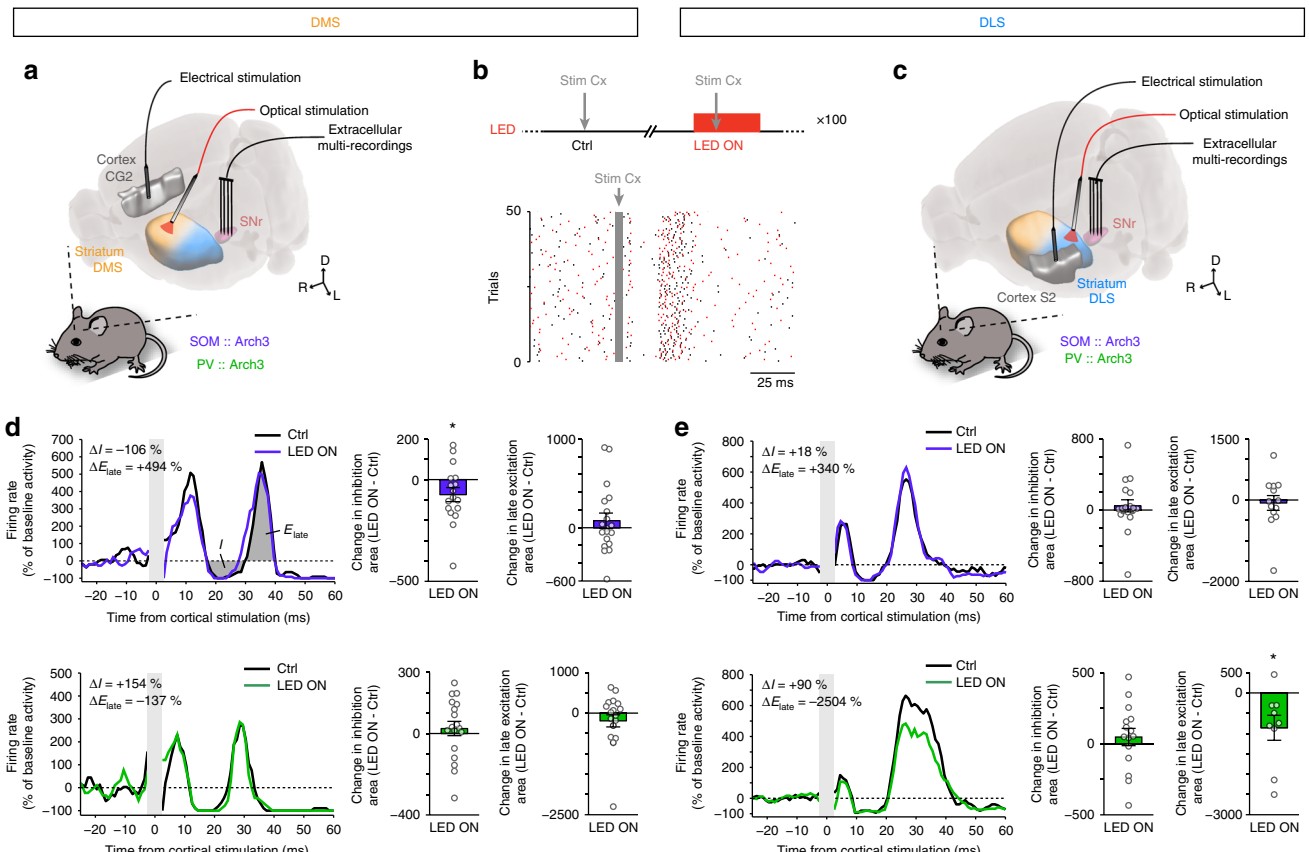

**Fig. 8** Selective control of cortico-nigral information transfer by SOM and PV cells. **a**, **c** Experimental set up: in vivo multi-channel extracellular recordings of SNr unit activity using 4-shank 32-site silicon probe in response to stimulation in CG2 (**a**) or S2 (**c**) cortex, while PV or SOM interneurons are opto-inhibited using an optic fiber implanted in the DMS (**a**) or DLS (**c**). **b** Top: A cortical stimulation is applied in the absence (left) or presence (right) of light (300 ms, 10 mW, stimulation 100 ms after light onset, cycle repeated at least 100 times, 2–6 s between trials). Bottom: Raster plot of a representative SNr unit recorded in the PV-DLS condition (unit in **e** bottom), showing the response to the cortical stimulation in interleaved Ctrl (black ticks) and LED ON (red ticks) trials. **d**, **e** Left: Activity of a representative SNr unit displaying a triphasic response upon cortical stimulation of CG2 (**d**) or S2 (**e**), in the absence of (Ctrl) or during the opto-inhibtion (LED ON) of SOM (top) or PV (bottom) interneurons in DMS (**d**) or DLS (**e**). The response is normalized to the baseline activity independently in each (Ctrl and LED ON) case. The measured areas of the trans-striatal phases (inhibition and late excitation) are illustrated in **d**, top, and the measured differences in area (LED ON – Ctrl) are indicated for each representative unit. Right: Change in response areas corresponding to the trans-striatal inhibition (left), and late excitation (right), in all units displaying the corresponding phase. The cortico-nigral inhibition phase is significantly reduced only by the opto-inhibition of SOM striatal interneurons in the DMS condition (**d**, top, $p = 0.0481$, $n = 17$ units, paired $t$-test), while the cortico-nigral late excitation phase is significantly reduced only by the opto-inhibition of PV interneurons in the DLS (**e**, bottom, $p = 0.0237$, $n = 9$ units, paired $t$-test); *$p < 0.05$. For the 3D images in **a** and **c**, brain and brain structures are captured from the Allen Institute for Brain Science's Mouse Brain Atlas[74] (© 2004 Allen Institute for Brain Science, Allen Mouse Brain Atlas available from: mouse.brain-map.org/) using Brain Explorer®2[75]

DMS and DLS; this observation being different from a recent study showing higher excitability of PV cells in DMS[37] (probably due to different experimental conditions such as composition of extra- and intracellular solutions and slice orientation). In contrast, despite their homogeneous density throughout the striatum, SOM cells exhibit distinct electrophysiological properties in DMS and DLS, more excitable in DMS than DLS. Functionally, PV cells, or fast-spiking cells, have been the most extensively characterized in the striatum. PV cells are known to exert a strong inhibitory weight on SPNs since they delay or even stop the spiking activity in SPNs[15,38,39]. SOM cells (also expressing NPY and NOS) are also able to delay AP in SPNs[15,38]; they were initially reported to have a lower connection probability with a weaker weight onto SPNs[40] compared to PV cells, but high amplitude of evoked responses of SOM-SPN connections after SOM opto-activation was recently reported[41,42], similar to what we observed in the present study. NPY expressing interneurons (overlapping with SOM cells) also strongly inhibit SPNs[43]. The differences in the characteristics of PV and SOM populations

between DMS and DLS would give rise to the territory specificity. Indeed, even though the unitary connections from PV to SPN or from SOM to SPN are similar in both territories, their global action as populations is higher for PV cells in DMS due to their density and to SOM cells in DMS due to their intrinsic properties. In addition, cross-comparison shows that unitary PV-SPN connections are stronger than SOM-SPN connections in DLS. Altogether, we propose that these properties are underlying the functional dichotomy of PV and SOM cells in DMS and DLS.

The differential localization of the synapses from SOM and PV cells on the SPN dendrites could contribute to the specific modulation of cortical inputs. Nevertheless, even though PV cells contact SPN closer to the soma than SOM cells, their contacts are both located within the first 250 μm of the SPN dendritic arborization[42], placing both of them in a strategic position to modulate the integration of glutamatergic inputs. In addition, we focused here on the two main subtypes of GABAergic interneurons but there are also other subtypes such as the electro-physiologically unidentified calretinin-expressing cells, and the

recently described tyrosine hydroxylase (TH)-expressing cells[43–45] that could also play a role in the modulation of SPN activity. Furthermore, SPN collaterals exert a feedback inhibition[46,47] though reported to be weaker than the feedforward inhibition[15,39,48]. Here, the differential effect between SOM and PV cells cannot be explained by the involvement of the feedback inhibition since we also observed such differential effect under subthreshold activity regime of SPNs (Fig. 7), which implies that feedback inhibition was not engaged in these conditions.

Our in vivo experiments show that local modulation of striatal interneurons influence the basal ganglia output. Our recordings of SNr spontaneous activity show an activation of SNr upon opto-inhibition of both interneurons in both territories, meaning that the net effect of SOM and PV cells on SNr spontaneous activity is inhibitory. This suggests that interneurons modulate the balance between the direct (inhibitory on SNr) and indirect (disin-hibitory) pathways, towards a relative activation of the direct pathway and/or inhibition of the indirect pathway, yet the striatal mechanism leading to this result could be multiple. Recent studies show that GABAergic interneurons contact both direct (dSPNs) and indirect pathway SPNs (iSPNs)[39,40,42], meaning that the local interneurons might control both SPN subtypes in a similar way. Since iSPNs are more excitable than dSPNs[49], their disinhibition would lead to a stronger increase in firing rate in iSPNs than dSPNs, thus explaining the observed overall increase of firing rate in SNr after silencing striatal interneurons. However, considering the dual depolarizing/hyperpolarizing effect of PV and SOM described here, the opposite hypothesis cannot be excluded: silencing striatal interneurons could lead to a decreased sum-mation of spontaneous cortical inputs, which, if stronger in dSPNs, would also shift the net balance towards an activation of SNr. Other members of striatal microcircuits are also likely involved in this effect, and could contribute to either a decreased activity of the direct pathway or an increased activity of the indirect pathway. The stronger effect of silencing PV than SOM interneurons on SNr spontaneous activity in both territories could be due to a stronger spontaneous activity of PV cells, classically described as "fast-spiking" and identified in vivo by their higher firing rates[11,36]. When stimulating cortical afferents, it is possible to visualize within the SNr different phases of response corresponding to hyperdirect, direct and indirect path-ways. Similar to local striatal control, we observed a specific modulation of cortically evoked SNr activity from SOM-DMS and PV-DLS. Interestingly, SOM-DMS seems to modulate the inhibition resulting from the direct pathway and PV-DLS the late excitation related to indirect pathway recruitment. Though it would be tempting to jump to the conclusion that SOM/PV interneurons only affect each direct/indirect pathway, respec-tively, we think the interpretation should be more careful since the activation of the direct and indirect pathways are overlapping, and the phases that can be measured, while mainly corresponding to each pathway, are still the sum of this overlap. Nevertheless, iSPNs are more excitable and are recruited faster with shorter responses than dSPNs, which maintain a longer activation in response to cortical stimulation[50]. The latency of interneuron activation is shorter for PV cells (Fig. 2 and coherent with kinetics described in[42]), which would be likely to modulate more effi-ciently the first recruited iSPNs while SOM cells would control more efficiently dSPNs, activated later. In addition, our obser-vation is also coherent with different organization of cortical inputs from cingulate cortex, which contact more dSPNs in DMS, and in DLS where iSPNs are more connected by somatosensory inputs[51]. In vivo recordings also show that the interplay of inhibitory and excitatory effects of GABA is complex on the resulting effect on output structures such as SNr. Both SOM-DMS and PV-DLS opto-inhibition lead to a decrease of a striato-

SNr pathway, meaning that the net effect of striatal interneurons would be excitatory locally on SPNs. A recent study also describes an overall decrease of SPN activity after silencing PV cells in vivo due to a disinhibitory loop from PV cells to NPY (overlapping with SOM cells) interneurons[52]. Another disinhibitory mechan-ism has been described between TH-positive interneurons and SOM cells[53]. A silencing of SOM might also lead to a decreased activity of SPNs, since SOM interneurons (but not PV) inhibit cholinergic interneurons[42] that in turn inhibit SPNs via their activation of NPY-NGF interneurons[54]. Therefore, cortical inputs are recruiting complex and various local striatal microcircuits whose interplay will lead to a complex effect combining inhibition and disinhibition. Local disinhibitory role of interneurons (such as vasoactive intestinal peptide or SOM interneurons in cortex or amygdala) has been recently highlighted in several brain struc-tures as an important mechanism to regulate the input/output flow of information[55].

Interestingly, we describe here that each interneuron subtype has a similar impact with a territory specificity, which translates into a differential effect on the downstream SNr. On cortically evoked activity, PV cells exhibit a strong weight in DLS while SOM cells more efficiently control DMS activity. We propose that potential underlying mechanisms to explain such specificity could come from the differences in the number of cells, their electro-physiological properties and the resulting local global con-nectivity. Therefore, the specificity stands in the fact that each GABAergic microcircuit, with its own intrinsic characteristics, has a specific role in DLS and DMS. The DLS is responsible for sensorimotor integration leading to habit formation. Sensory information requires to be quickly and reliably integrated and processed to produce a behavior adapted to the environmental stimuli. PV interneurons are reliably activated by the cortical activity, have fast-spiking characteristics and they modulate SPNs for any level of activity, which means that they tightly control sensory inputs of various amplitudes. Therefore, the intrinsic properties of PV cells and the fact that they are much more numerous in DLS[33,34] with denser arborizations[37] and their resulting action on SPNs are particularly adapted to the temporal precision needed to control sensorimotor information transmis-sion. The DMS is involved in associative functions, receiving mainly inputs from the frontal parts of the cortex[56]. Frontal cortex displays a lot of recurrent activity in the networks, parti-cularly during working memory tasks[57,58]. Fronto-corticostriatal inputs lead to recurrent activity in striatum that could be modulated by a more global inhibition, less precise in time but efficient to modulate network activity level. SOM cells are less dependent on cortical inputs to discharge and are not time locked. In addition, they more efficiently contol the first steps of build-up activity (for a small cortical activity) but then tend to lose their efficiency with increasing SPN activity. SOM cells would thus have the ability to drive the GABAergic modulation of associative integration. Therefore, the specificity of GABAergic microcircuits might be an active part of the different integration processes involved in the territory-specific striatal functions. We would like to point out that we focused here on feedforward inhibition mediated by cortical inputs but future studies should extend this work to another major striatal input coming from the thalamus.

A dual control of GABA on SPNs is functionally important because GABAergic interneurons would modulate in opposite direction sub- and suprathreshold events. The high amplitude of GABA currents we recorded for depolarized SPNs is coherent with a strong hyperpolarizing effect[15,38] and a depolarizing effect of GABA has also been previously described in striatum[59,60]. Of course, we cannot rule out that the depolarizing GABA could have a shunting effect and in that case decrease the efficiency of

cortical inputs to activate SPNs. We describe here a functional role of the depolarizing GABA favoring the integration of cortical inputs in the subthreshold range of SPNs. This is in accordance with previous models predicting that some depolarizing inputs coming when SPNs are at rest with $K_{IR}$ activated should shift the inactivation of $K_{IR}$, therefore reducing their availability and promoting summation of further inputs[61]. The GABAergic system would thus counterbalance intrinsic properties of SPNs by favoring depolarization in SPNs at rest and slowing them down when they reach the AP threshold.

Membrane potential variations displayed by SPNs are dependent on the vigilance state of the animals. During slow-wave sleep or deep anesthesia, SPNs display large up and down fluctuations[24]. The comparison of SPN fluctuations with various anesthetics and during wakefulness revealed that there are complex patterns of activity with depolarizing synaptic events of variable amplitude[25] and in a goal-directed sensorimotor task, a successful trial can lead to either sub- or suprathreshold activity in SPNs[62]. We also previously observed that subthreshold events can participate to Hebbian engram since they are involved in spike timing-dependent plasticity[63]. There is thus a large range of cortical activity which leads to subthreshold events in SPNs. Therefore, in addition to the modulation of the spiking activity, the role of PV and SOM cells in the modulation of subthreshold events in SPNs is functionally important. We show here that GABAergic inter-neurons strongly control integration of cortical activity in SPNs and are able to efficiently favor summation of cortical inputs. Given the properties of SPNs, the striatum might be a structure in which the depolarizing effect of GABA is particularly pronounced. Nevertheless, depolarizing GABA has also been described in other mature brain structures, such as cerebral cortex, hippocampus or amygdala[64–68]. These observations are widening the impact of the dual role of GABA within the brain. It is now of importance to consider that the impact of GABAergic circuits is dynamic and will be determined by the context of the activity of the targeted neurons.

## Methods

**Animals**. All experiments were performed in accordance with the European Union (EU) guidelines (directive 2010/63/UE) and local ethical committee. Animals were housed in temperature-controlled rooms with standard 12 h light/dark cycles and food and water were available ad libitum. Every precaution was taken to minimize stress and the number of animals used in each series of experiments. C57BL6 mice (*Mus musculus*) adult (>3 months old) for in vivo experiments and 4 to 6 weeks old for ex vivo experiments of both sexes were used (there was no significant differences between males and females). The transgenic lines (*SOM::ChR2* and *PV::ChR2*; *SOM::Arch3* and *PV::Arch3*) were obtained by crossing homozygous *SOM-IRES-Cre* mice (Stock 013044, Jackson Laboratory, ME, USA) and *PV-Cre* mice (Stock 008069, Jackson Laboratory) with homozygous ChR2(H134R)-eYFP mice (Stock 012569, Jackson Laboratory) or with homozygous Arch3-eGFP mice (Stock 012735, Jackson laboratory). The resulting offspring selectively express a channelrhodopsin2-yellow fluorescent protein (ChR2 (H134R)-eYFP) fusion protein or an archeorhodopsin3-green fluorescent protein (Arch3-eGFP) in either somatostatin- or parvalbumin-expressing neurons. In these transgenic mice, a specific expression of ChR2 or Arch3 was observed in a given cell type, which allows reliable activation (ChR2) or inhibition (Arch3) of PV or SOM interneurons (Supplementary Figs 1, 3, 5 and 6).

**In vivo optogenetic and extracellular multi-channel recordings**. For surgery, general anesthesia was induced with isoflurane inhalation, and maintained with urethane injection (1.3 g/kg, intraperitoneal (i.p.) initial injection, supplemented to reach a surgical plane of anesthesia for each mouse up to 2 g/kg). Body temperature was kept constant with a heating pad. The skull was exposed and craniotomies were drilled over stereotaxic locations of the SNr, and a combination of DMS and/or DLS and/or cortex depending on the experiment. All implantations were performed on the same hemisphere, thanks to angled insertions to reduce cumbersomeness (all coordinates are relative to bregma, except depth to brain surface). Two holes were drilled over the cerebellum to insert the ground and reference miniature screws. In the SNr (anterior-posterior (AP) −3.1 mm, medial-lateral (ML) 1.2–2.0), a multi-channel silicon probe (32 sites, 4 shanks with 200 μm spacing, high density buzsaki32 design, NeuroNexus) was inserted to 3.7 mm depth

and slowly lowered to find typical SNr unit pattern of activity (regular spiking, >10 Hz; recording depths: 3.8–5.2 mm). In a first group of mice (n = 5 SOM::Arch3 and n = 4 PV::Arch3 mice), 2 flat-cleaved 200 μm core, 0.39 NA diameter optic fiber cannulas (Thorlabs) were implanted in the DMS (AP +1.2 mm, ML 1.5 mm, depth 2.1 mm) and DLS (AP +0.5 mm with 10° angle towards the caudal pole, ML 2.85 mm, depth 2.3 mm). Two more groups of animals (n = 12 for DMS group and n = 9 for DLS group) were implanted with a bipolar concentric stimulation electrode (FHC) in the cortex and an optic fiber cannula in the striatum. In the DMS group (n = 8 SOM::Arch3 and n = 4 PV::Arch3 mice), the cingulate cortex was targeted for stimulation (AP +1.2 mm, ML 0.5 mm with 3° angle towards the midline and the caudal pole, depth = 1.6 mm), and the DMS for illumination (AP +1.2 mm, ML 2.1 mm with a 15° angle insertion towards the midline, depth = 1.85 mm). In the DLS group (n = 4 SOM::Arch3 and n = 5 PV::Arch3 mice), the secondary somatosensory cortex was targeted for stimulation (AP +0.1 mm, ML + 4.1 mm with a 3° angle insertion away from midline, depth 2.3 mm), and the DLS for illumination (AP +0.5 mm with a 10° angle towards the caudal pole, ML 2.6 mm with a 3° angle away from the midline, depth 2.3 mm). A paraffin wax mixture was used to seal the craniotomies. Before implantation, silicon probes were painted with 2% DiI solution (Sigma), and optic fibers and stimulation electrodes were painted with 4′,6-diamidino-2-phenylindole (DAPI) solution (5 μg/mL, Molecular Probes) to facilitate the ex vivo localization confirmation.

For optogenetic and electrical stimulation, light from a 635 nm laser diode lightsource (FLS-635 nm–50 mW; DIPSI) collimated into a custom patch cord (105 μm core, 0.22 NA; Thorlabs) was connected to the brain-implanted optic fiber cannulas. Light was delivered in square pulses with a light power of 10 mW at the tip of the fiber, adjusted before implantation in each experiment for both optic fibers using a photodiode power sensor coupled to a power meter (S130C and PM100USB; Thorlabs). Electrical stimulation was delivered using an isolated pulse stimulator (Model 2100, A-M Systems) delivering bipolar biphasic pulses of 200 μA during 0.5 ms. Both light and electrical stimulations were driven by a Master-9 pulse stimulator (AMPI, Science Products). In mice with 2 optic fibers in DMS and DLS, only light stimulation was used (1 s at 0.2 Hz, 50 repetitions). In cortically stimulated mice, we alternated 3 types of trials: electrical stimulation only, electrical stimulation with light (300 ms light pulse, 100 ms delay between the onset of light) and light only (300 ms), each repeated at least 100 times with 2–6 s between any type of trial.

For electrophysiological recordings, extracellular signal was amplified, multiplexed and acquired continuously at 20 kHz using a multi-channel KJE-1001 system (Ampliex) and stored for offline analysis.

Histological processing was used to confirm probe and optic fiber locations, the brain was removed at the end of the experiment, post-fixed overnight with 10% (vol/vol) formaline solution and cryoprotected in 30% (wt/vol) in phosphate-buffered saline (PBS) sucrose solution. Coronal slices (80 μm thick) were cut using a freezing microtome (Microm), permeabilized with Triton X-100 (0.2%), counterstained with fluorescent Nissl staining (Neurotrace, Life Technologies) and mounted in Fluoromount (Southern Biotech). Probe and optic fiber tracks were visualized and photographed, aided by DiI and DAPI, on a DMRB (Leica) microscope. Six mice where the stimulation electrode touched the corpus callosum were excluded from the analysis of cortical stimulation responses, but used for analysis of light-only trials.

For data analysis, data were visualized and processed using NeuroScope and NDManager[69] (http://neurosuite.sourceforge.net). For unit detection, the signal was high pass filtered using a median filter, spikes were extracted and single units were isolated using the semi-automatic spike classifier KlustaKwik (http://sourceforge.net/projects/klustakwik) and further refined manually using the graphical spike sorting application Klusters (http://neurosuite.sourceforge.net). Only units of good quality (<1% spikes in the refractory period) and displaying typical SNr features (frequency >10 Hz, CV of interspike interval <1) were considered for analysis. Data were analyzed by built-in or custom-built procedures in MATLAB (MathWorks) and R environment (R version 3.4.3, RStudio 1.1.383). To quantify the effect of light on spontaneous activity, results from the first 50 light-only trials from electrically stimulated mice were pooled with results from fiber-only mice (analyzing the 300 ms at the onset for the fiber-only mice). To quantify the effect of light on cortical stimulation-evoked response, for each unit and type of trial (with or without light), we calculated a 1 ms-binned post-stimulus time histogram of the response, normalized by the baseline activity (mean of 50 ms before stimulation), and smoothed over 5 ms. Data within 2 ms of the stimulation were not considered in the analysis because the stimulation artifact affected spike detection in this time window (indicated with a gray shaded area on Fig. 8). On peristimulus time histogram (PSTH) shown in Fig. 8, cortico-SNr responses corresponding to typical phases of the "triphasic" response were identified as follows. (1) Only responses within the first 50 ms were considered. (2) Excitations were considered if the peak crossed +50% of baseline activity in either or both of Ctrl or LED ON trials, inhibition threshold was −33%. (3) For area and duration measurements, onset and offset were defined as the bin where the PSTH crossed the baseline activity (0%), except for cases where two excitations were present without the inhibition, leading to two clearly separated peaks without crossing the baseline: in this case the trough between peaks was used as the offset of the early excitation (phase 1) and the onset of the late excitation (phase 3). If the onset of a component was masked by the stimulation artifact (since 2 ms on each side of the artifact are excluded from the analysis), it was taken as the first analyzable bin (i.e.,

the bin centered on 2.5 ms). (4) Response that were not clearly defined, such as excitations that could not be classified as early or late based on the peak latency or the presence of other identified phases, were not included in the analysis.

**Ex vivo optogenetic and multi-patch-clamp recordings.** For surgery for viral injections, stereotaxic intracranial injections were used to deliver adeno-associated viruses (AAVs) carrying opsins for ex vivo electrophysiological recordings in *SOM-IRES-Cre* and *PV-Cre* mice. Mice (P20–25) were anesthetized with isoflurane and placed in a stereotaxic frame. A small hole was drilled in the skull and 400 nL of AAVs (serotype 5/9) encoding either floxed ArchT-tdtomato or floxed ChR2 (H134R)-mCherry (>10[12] genomic copies per mL, UPennCore) was injected through a pulled glass pipette using a nanoinjector (WPI). The injections targeted the DMS (AP +1.1 mm, ML 1.2) or the DLS (AP +0.4, ML 2.3). After surgical procedures, mice returned to their home cages for 15–20 days to allow a good expression (Supplementary Fig. 4) and brain slices were then prepared. We observed similar levels of expression of both Arch and ChR2 in PV and SOM cells in transgenic mice and mice injected with viral vectors (Supplementary Figs. 1, 4 and 5).

For brain slice preparation, connections between the somatosensory cortex (layer 5) and the striatum are preserved in a horizontal plane. For somatosensory territory study, horizontal brain slices containing the somatosensory cortex and the corresponding corticostriatal projection field were prepared according to the methods previously described[70]. Concerning the associative territory study, we set up a new corticostriatal slice (para-sagittal with a 30° angle) preserving the connections between the cingulate cortex and the striatum, based on three-dimensional (3D) anatomical reconstruction of such connections[56]. We chose to use these two different slices to be able to preserve and stimulate specific cortical inputs to either DMS or DLS. In addition, both slice orientations do not preserve thalamic inputs, allowing a specific activation of only corticostriatal inputs from layer 5 pyramidal cells. Animals were anesthetized with isoflurane before extraction of the brains. We prepared either brain slices (300 μm) using a vibrating blade microtome (VT1200S, Leica Microsystems, Nussloch, Germany). Brains were sliced in a 95% $CO_2$ and 5% $O_2$-bubbled, ice-cold cutting solution containing (in mM) 125 NaCl, 2.5 KCl, 25 glucose, 25 $NaHCO_3$, 1.25 $NaH_2PO_4$, 2 $CaCl_2$, 1 $MgCl_2$ and 1 pyruvic acid, and then transferred into the same solution at 34 °C for 1 h and then moved to room temperature.

For electrophysiological recordings, whole-cell patch-clamp recordings (at 34 °C) of SPNs, PV interneurons (fast-spiking cells) and SOM interneurons (persistent and low-threshold spiking cells) were performed with borosilicate glass pipettes (5–8 MΩ) containing (mM): 127 K-gluconate, 13 KCl, 10 HEPES, 10 phosphocreatine, 4 ATP-Mg, 0.3 GTP-Na and 0.3 EGTA (adjusted to pH 7.35 with KOH). Slices were continuously superfused with the extracellular solution containing (mM): 125 NaCl, 2.5 KCl, 25 glucose, 25 $NaHCO_3$, 1.25 $NaH_2PO_4$, 2 $CaCl_2$, 1 $MgCl_2$ and 10 μM pyruvic acid bubbled with 95% $O_2$ and 5% $CO_2$. The reversal of chloride was set to be at the physiological value (~−60 mV) based on previous measures performed in SPNs with methods avoiding any ionic perturbations[18,26]. Slices were visualized under a microscope (Slicescope Scientifica, London, UK) using a 5×/0.15 objective for the placement of the stimulating electrode and a 40×/0.80 water immersion objective for localizing cells for whole-cell recordings. SPNs were distinguished from interneurons based on passive and active membrane properties[71,72] (Supplementary Table 1). Signals were amplified using EPC10-2 amplifiers (HEKA Elektronik, Lambrecht, Germany). Current-clamp recordings were filtered at 2.5 kHz and sampled at 5 kHz and voltage-clamp recordings were filtered at 5 kHz and sampled at 10 kHz using the program Patchmaster v2x32 (HEKA Elektronik).

Concerning cortical stimulation protocols, electrical stimulations were applied with a bipolar electrode (Phymep, Paris, France) placed in the layer 5 of either the somatosensory or the cingulate cortex. Electrical stimulations were monophasic at constant current (Iso-Flex, AMPI, Science Products). Single electrical stimulations or trains of stimulations were delivered; trains consist of 5 stimulations delivered at different frequencies (5, 10, 20, 50, 100 Hz). Single or trains of stimulations were applied at 0.1 Hz, a frequency for which no short- or long-term changes are observed[70,73]. Single stimulation duration was 100–150 μs. Evoked EPSP amplitude ranged from 1 mV to 30 mV. SPNs were held at their physiological membrane potential, in average −77.0 ± 0.3 mV (*n* = 140) and there was no statistical difference in the holding membrane potentials between the different experimental conditions.

For optogenetic stimulations, collimated LEDs placed at the back of the microscope generated wide-field stimulations through a water immersion objective 40×/0.8 NA. Activation of Arch3 was made at λ = 570 nm and consisted of long light pulses (300 ms to 1 s duration, 20.3 mW/mm²), delivered with a minimum interval of 8 s. Arch activation led to a strong and reliable silencing of both PV and SOM interneurons (Supplementary Fig. 3). Activation of ChR2 was made at λ = 473 nm and consisted of either a single square pulse of light of 5 ms duration (23.2 mW/mm²), or of 10, 20 or 50 Hz trains of five light pulses with similar intensity, 5 ms duration, delivered with a minimum interval of 8 s. ChR2 activation led to reliable spiking activity in both PV and SOM interneurons (Supplementary Fig. 6).

Data analysis was performed as following. Electrophysiological properties of PV cells, SOM cells and SPNs were quantified as follows: input resistance was measured by repeated current injections (−20 pA, 500 ms) and frequency was measured for current step +40 pA above AP threshold. The spike probability was measured as the occurrence of a single action potential induced by a single electrical stimulation of the cerebral cortex. For each cell, we repeated 6 to 8 control and LED ON interleaved trials to calculate the spike probability in each condition. To measure the effect of light on SPN spiking frequency, we triggered stable spike trains with 1 s current injection and measured either (1) the instantaneous frequency (Fig. 4b) by comparing the frequency between 2 spikes right before the 5 ms LED pulse and at the time of the LED pulse or (2) the averaged frequency within 300 ms before, 300 ms during and 300 ms after the 300 ms LED pulse (Fig. 4c). We assessed short-term dynamics of either light-induced IPSPs or cortically evoked EPSPs with trains of stimulation (light pulses or electrical stimulations). We measured both short-term depression/potentiation and temporal summation. For the short-term plasticity, we measured the amplitude of IPSP/EPSP induced by each stimulation in the train and normalized it to the amplitude of the first one. For temporal summation, we measured the total amplitude of each IPSP/EPSP (from baseline to the peak of the response) and normalized it to the amplitude of the first IPSP/EPSP. For the effect of interneuron silencing on EPSP summation (Fig. 7), the normalized amplitude corresponds to the ratio between the third EPSP of the train compared to the first one. We compared the effect for each EPSP of the train and did not observe significant differences, and hence the third one was chosen as a representative. Data analysis was carried out in Igor-Pro 6.0.3 (Wavemetrics, Lake Oswego, USA) with the Neuromatic package and Fitmaster (HEKA Elektronik, Germany).

**Statistical analysis.** Statistical analysis was performed using Prism 5.0 (GraphPad, San Diego, USA) and R environment (R version 3.4.3, RStudio 1.1.383). The sample size for the different sets of data is mentioned in the text or in the respective figure legends. Normality of each data set was checked using D'Agostino and Pearson's test. Unless otherwise stated, all data sets are normal, reported as mean ±SEM, and statistical significance was assessed using Student's *t*-test. Non-normal data sets are reported as median (interquartile range) (the use of the median is indicated in the text or figure legend to signal the non-normality), and statistical significance was assessed using Mann–Whitney's *U*-test and Wilcoxon's signed rank test for unpaired and paired data, respectively. Two-sample Kolmogorow–Smirnov test was used to compare the distribution of SNr unit firing rates in *PV::Arch3* vs. *SOM::Arch3* mice. To analyze the effect of light on SNr unit spontaneous activity, we used generalized linear mixed models (R package lme4 and lsmeans) to take into account the partial pairing of the data (mice with both DMS and DLS fibers). For comparing the proportions of modulated units, each SNr unit was individually classified as significantly modulated or not, in each tested condition (two-tailed Wilcoxon signed rank test, 300 ms before light, and 300 ms during light, *n* = 50 trials). This binary result (modulated = 1; non-modulated = 0) constituted the dependent variable, Fmod. Fmod was modeled by the factors "interneuron" (two levels: SOM and PV), "territory" (two levels: DMS and DLS) and their interaction, with a random effect (identity of the unit, allowing to take into account the partial pairing for territory), i.e., Fmod ~interneuron × territory + (1|ID_unit). This was performed in R, using the glmer function of lme4 package, with a probit link function, followed by post-hoc tests correcting for false discovery rate using the lsmeans function. The comparison of the effects on the median change in firing rate of all units was similarly analyzed using a linear mixed model (lmer function, FChange ~interneuron × territory + (1|ID_unit)) applied on the transformed data (using a signed 2/3 root transform $y = sign(x$-median$(x)) \times abs(x$-median$(x))^{(2/3)}$) to maximize the normality of the model residuals. One-way ANOVA was used to compare all the effects together of PV and SOM cells in DMS and DLS. Pearson's correlation was used for relationship between inhibitory effect and firing frequency discharge. Two-way ANOVA followed by Bonferroni post-hoc test was used to compare input/output curves, differences in short-term plasticity and to quantify the effect of silencing interneurons on temporal summation.

**Immunohistochemistry.** Quantification of interneurons and expression of Arch and ChR were done by immunohistochemistry targeting parvalbumin and soma-tostatin proteins and GFP/YFP or tdtomato/mCherry expressed together with the opsins. Mice were anesthetized with a mixture of ketamine (100 mg/kg) and xylazine (10 mg/kg) injected i.p., then transcardiacally perfused with first PBS and finally with 4% paraformaldehyde. Following perfusion, brains were post fixed in 4% paraformaldehyde for 16 h, extensively washed and sliced into 60 μm coronal sections using a vibratome (Microm HM 650V, ThermoScientific). Immuno-staining was performed on free-floating sections from right and left hemispheres using chicken anti-GFP (1:1000; Abcam AB13970, lot #GR89472-7/12/16), rat anti-SOM (1:200, Millipore MAB354, lot #2326265, 2739496, 2933843) and rabbit anti-PV (1:1000, Swant Technologies P25, lot #510, 2014), and the appropriate fluorescent-labeled secondary antibodies (Alexa 568 for PV and SOM and Alexa 488 for GFP). Sections were mounted on slides with VectaShield (Vector Labs).

Images were acquired using an SP5 confocal microscope (Leica, Germany). Small (30–40 μm) Z-stacks with 1 μm step size were acquired in DMS and DLS with a 20× objective (0.8 NA). For each animal, we acquired Z-stacks on 3 slices per territory (DMS and DLS) and in both hemispheres, so 6 Z-stacks per condition. We used Fiji software for analysis. Z-stack maximum projections were applied and quantification of neurons was performed in regions of interest (ROIs) of identical size in DMS and DLS. The number of cells was reported to the volume

corresponding to the ROIs. The sparseness of the striatal interneurons allowed us to easily count the number of cells in a given volume and to have unbiased analysis, the counting was performed by experimenters who were blind to the experimental conditions.

**Data availability**. The data sets generated during and/or analyzed during the current study are available from the corresponding author on reasonable request.

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

## Acknowledgements

The authors thank A. Di Nardo and A. Bacci for kindly providing *PV-Cre* and *SOM-Cre* mice and K. Deisseroth for sharing optogenetic construct which was obtained from UPenncore facility. The authors also thank P. Trifilieff, E. Valjent, M. Cazorla, D. Peterka, M. Albrieux and P.F. Mery for careful reading of the manuscript and helpful comments. The authors would also like to thank M.L. Kemel, P. Mailly and J.M. Deniau for help in setting up the DMS brain slice preserving connections and F. Appaix, G. Gangarossa, A. Foncelle, A.M. Godeheu, C. Piette and G. Zalcman for assistance. This work has been supported by grants from FP7 Marie Curie European Research Program (E.F.), Fondation pour la recherche sur le Cerveau/Rotary Club (L.V.), Agence Nationale de la Recherche program (F.S.), CNRS (E.F.) and INSERM (F.S. and L.V.). M.V. was supported by a fellowship from Fondation de France and S.P. by College de France.

## Author contributions

E.F. conceived and supervised the whole study, designed experiments, performed and analyzed ex vivo experiments and wrote the manuscript; M.V. performed and analyzed in vivo experiments; S.P performed ex vivo recordings; F.S. contributed reagents; and L.V. designed in vivo experiments and wrote the manuscript.

## Additional information

**Competing interests:** The authors declare no competing interests.

