## [Peer Review File · Nature Communications]

Reviewers' comments:

Reviewer #1 (Remarks to the Author):

The manuscript describes electrophysiological and immunohistochemistry experiments which characterize the action of two, PV and SOM-expressing, striatal GABAergic interneuron actions on the activity of spiny projection neurons of the dorsomedial (DMS) and dorsolateral (DLS) striatum. Although both interneurons have effects on SPN activity in both striatal subregions, the data shows a more pronounced effect of PV interneuron activity on DLS SPN neurons and SOM interneurons on DMS SPN neurons. Differential depolarization by these GABAergic inputs is also shown. Altogether these results provide new evidence of a role of GABA feedforward inhibition performed by two types of interneurons in controlling cortical synaptic integration by SPNs. These findings are intriguing and the authors are to be commended for taking on a project of this scope. Many of the experiments are rigorous and the data analysis is appropriate for the most part. Unfortunately, there are serious problems with several aspects of experimental design, some incomplete analyses, and a failure to consider new findings that complicate the interpretation of key experiments.

Major Concerns:

1) The authors mentioned in their discussion that there are other local GABAergic interneurons that could also alter SPN activity, but they failed to consider the impact of GABAergic afferents that innervate striatum on the interpretation of their findings. One example is the external part of the Globus Pallidus which sends a strong GABAergic input to SPNs and also to interneurons in the Striatum (see Glajch et al, 2016 and Corbit et al, 2016). Although the traditional arkypallidal neurons (pallidostriatal projections) do not express PV, some prototypical GPe neurons, which express PV, also project back to striatum and innervate SPNs, PV interneurons and NPY interneurons (Saunders et al, 2016). Another example is the GABAergic cortical projections to striatum. Cortical SOM neurons send projections to striatum (Rock et al., 2016) and this was further characterized in a very recent paper showing how SOM and PV cortical neurons innervate different cell types in the striatum (Melzer et al., published on May 2, 2017 in Cell Rep., although it is understandable that the authors were not aware of this manuscript at the time they submitted this paper). The presence of these afferents strongly confounds the experiments, because the authors used a breeding scheme between homozygous Cre lines and the rhodopsin mouse lines. ChR2 or Arch3 will be expressed in the terminals of the pallidal and/or cortical inputs, and could account for some of the effects observed with optogenetic activation or inhibition in these lines. The experiments should be compared with experiments using viral injections into the striatum of Cre-expressing adult mice to confirm that the opto-evoked results are due selectively to the striatum interneurons and not to other GABAergic projections arriving in the striatum.

2) There are problems with the in vivo experiments. First, performance of the experiments under anesthesia (which is not mentioned in the results and should be) makes it difficult to interpret how these circuits would work if examined during behavior. SPNs show little activity in anesthetized animals, and thus the impact of disinhibiting them in this state may be very different from what would be seen in the awake animal. This may account for the very small changes in SNr firing. The authors might consider using awake mice or providing cortical stimulation to mimic synaptic activation in this preparation. Recording from SPNs might also provide information on the actual disinhibition within striatum. The authors claim that the in vivo experiments indicate that distinct interneuron populations in different subregions of the striatum exert differential effects on the activity of SNr units. This is not what the first figure shows. Only the PV neurons had a significant impact on SNr firing if they were inhibited in DLS. SOM neurons do not have a differential effect on SNr activity when inhibited in the DMS or DLS. It is intriguing that although the authors found pronounced segregation of the effect of

PV and SOM between DLS and DMS in slices, this was not recapitulated in the recordings in SNr. Maybe the SOM difference could be observed in other striatal output targets such as GPe or GPe. It will be very helpful to add experiment to understand this issue.

3) The electrophysiological experiments in supplementary figures 3 and 4 are helpful, but incomplete. The paired recordings were only performed for the PV neurons (the authors refer to an experiment in SOM neurons on page 6, but it is not in the figure), and thus the estimate of how many SOM neurons were activated may not be accurate. The authors do not indicate in which striatal subregion the recordings in supplemental figures 3C, 3D and 4C were performed.

4) The cell counting in the different regions should be performed using unbiased stereological techniques, and it does not appear that this was the case. Also, regarding the data in supplementary figure 1, is there any overlap between the SOM and PV staining or between the GFP markers for these two neuronal subtypes (presumably not)?

5) Figure 5e, it may difficult to estimate the depression of IPSCs if the amplitudes are very small (e.g. as in the PV-DMS data). In particular, the amplitude of the later IPSCs may be hard to measure given the signal/noise ratio and the elevated baseline due to previous IPSCs.

6) Why did the authors use homozygous Cre mice for the breeding schemes? There are some reasons homozygous Cre should be avoided: Cre alone can produce phenotypes by affecting cell physiology or Cre insertion could dysregulate some specific genes which may change the cell physiology by itself. Cre could be express in germline cells and in homozygous breeding schemes there is a bigger change to cause germline recombination (may happen in PV Cre males because PV is expressed in the sperm).

7) The PV Cre and SOM Cre mice used in the study originated in different strains of mice (PV in 129P2/OlaHsd and SOM in C57BL/6 x 129S4/SvJae). Were the animals backcrossed sufficiently onto a homogeneous background (e.g. C7Bl6J) to allow direct comparison between those two lines?

8) Page 10, 1st paragraph, was the rheobase always calculated from the resting potential or were membrane potentials adjusted with constant current injection to set the potential at a particular resting value?

9) Bottom of page 8, the authors discuss the correlations shown in figure 3d and note a negative correlation between SPN firing rate and the effect of SOM inhibition on this rate. However, the data are plotted in such a way that the correlation appears to be positive. This appears to be due to the way the delta in firing was calculated/plotted. The authors should also provide the r or r^2 values for all correlations. Otherwise it is difficult to determine how much of the variability is accounted for by the correlation of the two variables.

10) The age of mice used for the in vivo and ex vivo experiments is very different. Is there any change in the pattern of PV or SOM promoter expression between those ages that could affect the results?

11) SNr units were identified as GABAergic neurons by having a firing rate of at least 10Hz or higher, but the authors showed an median \pm interquartile range of 17 ± 10 Hz (page 5 line 93) which may indicate that there were a few neurons with less than 10Hz. Authors should clarify this.

12) It's unclear how the Confocal Z-Stack was performed. What was the optical section used and how many stacks were reconstructed for the analyzed images?

13) What statistical test was used for comparisons of data in figure 1e? Was this a Fisher's exact test or chi square? Same question for the data in figures 2e and f.

14) As the authors are no doubt aware, depolarizing GABA responses can still inhibit neurons by current shunting and stabilizing the membrane potential at EGABA which is generally subthreshold. It is surprising that these concepts and their potential influence on the findings were not discussed.

15) The cellular location of synapses made by PV and SOM interneurons may influence their effectiveness in SPN depolarization and inhibition. PV neurons generally synapse near the soma, but is this also the case for SOM neurons? This issue should be discussed.

Minor points:

For the in vivo data, authors should not refer to the data as neurons but as units.

The lot numbers for the antibodies should be mentioned.

Page 3, 1st paragraph, the authors should also cite the newer findings from connectome studies using viral tracers to map corticostriatal inputs in more detail.

Page 3, 2nd paragraph, both hypotheses about the factors that account for functional differences among striatal regions are probably correct, so there is no need to discuss them as alternatives.

The authors should not refer to bar graphs as "histograms" (e.g. in supplementary figure 1 caption). The term histogram refers specifically to a graphical representation of the probability distribution of a continuous variable, usually presented as the frequency of occurrence of events with different numerical values.

Typo: Page 6, Line 116: "PV ad SOM" should be "PV and SOM"

Typo: Page 16, Line 351: "highe" should be "higher"

Reviewer #2 (Remarks to the Author):

The paper by Fino et al describes topographic differences in the inhibition of SPNs by somatostatin (SOM) vs. parvalbumin (PV) expressing neurons. The authors show that in DLS PV input is more dominant than SOM and the opposite is shown for DMS, where SOM input dominates. The authors also show that both types of inhibition have a dual effect on SPNs depending on the SPN membrane potential. The first figure shows the effects of PV/SOM inhibition in DLS and DMS on the firing of neurons in the SNR in-vivo. The following experiments are done in slices from SOM-ChR2/Arch3, PV-ChR2/Arch3 mice, showing regional differences in the synaptic and intrinsic properties of the respective types of interneurons. The paper is interesting and the data is of high quality. However, it has some limitations, as specified below.

Major comments:

1. The animals used in this study are SOM and PV ChR2 reporter lines (and not cre-dependent virus injection), which means that SOM/PV cells express ChR2 or Arch3 throughout the brain. This includes populations of these neurons in the cortex, midbrain, GPe, and others, many of which project to striatum. The consequence is that striatal axons and terminals of PV and SOM cells do not only

originate from striatal interneurons but are actually a mixed bag of cell populations. For example, two recent papers showed corticostriatal projections of SOM and PV cells (Rock et al *Elife* 2016, Melzer et al, *Cell Reports* 2017) to various striatal locations. PV cells in the GP project to striatum as well as to SNR (Saunders et al, *Plos One* 2016, Mastro et al, *Nature neurosci.* 2017), which is very likely to affect both structures upon optogenetic stimulation/inhibition. Optogenetic stimulation could affect non-striatal cells, but more importantly, it affects their striatal axons and terminals. The activation of non-striatal PV/SOM terminals has a bearing on the interpretation of both in-vivo and ex-vivo results. In order to exclude, or at least minimize these effects, experiments should be done with targeted injections of Cre-dependent (anterograde targeting) viruses in DLS/DMS in SOM or PC Cre mice.

2. An additional consideration pertains to the optogenetic inhibition using Arch, which was shown to induce release in terminals due to pH changes (Mahn et al, *Nature neurosci.*). This is the opposite than its inhibitory effect on cell bodies, and could evoke release of striatal terminals originating from local and afferent axons. This factor should be also taken into account when interpreting the results of optogenetic inhibition (both in-vivo and ex-vivo). Specifically, the authors should characterize and report the responses in SPNs to photo-inhibition in the Arch transgenic mice.

3. The response of SNR neurons to the inhibition of (putative) striatal PV cells in DLS is a slight elevation of the average firing rate. What is the authors' explanation for such an elevation? At least according to the standard model of basal ganglia organization, PV inhibition would lead to increased firing of D1-SPNs, which would, if anything reduce the spiking of SNR units. The authors remain a bit vague in describing an overall larger "change" in SNr activity when inhibiting PV in DLS, however, the fact that there is an increase in firing rate should provoke some questions regarding what is the neuronal population that is affected by the light, and how it is affected. Also, despite the fact that the authors report the strongest recruitment of SOM cells by cortex, as well as higher inhibitory impact in DMS, nothing is changed in vivo in the SNR firing properties.

4. The depolarizing effect of GABA inputs has been reported in numerous papers, including some done in the perforated patch configuration. It is clear that when SPNs are recorded ex-vivo, resting between -80 and -90mV, GABA_A will have a depolarizing effect, although the "real" reversal of GABA_A in SPNs has a wide range of reported values, from near threshold (~-45 mV) to lower than -70 mV. In addition, SPNs lie at such hyperpolarized membrane potentials under ex-vivo conditions, however in-vivo recorded SPNs display more depolarized and variable membrane potentials ranging between -75 to -55 mV (See examples Sippy et al 2015). It is not entirely clear what is new in the presented results, other than that stronger inhibition will result in stronger hyperpolarization above reversal and stronger depolarization below reversal. If the "depolarization" by GABA inputs was significant, wouldn't a ChR2 "pre pulse" before cortical stimulation enhance the SPN spiking probability in experiments presented in figure 2?

5. The effects depicted in figure 6 are very small (although statistically significant) and could arise from different reasons rather than the depolarization by GABA_A. In addition to the hyperpolarization and depolarization, convergent GABA input also affects the membrane conductance of recipient SPNs. This would predict opposite results in terms of the decay time constant (more inhibitory conductance shortening membrane time constant). What mechanism is proposed for the small changes observed only in the decay time-constant and not any other response properties? The authors should report changes in membrane conductance and time constant in SPNs in different conditions, especially in view of the potential release induced by Arch inhibition, which could actually result in increased GABA input rather than decrease. Moreover, the complete lack of effect of DMS-PV or DLS-SOM photo-inhibition indicates no connectivity whatsoever, which is not in line with the previous data.

6. Previous data suggested differences in the inhibition of dSPNs and iSPNs by PV interneurons (Gittis

et al, 2010 and 2011). It would be important to provide this information for the DLS and DMS as well as for selectivity of SOM inputs to the 2 SPN pathways. Was there any attempt to uncover the type of SPN following recordings?

7. The results regarding the connectivity and relative synaptic amplitudes in SOM and PV cells do not agree with several other papers such as Gittis (2010), showing that excitatory input to LTS cells is much smaller than that to PVs, and that FS to SPN connectivity is much stronger and prevalent than LTS-SPN connectivity. Similar results using optogenetic stimulation in PV/SOM Cre mice are shown in Straub et al (2016) showing stronger responses in SPNs following PV than SOM optogenetic stimulation. The authors should address these discrepancies.

8. The use of "specific" circuits engaged in DLS vs DMS is slightly misleading as there seem to be somewhat of a gradient but no evidence for specific circuitry devoted for the respective striatal regions.

Minor comments:

1) Distribution and activity of PV cells has already been shown to follow a dorsolateral to ventro-medial gradient (Berke, 2004). This paper should be referred to in the context of regional differences in the interneuron populations.

2) Is there any reason for using 635 nm in-vivo vs. 570 nm in ex-vivo optogenetic stimulation?

3) The assumption of stimulating only 3-4 cells (page 7) per slice is not justified and was not rigorously tested by recording a responding interneuron and shifting the objective to measure the "envelope" surrounding the cell body which would still evoke spiking. Moreover, this may also be used to test whether DMS interneurons might be activated by DLS optogenetic activation/inhibition, and vice versa.

4) The recruitment curves in figure 2 have no error bars, and it is not clear how they were obtained using "a.u." in different slices. How was it established that SOM cells are recruited with lower stimulation than PV cells if they were not recorded under the same slices and electrode positioning and using arbitrary units? What does this graph actually tell us about the relative recruitment order if they were not consistently recorded simultaneously (PV,SOM,SPN)?

5) In addition to the cortical input there is strong innervation from thalamic nuclei. The authors should address this pathway as well, since it was shown to: a) be a strong input to the striatal circuit, with comparable numbers of afferent terminals b) have different target preferences in terms of striatal neuron targets (Ding et al, Neuron 2010, and Parker et al, Neuron 2016).

6) Another result of using the ChR2/Arch3 reporter as opposed to viral expression is that using the SOM-Cre with reporter will also infect some SPNs (Straub et al Neuron 2016).

7) Optogenetic stimulation of SOM and PV expressing ChR2 would cause different number of spikes, even in relatively short (5 ms) and definitely in longer (several hundreds of ms) in the respective cell types. How was the output of the interneurons controlled in the activation experiments? Were differences in the responses of SOM and PV cells to the same stimulation observed? How many spikes were evoked by a 5 ms light pulse?

8) Numerous typos and grammatical errors throughout the text. Please check thoroughly. A few examples:

- p2 (Abstract): "...in spiking SPNs, GABAergic interneurons potently inhibit spiking SPNs (feed-forward inhibition) while in resting SPNs, they favoring cortical activity..."

- p5: "...Striatum acts as coincidence detectors..."

- p6: "...To investigate the cause of this differential effects,..."

- p7 (143-146): decreased instead of increased? "...PV cells opto-activation increased the spiking probability of SPNs in DLS ($p=0.0059$, $n=9$) but not in DMS ($p=0.9816$, $n=9$) while SOM opto-activation increased spiking probability in DMS ($p=0.0002$, $n=7$) but not in DLS ($p=0.9114$, $n=6$) (Fig.

2h-i)..."

- p10: references the wrong figure (3 instead of 4).
- p12: "...GABA current show a strong depolarizing effect,..."
- p16: "...but high amplitude of evoked-responses..."
- p17: "...Indeed, NPY expressing interneuron (also expressing somatostatin) also strongly inhibits SPNs..."
- p17: "...In addition, it exists also a feedback inhibition..."
- p17: "...recent evidences show that GABAergic..."

Reviewer #3 (Remarks to the Author):

This is a very interesting manuscript by Fino et al studying subregion specific properties of GABA interneurons in the striatum. The authors provide evidence that PV interneurons are more effective in controlling MSN activity in the DLS (due to their higher density) whereas SOM interneurons are more effective in the DMS (possibly due to their higher excitability in the DMS). Both neurons subtypes provide feed forward inhibition for above threshold cortical input but have depolarizing effects at resting state thereby positively affecting cortical integration/summation.

This is an extensive and carefully performed slice physiology study. The authors add in vivo significance to their findings by measuring basal ganglia output (SNr) activity after inhibition of PV and SOM interneurons anesthetized animals.

The manuscript is well written and of interest for a broad audience. I have some technical points that would need to be addressed:

Figure 1: I couldn't find the number of animals in the experiments for figure 1. This is important to know because bias could have been induced by one or two animals. Also, although I believe that the recordings have been performed in anesthetized animals this hasn't been explicitly stated in the result section.

Figure 1: It seems that the location of the recording probe is the same for DMS and DLS stimulations. Therefore differences in the response to interneuron inhibition when DMS and DLS are compared could be due to differential connectivity of to the specific recording site. For example DMS interneuron inhibition may just have missed the target. This needs to be clarified.

Figure 2: It looks like that PV::Arch3 inhibition has a similar effect size though with higher variability in DMS compared with DLS. A higher n would be needed for the PV-DMS condition to clarify this.

Page 7 line 143 the authors write: "PV cells opto-activation increased the spike probability"; to my understanding this should read: "decreased the spiking probability"

Page 8: line 171 Fig. 2e should be Fig. 3c

Figure 3: The authors should provide some quantitative data to confirm that the optogenetic stimulation employed leads to a comparable activation of SOM and PV interneurons in DLS and DMS (the four conditions). For example, a 5 ms optical stimulus with stimulation intensity X induces 1 AP in Y% of neurons; 300 ms optical stimulation with stimulation intensity X induces Y+-Z APs in N % of neurons.

Figure 5C is described but not explicitly mentioned in the result section.

Answers to the Reviewers

Reviewer #1 (Remarks to the Author):

The manuscript describes electrophysiological and immunohistochemistry experiments which characterize the action of two, PV and SOM-expressing, striatal GABAergic interneuron actions on the activity of spiny projection neurons of the dorsomedial (DMS) and dorsolateral (DLS) striatum. Although both interneurons have effects on SPN activity in both striatal subregions, the data shows a more pronounced effect of PV interneuron activity on DLS SPN neurons and SOM interneurons on DMS SPN neurons. Differential depolarization by these GABAergic inputs is also shown. Altogether these results provide new evidence of a role of GABA feedforward inhibition performed by two types of interneurons in controlling cortical synaptic integration by SPNs. These findings are intriguing and the authors are to be commended for taking on a project of this scope. Many of the experiments are rigorous and the data analysis is appropriate for the most part. Unfortunately, there are serious problems with several aspects of experimental design, some incomplete analyses, and a failure to consider new findings that complicate the interpretation of key experiments.

Major Concerns:

1) The authors mentioned in their discussion that there are other local GABAergic interneurons that could also alter SPN activity, but they failed to consider the impact of GABAergic afferents that innervate striatum on the interpretation of their findings. One example is the external part of the Globus Pallidus which sends a strong GABAergic input to SPNs and also to interneurons in the Striatum (see Glajch et al, 2016 and Corbit et al, 2016). Although the traditional arky pallidal neurons (pallido-striatal projections) do not express PV, some prototypical GPe neurons, which express PV, also project back to striatum and innervate SPNs, PV interneurons and NPY interneurons (Saunders et al, 2016). Another example is the GABAergic cortical projections to striatum. Cortical SOM neurons send projections to striatum (Rock et al., 2016) and this was further characterized in a very recent paper showing how SOM and PV cortical neurons innervate different cell types in the striatum (Melzer et al., published on May 2, 2017 in Cell Rep., although it is understandable that the authors were not aware of this manuscript at the time they submitted this paper). The presence of these afferents strongly confounds the experiments, because the authors used a breeding scheme between homozygous Cre lines and the rhodopsin mouse lines. ChR2 or Arch3 will be expressed in the terminals of the pallidal and/or cortical inputs, and could account for some of the effects observed with optogenetic activation or inhibition in these lines. The experiments should be compared with experiments using viral injections into the striatum of Cre-expressing adult mice to confirm that the opto-evoked results are due selectively to the striatum interneurons and not to other GABAergic projections arriving in the striatum.

We do understand the concerns of Reviewer#1 since the expression of the opsins in the transgenic mice we used in the present study is indeed not restricted to the striatum.

Recently, several studies showed evidences of GABAergic projections from the GPe and the cortex to the striatum. Concerning the GPe, Saunders et al., 2016 show that a small proportion (17%) of GPe neurons projecting to the striatum are PV+. In addition they contact only interneurons and not SPNs, the Npas+ cells from the GP would be the ones contacting directly SPNs (Glajch et al., 2016).

Recent papers also showed long-range GABAergic connections from the cortex to the striatum (Rock et al., 2016; Melzer et al., 2017, which was published during the submission process, as mentioned by the Reviewer#1).

We do agree with Reviewer#1 on the need to demonstrate and give evidences that projections from GP and cortex are not confounding in the observations we made in the manuscript. We thus performed additional experiments with local viral injections in the striatum as suggested by Reviewer#1. We reproduced the key experiments from the manuscript using viral injections of

AAV5.CBA.Flex.ArchT-tdtomato.WPRE.SV40 and AAV9.Ef1a.DIO.dfloX.hChR2(H134R)-mCherry.WPRE.hGH (UPenn Core) in SOM-IRES-Cre and PV-Cre mice.

To go into the details of the additional experiments presented here, we first showed that the viral injections were specifically targeting either DMS or DLS and that the expression of the virally delivered opsins was specific to either PV or SOM cells (new Supplementary Fig. 4). In addition, we observed that the proportions of infected striatal interneurons were similar in transgenic and virally transfected animals (Supplementary Fig. 1 and 4 for transgenic and virally-transfected animals respectively).

Using the injected animals we reproduced key experiments:

1/ the effect of opto-inhibition and opto-activation of interneurons on spiking probability was different depending on the interneurons and the striatal territory, with a strong effect of SOM cells in DMS and PV cells in DLS (new Supplementary Fig. 5).

2/ the local inhibitory weight on the frequency modulation of SPNs is different in DMS and DLS: SOM cells have a stronger weight in DMS and PV cells in DMS (new Supplementary Fig. 7)

3/ we observed this differential effect on the depolarizing effect of interneurons on the summation of cortically-evoked subthreshold events in SPNs (new Supplementary Fig. 11).

The results indicate that the effect we observed with the transgenic mice were due to local striatal interneurons since we observed similar effect with viral expression of the opsins limited to striatal PV and SOM cells.

All these new results obtained with viral infections are now presented in the manuscript and illustrated in the Supplementary Fig. 4, 5, 7 and 11 and described in the corresponding part of the Results sections and in the Supplementary Information.

2) There are problems with the *in vivo* experiments. First, performance of the experiments under anesthesia (which is not mentioned in the results and should be) makes it difficult to interpret how these circuits would work if examined during behavior. SPNs show little activity in anesthetized animals, and thus the impact of disinhibiting them in this state may be very different from what would be seen in the awake animal. This may account for the very small changes in SNr firing. The authors might consider using awake mice or providing cortical stimulation to mimic synaptic activation in this preparation. Recording from SPNs might also provide information on the actual disinhibition within striatum. The authors claim that the *in vivo* experiments indicate that distinct interneuron populations in different subregions of the striatum exert differential effects on the activity of SNr units. This is not what the first figure shows. Only the PV neurons had a significant impact on SNr firing if they were inhibited in DLS. SOM neurons do not have a differential effect on SNr activity when inhibited in the DMS or DLS. It is intriguing that although the authors found pronounced segregation of the effect of PV and SOM between DLS and DMS in slices, this was not recapitulated in the recordings in SNr. Maybe the SOM difference could be observed in other striatal output targets such as GPe or STN. It will be very helpful to add experiment to understand this issue.

According to the Reviewer#1's suggestion, we have now indicated in the Results that the experiments were performed under urethane anesthesia.

We agree with Reviewer#1 that in our original set of experiments, the subtle effect of striatal interneuron opto-inhibition (except for PV interneurons in DLS, and though a subset of SNr units were significantly modulated in all conditions) might be explained by several reasons, including (1) the low spontaneous activity of SPNs in this anesthetized preparation, (2) a probably lower spontaneous activity of SOM striatal interneurons compared to PV striatal interneurons (typically referred to as "fast-spiking") and (3) maybe a sub-optimal opto-inhibition: too low light intensity (being slightly off-peak at 635nm, because the laser diodes used in *in vivo* experiments do not exist at 570 nm) and/or too long protocol, with the opsin potentially less efficient in the later part of the protocol.

To tackle these potential issues, we performed a new series of *in vivo* experiments (under urethane anesthesia). We would like to point out that we performed our *in vivo* experiments in transgenic

animals since we had a confirmation with our set of virally injected animals that the results were the same in the two models. The new sets of experiments were the following:

- We used similar illumination protocols as in slices (300 ms opto-inhibition, stim at 100 ms after the light onset)
- We used increased light power (10 mW instead of 5 mW)
- We used cortical stimulations, as suggested by Reviewer#1, in the same cortical areas as used in slices (S2 for DLS experiments, CG2 for DMS experiments) to study the modulation of cortically-evoked responses in SNr units.

In these protocols, we also recorded the effect of light-only stimulations, and pooled the new data with the neurons recorded in the previous set of experiments (analyzing only the first 300ms for the previous neurons, and only on trials done with a light intensity of 10 mW).

These protocol modifications allowed us to observe several new results:

1/ Concerning experiments on SNr spontaneous activity, in light-only trials: in these conditions (shorter duration, stronger light) we observed an overall increase of activity of the SNr units in the 4 conditions (PV/SOM x DLS/DMS). Consistently with our results with the previous protocol (1s / 5 mW), the PV-DLS condition still showed the strongest effect (significantly stronger than both SOM-DLS and PV-DMS) both in the proportion of individually modulated units, as well as in the magnitude of the population effect. In addition, we observed a differential effect between PV-DMS and PV-DLS with a higher magnitude from PV-DLS. These new results are now illustrated in the Figure 1 and Supplementary Fig.2.

2/ To investigate the role of striatal interneurons on SNr activity without the bias induced by their likely different spontaneous activity, as suggested by Reviewer#1, we explored the effect on cortically-evoked activity in SNr. We compared the response of SNr units to cortical stimulations with and without the opto-inhibition of PV or SOM interneurons in DMS or DLS. We observed different effects depending on the territory and interneuron type: inhibiting SOM interneurons in DMS specifically decreased the inhibitory cortico-nigral response, and inhibiting PV-DLS specifically decreased the late excitatory cortico-nigral response. More specifically, for these experiments, we stimulated cortical areas projecting either to DMS (CG2) or DLS (S2), similarly as our *ex vivo* experiments. We observed typical polyphasic responses in the SNr upon cortical stimulation in patterns and proportions similar to previous studies in rats and mice (Fujimoto et al., 1992; Kolomiets et al., 2003; Sano et al., 2013)), displaying a combination of 3 phases among which an inhibition thought to reflect the activation of the trans-striatal direct pathway, and a late excitation corresponding to the activation of the trans-striatal indirect pathway. We observed that striatal interneurons had specific effects on the “triphasic” response, depending on the territory: opto-inhibition of SOM-DMS led to a decrease of the inhibition phase, and opto-inhibition of PV-DLS led to a decrease of the late excitation phase, while other combinations had no effect on any of the parameters tested. These new experiments are now included in the Results Section and illustrated in Figure 7 and Supplementary Fig. 12.

These *in vivo* results strongly support a differential role of SOM and PV cells in DMS and DLS since they have selective effect on the cortico-striato-nigral information transfer. This data is now included in the Results part and is illustrated in the Fig.1 and Fig.7 and in Supplementary Fig. 2 and 12.

3) The electrophysiological experiments in supplementary figures 3 and 4 are helpful, but incomplete. The paired recordings were only performed for the PV neurons (the authors refer to an experiment in SOM neurons on page 6, but it is not in the figure), and thus the estimate of how many SOM neurons were activated may not be accurate. The authors do not indicate in which striatal subregion the recordings in supplemental figures 3C, 3D and 4C were performed.

We do agree with Reviewer#1 that we showed paired recordings of connected PV-SPN but no SOM-SPN. To complete this set of experiments, we performed paired recordings of SOM-SPN to measure unitary synaptic weight and thus estimate the numbers of recruited SOM cells with optogenetics. The results describing the IPSPs recorded in SPNs following paired recordings or light-induced currents after opto-activation of PV and SOM cells both in DMS and DLS are described in the attached figure. For both PV and SOM cells we did not see any significant differences between DMS and DLS in unitary connections. As previously described in Figure 5, we observe significant differences in the light-induced currents in DMS and DLS, currents are stronger for SOM cells in DMS and for PV cells in DLS. This is explained by the highest density of PV cells in DLS and the highest excitability of SOM cells in DMS. In the new Supplementary Fig. 6 we included the comparison between pairs and light-induced currents for either PV cells or SOM cells (“all” condition) which pools data from DMS and DLS. We pooled the data because there is no significant difference between unitary connections in DMS and DLS for both PV and SOM cells. Our aim was to show globally how many synaptic interneurons were at play in the effects we were observing on SPNs when doing wide field opto-activations. We detailed in the present Figure the results for each interneuron and each territory to show Reviewer#1 all the comparisons of all the conditions.

We indeed did not indicate where the recordings were from in supplementary Fig. 3c and 4c. We performed control recordings of PV::ChR2-YFP and SOM::ChR2-YFP cells in either DMS or DLS and made the same observation for all the conditions: 100% of ChR2-YFP-expressing cells were spiking in response to the light stimulation. We illustrated only one representative cell for each population.

We clarified this information in the legend of the new Supplementary Fig. 6. “In all of the PV cells recorded (n=8, one representative shown), we were able to evoke APs with each 5ms light stimulation with 100% success rate, at all tested frequencies and both in DMS or DLS” and “In all of the SOM cells recorded (n=7, one representative shown), we were able to evoke APs with 5ms light stimulation with 100% success rate, at all tested frequencies and both in DMS or DLS.”

4) The cell counting in the different regions should be performed using unbiased stereological techniques, and it does not appear that this was the case. Also, regarding the data in supplementary figure 1, is there any overlap between the SOM and PV staining or between the GFP markers for these two neuronal subtypes (presumably not)?

We do agree that the classical technique used to count the cells in 3D would be stereology. We did not have a system available and thought that for our purpose stereology was not absolutely necessary for 2 main reasons: (1) our aim was not to have the absolute number of interneurons in the whole DMS and DLS but to compare the numbers of cells between the two territories and calculate a ratio, (2) both PV and SOM interneurons are really sparse in the striatum (as illustrated by the immunostainings) and given the small size of the Z-stack (30-40µm with 1µm step size), counting them in 3D is not absolutely necessary. We therefore performed Z-stack maximum projections and did the analysis on

these images in Fiji by placing similar size ROI in either DMS or DLS and reporting the number of the counted cells to the full volume of the Z-stack. Finally, we do agree that the counting should be unbiased so we blindly performed all the counting of the immunostainings in the different conditions.

It is true that we did not detail enough the analysis in the previous version of the manuscript so we have now included a detailed explanation in the Material and Methods section: “Images were acquired using an SP5 confocal microscope (Leica, Germany). Small (30-40 μ m) Z-stacks with 1 μ m step size were acquired in DMS and DLS with a 20X objective (0.8 NA). For each animal, we acquired Z-stacks on 3 slices per territory (DMS and DLS) and in both hemispheres, so 6 Z-stacks per condition. We used Fiji software for analysis.

Z-stack maximum projections were applied quantification of neurons was performed in regions

of interest (ROIs) of identical size in DMS and DLS. The number of cells was reported to the volume corresponding to the ROIs. The sparseness of the striatal interneurons allowed us to easily count the number of cells in a given volume and to have unbiased analysis, the counting was performed by experimenters that were blind to the experimental conditions.”

Finally, we did not perform co-immunostainings of PV and SOM for every experiment since we were focusing on each subtype in the corresponding mouse lines. Nevertheless, we performed some co-immunostainings and, as shown on the representative example on the figure, there is no overlap between SOM and PV stainings.

5) Figure 5e, it may difficult to estimate the depression of IPSCs if the amplitudes are very small (e.g. as in the PV-DMS data). In particular, the amplitude of the later IPSCs may be hard to measure given the signal/noise ratio and the elevated baseline due to previous IPSCs.

It is indeed true that the example given in Fig. 5e for PV-DMS was not nicely representing the recordings we made. In average, IPSP amplitude was 1.86 ± 0.11 mV and the signal to noise allowed us to measure small amplitude IPSPs as well as high amplitude ones. Thus our measures allowed us to reliably study short-term dynamics of light-evoked currents in SPNs. Accordingly, we changed the representative trace in the new version of the Fig.5e to better illustrate our recordings.

6) Why did the authors use homozygous Cre mice for the breeding schemes? There are some reasons homozygous Cre should be avoided: Cre alone can produce phenotypes by affecting cell physiology or Cre insertion could dysregulate some specific genes which may change the cell physiology by itself. Cre could be express in germline cells and in homozygous breeding schemes there is a bigger change to cause germline recombination (may happen in PV Cre males because PV is expressed in the sperm).

For mouse breeding, we followed the husbandry scheme described by Jackson Laboratory (homo x homo). Our breedings PV-Cre :: Arch3-YFP and PV-Cre :: ChR2-YFP were made between females PV-Cre and males Arch3-YFP and ChR2-YFP to avoid germline recombination. In addition, these breedings gave us hybrids that were all heterozygous animals and only these animals were used for the experiments. Lastly, since we are concerned by limiting the number of animals used in each project, this breeding scheme allowed us to use all the animals whereas heterozygous breedings, on the contrary, would leave us with 25% of wild-type animals that would be useless for our study.

7) The PV Cre and SOM Cre mice used in the study originated in different strains of mice (PV in

129P2/OlaHsd and SOM in C57BL/6 x 129S4/SvJae). Were the animals backcrossed sufficiently onto a homogeneous background (e.g. C7Bl6J) to allow direct comparison between those two lines?

The PV-cre mice were generated first in 129P2/OlaHsd embryonic stem cells and SST-Cre mice in C57BL/6 x 129 cells. In both cases, the resulting chimeric animals were crossed to C57Bl6 mice for minimum 2 generations at JAX laboratory. In addition, for our experiments with transgenic mouse lines, we mated both PV-Cre and SST-Cre with the two mouse lines Arch3-YFP and Chr2-YFP. Both lines expressing opsins have a mixed genetic background C57BL/6 x 129 and are backcrossed at least 3 times at JAX on C57BL/6 mice.

This would mean that both lines have a C57Bl6 background because of the 6 backcross leading to our hybrids. The background is slightly different in the mix with 129 background with 5/16 of 129 background for PV-ChR2 and 17/64 129 background for SOM-ChR2. We do think that this background difference is minor and, more importantly, we believe that it could not be responsible for our results for different reasons: (1) we observed differences between both territories in the exact same animal (*in vivo* experiments presented in Figure1), (2) the heterogeneous distribution of PV cells we observed is coherent with previous reports made in wild-type animals, rats and humans (Gerfen, 1995, Bernacer, 2012) and it is therefore very unlikely that small background differences would explain differences in synaptic weight within the same brain territory and (3) we used a breeding scheme that is classically used by many studies using Cre-mouse lines.

Altogether, we think that it is reasonable to compare our mouse lines.

8) Page 10, 1st paragraph, was the rheobase always calculated from the resting potential or were membrane potentials adjusted with constant current injection to set the potential at a particular resting value?

The rheobase was always calculated from the resting membrane potential (RMP). We thought it was important to keep the neurons at their physiological RMP to detect any difference in electrophysiological properties.

9) Bottom of page 8, the authors discuss the correlations shown in figure 3d and note a negative correlation between SPN firing rate and the effect of SOM inhibition on this rate. However, the data are plotted in such a way that the correlation appears to be positive. This appears to be due to the way the delta in firing was calculated/plotted. The authors should also provide the r or r2 values for all correlations. Otherwise it is difficult to determine how much of the variability is accounted for by the correlation of the two variables.

We do agree with Reviewer#1 that this was not clear in the previous version. The effect of SOM cells on SPN firing rate in negatively correlated because the frequency ratio (LED on / LED off) increases with frequencies, meaning that SOM cells are more efficient to modulate lower frequencies than higher ones.

In addition, we have now added the r2 values in the Fig.3 legends and indicated on the graphic itself that only SOM correlation was significant: "Correlation of PV (green) and SOM (purple) cells inhibitory weight (discharge frequency ratio) and initial SPN discharge frequency. There is a significant correlation for SOM cells, in both DMS ($r^2=0.3481$, Pearson correlation, $p=0.0437$, $n=14$) and DLS ($r^2=0.4648$, $p=0.0127$, $n=14$), and there was no significant correlation for PV cells ($r^2=0.097$ for DMS and $r^2=0.1739$ for DLS)."

10) The age of mice used for the *in vivo* and *ex vivo* experiments is very different. Is there any change in the pattern of PV or SOM promoter expression between those ages that could affect the results?

To answer this question, we performed immunostainings in adult mice to label PV and SOM cells and quantified the expression of both interneurons in adult (~6 month-old) animals. We observed that the pattern of expression of both PV and SOM cells is similar in old than in younger animals. These results are now included in the new Supplementary Fig. 8.

It should be noted that our results are in accordance with previous studies showing the different expression pattern of PV interneurons (Gerfen, 1995 and Berke et al., 2004 in rodents and Bernacer et al., 2012 in humans) and which were performed in adult animals.

11) SNr units were identified as GABAergic neurons by having a firing rate of at least 10Hz or higher, but the authors showed an median \pm interquartile range of 17 ± 10 Hz (page 5 line 93) which may indicate that there were a few neurons with less than 10Hz. Authors should clarify this.

There was no error in this notation, and no unit selected for analysis was firing below 10Hz. The source of the confusion however is our use of “ \pm ”, which is more appropriate for symmetrical intervals such as standard deviations. As the reviewer is certainly aware of, the interquartile range is not necessarily symmetrical around the median, especially if the variable does not follow a Gaussian distribution. This is indeed the case here with the firing rates (if anything, non-gaussian precisely because of the cutoff at 10 Hz), hence our use of median(IQR) descriptive statistics rather than mean \pm SD. To clarify, in this particular case, the full range was 10.18 – 53.8 Hz, with the 25% quartile at 13.33 Hz and the 75% quartile at 23.37 Hz. The IQR was therefore indeed 10.04Hz, with the median at 17 Hz, and despite the absence of any unit firing below 10 Hz. (Note that these figures are now updated with the new subset of units added, now but the same reasoning applies: median(IQR) = 18.7(10.3) Hz, n= 239 units; full range = 10.2-59.1Hz, 25%/75% quantiles 14.8/25.1Hz).

To avoid the confusion, we therefore changed all the “median \pm IQR” notations to the more appropriate “median(IQR)”.

12) It's unclear how the Confocal Z-Stack was performed. What was the optical section used and how many stacks were reconstructed for the analyzed images?

We acquired confocal Z-stack of 30-40 μ m with 1 μ m step size with a 20X objective (0.8 NA). The resolution, given by the formula $1.4 n\lambda / (NA)^2$, was coherent with the 1 μ m step size for the different wavelengths used (488nm for GFP or YFP, 568nm for Alexa568, 561nm for mCherry or tdTomato). For each animal, we performed acquisition on 3 slices per territory (DMS and DLS) and in both hemispheres, so 6 Z-stacks per condition. We then used Z-stack maximum projections to perform analysis and count the cells in Fiji by placing similar size ROI in either DMS or DLS and reporting the number of the counted cells to the full volume of the Z-stack. The sparseness of the striatal interneurons allowed us to easily count the number of cells in a given volume and this analysis was performed by experimenters that were blind to the experimental conditions.

These details have now been detailed in the Material and Methods section: “Images were acquired using an SP5 confocal microscope (Leica, Germany). Small (40 μ m) Z-stacks with 1-2 μ m step size were acquired in DMS and DLS and analysis was made on stack projections. For each animal, we performed acquisition on 3 slices per territory (DMS and DLS) and in both hemispheres, so 6 Z-stack per condition. We used Fiji software for analysis. Z-stack maximum projections were applied and Quantification of neurons was performed () in regions of interest (ROIs) of identical size in DMS and DLS. The number of cells was reported to the volume corresponding to the ROIs. The sparseness of the striatal interneurons allowed us to easily count the number of cells in a given volume and to have unbiased analysis, the counting was performed by experimenters that were blind to the experimental conditions.”

13) What statistical test was used for comparisons of data in figure 1e? Was this a Fisher's exact test or chi square? Same question for the data in figures 2e and f.

For Figure 1e, the statistical method was explained in the 'in vivo' method sub-part. Neither a Fisher's exact nor a chi-square were used, because they would have been unable to take into account the pairing of the data on the territory factor: indeed, in the original set of experiments, for each type of interneuron, SNr units were tested for modulation applied in both DMS and DLS (2 fibers in the same mouse). This leads to a mixed model where the “territory” factor is paired, while the “interneuron” factor is not, which cannot be modeled by Fisher or chi-square tests. We therefore used a general linear mixed model with a probit link function, as follows:

1/ first, each SNr unit was individually classified as significantly modulated or not, in each tested condition, using a wilcoxon signrank test. This binary result (modulated = 1; non-modulated=0) constituted the dependent variable, Fmod.

2/ Fmod was modelled by the factors "interneuron" (2 levels : SOM and PV), territory (2 levels : DMS and DLS), and their interaction, with a random effect (identity of the unit, allowing to take into account the pairing for territory levels). In short : $F_{mod} \sim \text{interneuron} * \text{territory} + (1|ID_unit)$. This was performed in R, using the glmer function of lme4 package, with a probit link function.

3/ The model was used to compare the proportion of modulated units between conditions using posthoc contrasts (lsmeans package), correcting for the false discovery rate.,

In this new version of the manuscript, the pairing in the territory factor is not respected for the additional units recorded in the stimulation paradigm (where each of the 4 conditions is tested in different sets of animals). The model is still able to take this partial pairing into account thanks to the random factor, and was updated accordingly with the new units.

The comparison of the effect on the firing rate (Fig. 1f) between the 4 conditions was similarly analyzed, using a linear mixed model (lmer function, lme4 package) to model the change in firing rate : $F_{Change} \sim \text{interneuron} * \text{territory} + (1|ID_unit)$. The dependent variable was transformed using a signed 2/3 root function to optimize the normality of the residuals.

The methods have been updated to better describe the models used, and have been moved to the general 'statistics' paragraph in the Material and Methods section for clarity.

14) As the authors are no doubt aware, depolarizing GABA responses can still inhibit neurons by current shunting and stabilizing the membrane potential at EGABA which is generally subthreshold. It is surprising that these concepts and their potential influence on the findings were not discussed.

The shunting effect of depolarizing GABA could indeed act at the opposite way of what we are describing. We have now included this idea in the Discussion section: "A dual control of GABA on SPNs is functionally important because GABAergic interneurons would modulate in opposite direction sub- and suprathreshold events. The high amplitude of GABA currents we recorded for depolarized SPNs is coherent with a strong hyperpolarizing effect^{15,36} and a depolarizing effect of GABA has also been previously described in striatum^{55,56}. Of course, we cannot rule out that the depolarizing GABA could have a shunting effect and in that case decrease the efficiency of cortical inputs to activate SPNs. We describe here a functional role of the depolarizing GABA favoring the integration of cortical inputs in the subthreshold range of SPNs. This is in accordance with previous models predicting that some depolarizing inputs coming when SPNs are at rest with K_{IR} activated should shift the inactivation of K_{IR} , therefore reducing their availability and promoting summation of further inputs⁵⁷. The GABAergic system would thus counterbalance intrinsic properties of SPNs by favoring depolarization in SPNs at rest and slowing them down when they reach the AP threshold."

15) The cellular location of synapses made by PV and SOM interneurons may influence their effectiveness in SPN depolarization and inhibition. PV neurons generally synapse near the soma, but is this also the case for SOM neurons? This issue should be discussed.

The laboratory of Bolam showed that anatomical connections of PV+ and NO+ (also SOM+) interneurons onto SPNs were both on the aspiny part of the SPN dendritic arborizations, the NO+ cells being more distal than PV+ cells which contact mainly the perisomatic region (Smith and Bolam, 1990). Recently, Sabatini's laboratory reported that PV cells efficiently contact the SPN soma while SOM cells would contact more distal part of the dendritic arborization (Straub et al., 2016). Nevertheless, they showed that for both cells, the contacts are made within the first 250µm around the soma. The cortical synapses on SPNs are formed on the spiny part of the dendrites further than 300µm. Therefore, contacts from both PV and SOM cells are differentially located but not too far away

and they are both downstream the cortical synapses and will be in position to modulate the integration of cortical inputs.

We have now added this point in the Discussion part: “Finally, the localization of the synapses from PV and SOM cells on the SPN dendrites could rise specific modulation of cortical inputs. Nevertheless, even though PV cells contact SPN closer to the soma than SOM cells, their contacts are both located within the first 250µm of the SPN dendritic arborization³⁶, placing both of them in the strategic position to modulate the integration of glutamatergic inputs.”

Minor points:

For the *in vivo* data, authors should not refer to the data as neurons but as units.

We are now referring to them as units for *in vivo* data throughout the manuscript.

The lot numbers for the antibodies should be mentioned.

We have now indicated the lot numbers of the primary antibodies in the Material and Methods section.

“Immunostaining was performed on free-floating sections from right and left hemispheres using chicken anti-GFP (1:1000; AbCam AB13970, lot #GR89472-7/12/16), rat anti-SOM (1:200, Millipore MAB354, lot #2326265, 2739496, 2933843) and rabbit anti-PV (1:1000, Swant Technologies P25, lot #510, 2014), and the appropriate fluorescent-labeled secondary antibodies (Alexa 568 for PV and SOM and Alexa 488 for GFP).”

Page 3, 1st paragraph, the authors should also cite the newer findings from connectome studies using viral tracers to map corticostriatal inputs in more detail.

We have now added two recent studies describing corticostriatal connectome (Hintiryan et al., 2016, Nat Neuro; Hunnicutt et al., 2016, eLife) in this paragraph.

Page 3, 2nd paragraph, both hypotheses about the factors that account for functional differences among striatal regions are probably correct, so there is no need to discuss them as alternatives.

We agree with Reviewer#1 and we modified the text to present them as complementary hypotheses.

The authors should not refer to bar graphs as “histograms” (e.g. in supplementary figure 1 caption). The term histogram refers specifically to a graphical representation of the probability distribution of a continuous variable, usually presented as the frequency of occurrence of events with different numerical values.

We have modified the term “histograms” for “bar graphs”.

Typo: Page 6, Line 116: “PV ad SOM” should be “PV and SOM”

Typo: Page 16, Line 351: “highe” should be “higher”

We have now corrected these typos.

Reviewer #2 (Remarks to the Author):

The paper by Fino et al describes topographic differences in the inhibition of SPNs by somatostatin (SOM) vs. parvalbumin (PV) expressing neurons. The authors show that in DLS PV input is more

dominant than SOM and the opposite is shown for DMS, where SOM input dominates. The authors also show that both types of inhibition have a dual effect on SPNs depending on the SPN membrane potential. The first figure shows the effects of PV/SOM inhibition in DLS and DMS on the firing of neurons in the SNR in-vivo. The following experiments are done in slices from SOM-ChR2/Arch3, PV-ChR2/Arch3 mice, showing regional differences in the synaptic and intrinsic properties of the respective types of interneurons. The paper is interesting and the data is of high quality. However, it has some limitations, as specified below.

Major comments:

1. The animals used in this study are SOM and PV ChR2 reporter lines (and not cre-dependent virus injection), which means that SOM/PV cells express ChR2 or Arch3 throughout the brain. This includes populations of these neurons in the cortex, midbrain, GPe, and others, many of which project to striatum. The consequence is that striatal axons and terminals of PV and SOM cells do not only originate from striatal interneurons but are actually a mixed bag of cell populations. For example, two recent papers showed corticostriatal projections of SOM and PV cells (Rock et al Elife 2016, Melzer et al, Cell Reports 2017) to various striatal locations. PV cells in the GP project to striatum as well as to SNR (Saunders et al, Plos One 2016, Mastro et al, Nature neurosci. 2017), which is very likely to affect both structures upon optogenetic stimulation/inhibition. Optogenetic stimulation could affect non-striatal cells, but more importantly, it affects their striatal axons and terminals. The activation of non-striatal PV/SOM terminals has a bearing on the interpretation of both in-vivo and ex-vivo results. In order to exclude, or at least minimize these effects, experiments should be done with targeted injections of Cre-dependent (anterograde targeting) viruses in DLS/DMS in SOM or PC Cre mice.

We do understand the concerns of Reviewer#2 since the expression of the opsins in the transgenic mice we used in the present study is indeed not restricted to the striatum.

Recently, several studies showed evidences of GABAergic projections from the GPe and the cortex to the striatum. Concerning the GPe, Saunders et al., 2016 show that a small proportion (17%) of GPe neurons projecting to the striatum are PV+. In addition they contact only interneurons and not SPNs, the Npas+ cells from the GP would be the ones contacting directly SPNs (Glajch et al., 2016).

Recent papers also showed long-range GABAergic connections from the cortex to the striatum (Rock et al., 2016; Melzer et al., 2017, which was published during the submission process, as mentioned by the Reviewer#1).

We do agree with Reviewer#2 on the need to demonstrate and give evidences that projections from GP and cortex are not confounding in the observations we made in the manuscript. We thus performed additional experiments with local viral injections in the striatum as suggested by Reviewer#1. We reproduced the key experiments from the manuscript using viral injections of AAV5.CBA.Flex.ArchT-tdtomato.WPRE.SV40 and AAV9.Ef1a.DIO.dfloX.hChR2(H134R)-mCherry.WPRE.hGH (UPenn Core) in SOM-IRES-Cre and PV-Cre mice.

To go into the details of the additional experiments presented here, we first showed that the viral injections were specifically targeting either DMS or DLS and that the expression of the virally delivered opsins was specific to either PV or SOM cells (new Supplementary Fig. 4). In addition, we observed that the proportions of infected striatal interneurons were similar in transgenic and virally transfected animals (Supplementary Fig. 1 and 4 for transgenic and virally-transfected animals respectively).

Using the injected animals we reproduced key experiments:

1/ the effect of opto-inhibition and opto-activation of interneurons on spiking probability was different depending on the interneurons and the striatal territory, with a strong effect of SOM cells in DMS and PV cells in DLS (new Supplementary Fig. 5).

2/ the local inhibitory weight on the frequency modulation of SPNs is different in DMS and DLS: SOM cells have a stronger weight in DMS and PV cells in DMS (new Supplementary Fig. 7)

3/ we observed this differential effect on the depolarizing effect of interneurons on the summation of cortically-evoked subthreshold events in SPNs (new Supplementary Fig. 11).

The results indicate that the effect we observed with the transgenic mice were due to local striatal interneurons since we observed similar effect with viral expression of the opsins limited to striatal PV and SOM cells.

All these new results obtained with viral infections are now presented in the manuscript and illustrated in the Supplementary Fig. 4, 5, 7 and 11 and described in the corresponding part of the Results sections and in the Supplementary Information.

2. An additional consideration pertains to the optogenetic inhibition using Arch, which was shown to induce release in terminals due to pH changes (Mahn et al, Nature neurosci). This is the opposite than its inhibitory effect on cell bodies, and could evoke release of striatal terminals originating from local and afferent axons. This factor should be also taken into account when interpreting the results of optogenetic inhibition (both in-vivo and ex-vivo). Specifically, the authors should characterize and report the responses in SPNs to photo-inhibition in the Arch transgenic mice.

In their paper, Mahn et al. indeed described an opposite effect of Arch activation but this undesirable effect appears for very long light stimulations since they apply several minutes of light ON. Their results using shorter pulses (200 ms, figure 2 of the article) show that illumination of axonal terminals expressing Arch3 does slightly attenuate synaptic transmission but does not increase it. In our conditions, the light pulses were maximum 1 sec long in the initial version (now a maximum of 300 ms is analyzed) so we do believe that with short pulses there would be no artefactual release.

To control this more directly, we agree with Reviewer#2 that we needed to characterize the effect of light pulses on SPNs. We thus performed such analysis and observed that the LED has no effect on the membrane potentials of SPNs, excluding the possibility of Arch-triggered synaptic release in our conditions. We have now included a quantification of the effect of Arch activation in SPNs in the new Supplementary Fig. 3.

3. The response of SNR neurons to the inhibition of (putative) striatal PV cells in DLS is a slight elevation of the average firing rate. What is the authors' explanation for such an elevation? At least according to the standard model of basal ganglia organization, PV inhibition would lead to increased firing of D1-SPNs, which would, if anything reduce the spiking of SNR units. The authors remain a bit vague in describing an overall larger "change" in SNr activity when inhibiting PV in DLS, however, the fact that there is an increase in firing rate should provoke some questions regarding what is the neuronal population that is affected by the light, and how it is affected. Also, despite the fact that the authors report the strongest recruitment of SOM cells by cortex, as well as higher inhibitory impact in DMS, nothing is changed in vivo in the SNr firing properties.

Initially, we didn't emphasize the increase in activity because while the overall effect in PV-DLS was an increase in activity, we also observed units with a significantly decreased activity. Following our new experiments with increased light power and shorter light duration, done to answer Reviewer#1 and 2's comments, we have observed that the overall increase was significant in all conditions, and that nearly all individually significantly modulated neurons were activated. Therefore and according to Reviewer#2's suggestion, we have more clearly indicated that the SNr activity was increased upon opto-inhibition of PV or SOM striatal interneurons in DMS and DLS (though with a stronger effect in PV than in SOM, and in PV-DLS than PV-DMS) (see Fig. 1).

This result suggests that in all conditions, the net effect of opto-inhibition is a shift of the balance between the direct, inhibitory, and the indirect, disinhibitory striato-nigral pathways towards a stronger relative activity of the indirect pathway. This would suggest that opto-inhibition of striatal interneurons disinhibits more strongly indirect pathway neurons (iSPNs) than direct pathway neurons (dSPNs). PV and SOM interneurons connect dSPNs and iSPNs rather symmetrically (Planert et al., 2010, Straub et al., 2016). The marginal asymmetry described in Gittis et al., 2010 and 2011, with a preferential connection to dSPNs, would no more explain our results, since it would lead to a stronger disinhibition of the inhibitory direct pathway. Therefore, it is unlikely that our result on SNr activity is due to an asymmetrical connection of striatal interneurons on SPNs. However, iSPNs are known to be more

excitable than dSPNs (Kreitzer and Malenka, 2007, Kravitz et al., 2010, Planert et al., 2013). Hence, while the effect opto-inhibition of striatal interneurons might be similar on dSPNs and iSPNs, under spontaneous anesthetized network state it might translate into iSPNs more easily reaching spiking threshold than dSPNs, resulting in a stronger indirect pathway activity relative to the direct pathway, and thus a net activation of SNr units. However, considering our *ex vivo* results on the dual effect of interneurons (Fig. 5), we cannot exclude that the SNr activation could result from just the opposite mechanism: rather than a stronger disinhibition of iSPNs, opto-inhibition could as well result in the removal of an excitatory component on SPNs (favoring summation of EPSPs, cf Fig. 6), which, if the effect is stronger in dSPNs (for example if PV interneuron do connect preferentially dSPNs, according to Gittis et al.), would result in the same net effect on SNr activity (and of course, this effect could result from a mix of both mechanisms).

The fact that opto-inhibiting SOM interneurons leads to a smaller effect on SNr activity than opto-inhibiting PV interneurons in both DMS and DLS is not inconsistent with the strongest recruitment of SOM interneurons by cortical activation. Indeed, in our anesthetized *in vivo* experiments on spontaneous SNr activity, it is likely that PV interneurons have a stronger spontaneous activity than SOM interneurons : PV interneurons are classically described as “fast-spiking” *in vivo* and typically identified by a short waveform duration and a higher activity compared to MSNs or putative cholinergic interneurons (e.g. in Berke et al., 2004; Mallet et al., 2005); while little data is available on identified SOM interneurons, it is likely that their spontaneous activity would be weaker than “fast-spiking” interneurons. Therefore, it is consistent that the opto-inhibition of more active population (PV) has a stronger effect than the opto-inhibition of a less active population (SOM).

These ideas have been now included in the Discussion part: “Our *in vivo* recordings show an activation of SNr upon opto-inhibition of both interneurons in both territories, meaning that the net effect of SOM and PV cells on SNr spontaneous activity is inhibitory. This suggests that interneurons modulate the balance between the direct (inhibitory on SNr) and indirect (disinhibitory) pathways, towards a relative activation of the direct pathway, and/or inhibition of the indirect pathway, yet the striatal mechanism leading to this result could be multiple. Recent studies show that GABAergic interneurons contact both direct (dSPNs) and indirect pathway SPNs (iSPNs)^{37,38,40}, meaning that the local interneurons might control both SPN subtypes in a similar way. Since iSPNs are more excitable than dSPNs⁴⁷, their disinhibition could lead to a stronger increase in firing rate in iSPNs than dSPNs, thus explaining the observed overall increase of firing rate in SNr after silencing striatal interneurons. However, considering the dual depolarizing/hyperpolarizing effect of PV and SOM described here, the opposite hypothesis cannot be excluded: silencing striatal interneurons could lead to a decreased summation of spontaneous cortical inputs, which, if stronger in dSPNs, would also shift the net balance towards an activation of SNr. Other members of striatal microcircuits are also likely involved in this effect, and could contribute to either a decreased activity of the direct pathway, or an increased activity of the indirect pathway. The stronger effect of silencing PV than SOM interneurons on SNr spontaneous activity in both territories could be due to a stronger spontaneous activity of PV cells, classically described as “fast-spiking” and identified *in vivo* by their higher firing rates^{11,35}.”

To investigate the role of striatal interneurons on SNr activity without the bias induced by their likely different spontaneous activity, we compared the response of SNr units to cortical stimulations with and without the opto-inhibition of PV or SOM interneurons in DMS or DLS. We would like to point out that we performed *in vivo* experiments in transgenic animals since we had a confirmation with our set of virally injected animals that the results were the same with the two models. These new experiments are now included in the Results Section and illustrated in Figure 7 and Supplementary Fig. 12. For this, we stimulated cortical areas projecting either to DMS (CG2) or DLS (S2), similarly as our *ex vivo* experiments. We observed typical polyphasic responses in the SNr upon cortical stimulation in patterns and proportions similar to previous studies in rats and mice (Fujimoto et al., 1992; Kolomiets et al., 2003; Sano et al., 2013)), displaying a combination of 3 phases among which an inhibition thought to reflect the activation of the trans-striatal direct pathway, and a late excitation corresponding to the activation of the trans-striatal indirect pathway. We observed that striatal interneurons had specific effects on the “triphasic” response, depending on the territory: opto-inhibition of SOM-DMS led to a decrease of the inhibition phase, and opto-inhibition of PV-DLS led to a decrease of the late excitation phase, while other combinations had no effect on any of the parameters tested for the 2

trans-striatal phases. The early excitation, corresponding to the activation of the hyper-direct cortico-STN-SNr (and is not going through the striatum) was not affected in any of the 4 conditions where striatal interneurons were opto-inhibited. Though it would be tempting to conclude that SOM/PV interneurons only affect each direct/indirect pathway, respectively, we think the interpretation should be more careful. Indeed the activation of the direct and indirect pathways are overlapping, and the phases that can be measured, while mainly corresponding to each pathway, are still the sum of this overlap. For example, in the case of the inhibition, the fact that the duration (and area) are reduced when opto-inhibiting SOM interneurons in the DMS could result as well from an actual decrease in the activation of the direct pathway, as from a reduced latency of the activation of the indirect pathway. Additional explanations are related to specific properties of iSPNs and dSPNs and specificities in the cortical inputs innervating DMS and DLS. iSPNs are more excitable and are recruited faster with shorter responses than dSPNs, which maintain a longer activation in response to cortical stimulation (Flores-Barrera et al., 2010). The latency of interneurons activation is shorter for PV cells (Fig. 2), which would be likely to modulate more efficiently the first recruited iSPNs while SOM cells would control more efficiently dSPNs, activated after and longer. In addition, our observation is also coherent with different organization of cortical inputs described with rabies viruses. Inputs from cingulate cortex contact more dSPNs in DMS while iSPNs are more connected by somatosensory inputs in DLS (Wall et al., 2013). Finally, we observed that both SOM-DMS and PV-DLS opto-inhibition lead to a decrease of a striato-SNr pathway (direct pathway for SOM-DMS and indirect for PV-DLS), meaning that the net effect of striatal interneurons would be excitatory locally on SPNs. This is coherent with a recent study showing that locally within the striatum, silencing PV cells lead to a decrease in SPN activity (Lee et al., 2017). This unexpected effect is due to disinhibitory loops between PV and NPY cells (Lee et al., 2017). Other GABAergic loops have been recently described (Straub et al., 2016, Assous et al., 2017, English et al., 2011) and could explain the overall excitatory effect we see in the SNr after silencing striatal PV and SOM cells. Indeed, local disinhibitory role of interneurons has been recently highlighted in several brain structures as an important mechanism to regulate the input/output flow of information (Artinian et al., 2017).

We have now included these ideas in the Discussion part: “When stimulating cortical afferents, it is possible to visualize within the SNr different phases of response corresponding to hyper-direct, direct and indirect pathways. Similarly to local striatal control, we observed a specific modulation of cortically-evoked SNr activity from SOM-DMS and PV-DLS. Interestingly, SOM-DMS seem to modulate the inhibition resulting from the direct pathway and PV-DLS the late excitation related to indirect pathway recruitment. iSPNs are more excitable and are recruited faster with shorter responses than dSPNs, which maintain a longer activation in response to cortical stimulation⁴⁸. The latency of interneurons activation is shorter for PV cells (Fig. 2 and coherent with kinetics described in⁴⁰), which would be likely to modulate more efficiently the first recruited iSPNs while SOM cells would control more efficiently dSPNs, activated after and longer. In addition, our observation is also coherent with different organization of cortical inputs from cingulate cortex, which contact more dSPNs in DMS, and in DLS where iSPNs are more connected by somatosensory inputs⁴⁹. *In vivo* recordings also show that the interplay of inhibitory and excitatory effects of GABA is complex on the resulting effect on output structures such as SNr. Both SOM-DMS and PV-DLS opto-inhibition lead to a decrease of a striato-SNr pathway, meaning that the net effect of striatal interneurons would be excitatory locally on SPNs. A recent study also describes an overall decrease of SPN activity after silencing PV cells *in vivo* due to a disinhibitory loop from PV cells to NPY (overlapping with SOM cells) interneurons⁵⁰. Another disinhibitory mechanism has been described between TH-positive interneurons and SOM cells⁵¹. A silencing of SOM might also lead to a decreased activity of SPNs, since SOM interneurons (but not PV) inhibit cholinergic interneurons⁴⁰, that in turn inhibit SPNs via their activation of NPY-NGF interneurons⁵². Therefore cortical inputs is recruiting recently revealed complex and various local striatal micro-circuits whom interplay will lead to complex effect combining inhibition and disinhibition. Local disinhibitory role of interneurons (such as VIP or SOM interneurons in cortex or amygdala) has been recently highlighted in several brain structures as an important mechanism to regulate the input/output flow of information⁵³.”

4. The depolarizing effect of GABA inputs has been reported in numerous papers, including some done in the perforated patch configuration. It is clear that when SPNs are recorded ex-vivo, resting

between -80 and -90mV, GABA_A will have a depolarizing effect, although the “real” reversal of GABA_A in SPNs has a wide range of reported values, from near threshold (~-45 mV) to lower than -70 mV. In addition, SPNs lie at such hyperpolarized membrane potentials under ex-vivo conditions, however in-vivo recorded SPNs display more depolarized and variable membrane potentials ranging between -75 to -55 mV (See examples Sippy et al 2015). It is not entirely clear what is new in the presented results, other than that stronger inhibition will result in stronger hyperpolarization above reversal and stronger depolarization below reversal. If the “depolarization” by GABA inputs was significant, wouldn't a ChR2 “pre pulse” before cortical stimulation enhance the SPN spiking probability in experiments presented in figure 2?

We agree with Reviewer#2 that the depolarizing effect of GABA has been described before and our intent was not claim priority for this. Nevertheless, we are convinced that showing a functional role for this depolarizing effect in the summation of sustained subthreshold cortical activity (Fig. 6) is new and of significant importance for striatal physiology.

We performed pre-pulse light stimulation before cortical stimulation but we did not see any major differences. We think the reasons are multiple: (1) the stimulation to trigger the spike is too strong since it recruits a lot of cortical cells simultaneously. Maybe for single inputs it could be easier to see but very hard to reveal experimentally and (2) another reason could be that the depolarizing effect is effective only below -60mV (and decreasing in power as the SPN will get closer to -60mV), which is the reversal of chloride in our experiments. Above -60mV, the effect would become hyperpolarizing. Therefore the effect of GABA we observed would be more relevant for small amplitude EPSPs below -60mV but not for activity whose amplitude crosses the reversal of chloride, which is the case for the near-suprathreshold stimulation used here. For such stimulations, the relative latencies of the pre-pulse and cortical stimulation would be the critical factor, so the (depolarizing) IPSP would fall on the rising phase of the EPSP, but before it crosses E_{Cl} . Considering the very fast rising phase of the cortically-evoked EPSPs in this protocol, it would be difficult to fall in the precise window where the IPSP could be depolarizing.

Finally, our *in vivo* results show an overall increase in spontaneous SNr activity after silencing striatal PV or SOM interneurons. One hypothesis (among others) could be that the depolarizing effect of GABA plays a role *in vivo* in the integration of cortical inputs. Therefore, silencing striatal interneurons could lead to a decreased summation of spontaneous cortical inputs, and an overall increase in SNr firing.

Altogether, the depolarizing GABA might have a determinant role in the integration of weak inputs from the cortex. This additional regulation to the modulation of the spiking activity provide to the interneurons a dual role that we bring together in the present manuscript.

5. The effects depicted in figure 6 are very small (although statistically significant) and could arise from different reasons rather than the depolarization by GABA_A. In addition to the hyperpolarization and depolarization, convergent GABA input also affects the membrane conductance of recipient SPNs. This would predict opposite results in terms of the decay time constant (more inhibitory conductance shortening membrane time constant). What mechanism is proposed for the small changes observed only in the decay time-constant and not any other response properties? The authors should report changes in membrane conductance and time constant in SPNs in different conditions, especially in view of the potential release induced by Arch inhibition, which could actually result in increased GABA input rather than decrease. Moreover, the complete lack of effect of DMS-PV or DLS-SOM photo-inhibition indicates no connectivity whatsoever, which is not in line with the previous data.

The effects on subthreshold activity summation we described in Fig.6 are significant and we do believe that they show functional importance of depolarizing GABA in the integration of “weak” cortical inputs.

We think that the mechanism explaining a strong effect on the EPSP decay and not on other parameters is the synaptic delays. When stimulating the cortex, we recruit both SPNs and interneurons, cortically-evoked EPSP will come before the IPSP resulting from the feed-forward

inhibition. The IPSP would thus fall in the decay time of the EPSP and could not influence the amplitude or rise but mainly the decay.

As previously discussed in the Point 2, it is very unlikely that we have release induced by Arch stimulation (Mahn et al.) given the short duration of the LED pulses we are using. In addition, we are showing in new Supplementary Fig. 3 that yellow LED has no effect on the membrane fluctuations of SPNs while they are at their resting membrane potential, which is way below the Cl⁻ reversal so supposedly where depolarizing GABA could have a stronger effect.

The lack of effect of DMS-PV and DLS-SOM is not due to an absence of connectivity; we performed paired recordings of PV-SPN and SOM-SPN on DMS and DLS and observed connectivity in all the conditions. These results are shown in Supplementary Fig. 6 and in the attached figure. The connectivity and the synaptic weight is present for PV and SOM cells in DMS and DLS, but as described in the previous Figures of the paper, it is not homogeneous. Therefore the inhibitory weight of PV-DMS cells and SOM-DLS cells is weaker as compared to SOM-DMS

and PVDLS. As a consequence, some SPNs are influenced by PV opto-inhibition in DMS and SOM opto-inhibition in DLS. The results presented in Figure 6 are in line with other results from the study such as results presented in Figure 2 where we can see that the overall effect on the spiking probability is stronger for SOM-DMS and PV-DLS even though the connections from PV-DMS and SOM-DLS are present. We think that the observed differences are mainly due to distinct properties of PV and SOM cells in the two territories such as anatomical and excitability differences described in Figure 5.

6. Previous data suggested differences in the inhibition of dSPNs and iSPNs by PV interneurons (Gittis et al, 2010 and 2011). It would be important to provide this information for the DLS and DMS as well as for selectivity of SOM inputs to the 2 SPN pathways. Was there any attempt to uncover the type of SPN following recordings?

Several papers indeed addressed the issue of the repartition of the striatal inhibition between dSPNs and iSPNs. Gittis et al reported a slight difference in PV innervation between dSPNs and iSPNs (52% dSPNs and 36% iSPNs in Gittis et al., 2010 and 60% dSPNs and 47% in iSPNs in Gittis et al., 2011). The difference is quite low and the same year another study described no difference in PV innervation onto dSPNs and iSPNs (Planert et al., 2010). More recently Sabatini's lab compared the innervation of PV but also SOM interneurons on both subtypes of SPNs and they described no difference (Straub et al., 2016).

Our main focus in this study was to compare inhibitory networks between DMS and DLS so, based on the literature, we assumed that the inhibitory weight from PV and SOM were similar on both direct and indirect-pathway SPNs.

In addition, our *in vivo* experiments using stimulation of cortical afferents show that striatal interneurons had specific effects on the evoked response in SNr, depending on the territory: opto-inhibition of SOM-DMS led to a decrease of the inhibition phase, and opto-inhibition of PV-DLS led to a decrease of the late excitation phase, while other combinations had no effect. We could have concluded that SOM/PV interneurons only affect each direct/indirect pathway, respectively, but we think the interpretation has to be more careful. Indeed, as mentioned in the answer to the point 3, several explanations such as the mixed kinetics of direct/indirect pathway responses in SNr, the difference in excitability of iSPNs and dSPNs and the specificities in the cortical inputs innervating DMS and DLS can explain these results. Therefore we do not think that there is a specific innervation of PV and SOM cells to one or the other subtype of SPNs.

7. The results regarding the connectivity and relative synaptic amplitudes in SOM and PV cells do not agree with several other papers such as Gittis (2010), showing that excitatory input to LTS cells is much smaller than that to PVs, and that FS to SPN connectivity is much stronger and prevalent than LTS-SPN connectivity. Similar results using optogenetic stimulation in PV/SOM Cre mice are shown in Straub et al (2016) showing stronger responses in SPNs following PV than SOM optogenetic stimulation. The authors should address these discrepancies.

We do believe that there are not real discrepancies if we consider the DLS territory, precisely where the studies (Straub et al., 2016, Gittis et al., 2010) have been made. Indeed, in DLS we also see a significant larger responses from PV compared to SOM cells (both in paired recordings and optogenetic experiments). The previous studies did not compare PV and SOM in DMS

where we actually see a reversed picture compared to DLS: SOM cells elicit stronger light-evoked IPSP compared to PV cells. It should be noted that we did not see any significant difference in the unitary IPSP amplitudes with paired recordings. The results are described in the figure included here.

We have included these aspects and compare our results with the literature in the Discussion section:” PV cells are known to exert a strong inhibitory weight on SPNs since they delay or even stop the spiking activity in SPNs^{15,36,37}. SOM cells (also expressing NPY and NOS) are also able to delay AP in SPNs^{15,36}; they have been initially reported to have a lower connection probability with a weaker weight onto SPNs³⁸ but higher amplitude of evoked-responses of SOM-SPN connections after SOM opto-activation was recently reported^{39,40}, similar to what we observed in the present study. Also, NPY expressing interneurons (overlapping with SOM cells) also strongly inhibit SPNs⁴¹. The discrepancies in the amplitude of SOM-SPN connections between studies could be explained by the location of recordings since most of the recordings in these studies have been done in the DLS where the synaptic weight of the SOM cells onto SPNs is weaker compared to DMS.”

Concerning the differences in cortical inputs on the 2 cell types. Gittis et al described a smaller subthreshold amplitude (EPSCs) in SOM cells compared to PV cells. In the present study, our aim was not to study the subthreshold responses of interneurons to cortical activity but their spiking activity. We describe that the spiking probability and the ability of evoking a spike are highly enhanced in SOM cells due to their very high Ri and low rheobase but it does not disagree with eventual differences in subthreshold events. Since it is the spiking activity that is of interest here (we need to make them spike to observe their weight on MSNs), we are using suprathreshold stimulations and the subthreshold activity is not at play here.

Therefore we do not think that there is a discrepancy about the cortically-evoked responses since we did not look at the same parameters.

8. The use of “specific” circuits engaged in DLS vs DMS is slightly misleading as there seem to be somewhat of a gradient but no evidence for specific circuitry devoted for the respective striatal regions.

We used the term “specific” for the combination of the differences between the numbers and the electrical properties of PV and SOM cells in both territories. It is indeed the same cells involved so maybe the word specific is a bit misleading. We changed this to more accurate words such as “different circuits” or “functional differences” in the manuscript.

Minor comments:

1) Distribution and activity of PV cells has already been shown to follow a dorsolateral to ventro-medial gradient (Berke, 2004). This paper should be referred to in the context of regional differences in the interneuron populations.

We referred to the very first description of the PV gradient by Gerfen in 1995 but we agree that Berke et al. 2004 also described this regional difference. We have now added this reference in the manuscript.

2) Is there any reason for using 635 nm in-vivo vs. 570 nm in ex-vivo optogenetic stimulation?

We are using a laser-diode coupled to an optical fiber for *in vivo* experiments, which requires a stronger source than the LED used in *ex-vivo* setups in wide-field (due to the loss of power induced by the collimation into a 200 μ m-core optic fiber). Such laser diodes do not exist with enough power at 570nm but only at 635nm, which is slightly off-peak, but well within the excitation range for Arch3.

3) The assumption of stimulating only 3-4 cells (page 7) per slice is not justified and was not rigorously tested by recording a responding interneuron and shifting the objective to measure the “envelope” surrounding the cell body which would still evoke spiking. Moreover, this may also be used to test whether DMS interneurons might be activated by DLS optogenetic activation/inhibition, and vice versa.

The number of 3 to 4 cells that were activated by the light is just a simple ratio between the light-evoked IPSP amplitude and unitary (paired recordings) IPSP amplitudes. We just wanted to have a global idea of how many synaptic inputs were at play in the effects we were observing on SPNs. We do agree that it is a simple way of measuring it and probably it could correspond to more cells but “incompletely” stimulated because in our wide field stimulation we have few somas but a lot of dendrites/axons susceptible to be activated by the LED.

Concerning the fact that we could influence the DMS while opto-stimulating/inhibiting the DLS and vice-versa, we think we can rule out this hypothesis for several reasons: (1) in our DMS corticostriatal slices, there is only DMS and in the corticostriatal DLS slices there is only DLS, which means that it would be impossible for us to activate the other territory within the same experiment and, (2) our new set of experiments using viral infections of either DMS or DLS show similar results, meaning that only local circuits are implicated.

4) The recruitment curves in figure 2 have no error bars, and it is not clear how they were obtained using “a.u.” in different slices. How was it established that SOM cells are recruited with lower stimulation than PV cells if they were not recorded under the same slices and electrode positioning and using arbitrary units? What does this graph actually tell us about the relative recruitment order if they were not consistently recorded simultaneously (PV,SOM,SPN)?

We indeed did not record systematically simultaneously the three SOM, PV and SPNs to compare the I/O curves of the different cell types. Nevertheless, we did not mean that the stimulations we used were done with arbitrary units since they correspond to a real increasing scale of intensities in our stimulator. We did remove the term “a.u.” which is confusing. We normalized the spiking probability to the minimal intensity stimulations we were able to detect a subthreshold response in the different cell types. Thus we think we can use statistical approaches to compare results from independent recordings. In addition, we would like also to emphasize that we are using brain slices preserving specifically the connections between layer 5 cortex and adjacent striatum. The intensities used in different sets of experiments are very reproducible to recruit the different striatal cells.

We do agree with Reviewer#2 that we had to include error bars on these graphs. We now included them and they show a low variability for each cell type.

Our point with these I/O curves was to show that both PV and SOM interneurons are recruited before SPNs in our conditions both in DMS and DLS. Indeed, we already described shorter latencies of

cortically-evoked responses in PV interneurons (Fino et al., 2008; Paillé et al., 2010) but wanted to make sure that it was similar for SOM cells and for both PV and SOM in DMS.

To be completely convinced, in our subset of experiments performing paired recordings of PV-SPN and SOM-SPN, we were able to compare the recruitment of both cell types in the same recordings both in DMS and DLS. In all the recorded pairs, PV and SOM interneurons are recruited for lower intensity than SPNs (PV-SPN: n=7 pairs in DLS and n=5 pairs in DMS; SOM-SPN: n=3 pairs in DMS and n= pairs in DLS). These results are in accordance with the I/O curves we obtained with independent recordings.

Finally, the fact that SOM cells are recruited for lower cortical stimulation intensity is also in accordance with our results illustrated in Figure 4 showing a higher excitability of SOM cells compared to PV cells.

5) In addition to the cortical input there is strong innervation from thalamic nuclei. The authors should address this pathway as well, since it was shown to: a) be a strong input to the striatal circuit, with comparable numbers of afferent terminals b) have different target preferences in terms of striatal neuron targets (Ding et al, Neuron 2010, and Parker et al, Neuron 2016).

We focused here on cortical inputs to the striatum. We agree that thalamic inputs are also strongly influencing striatal activity and are specific to their targets. In addition a recent study showed a differential integration of thalamic inputs by several striatal interneurons (Assous et al., 2017). Thus it would be important to decipher how interneurons modulate these inputs in SPNs.

We have added this in the Discussion part: "We would like to point out that we focused here on feed-forward inhibition mediated by cortical inputs but future studies should extend this work to another major striatal input coming from the thalamus."

6) Another result of using the Chr2/Arch3 reporter as opposed to viral expression is that using the SOM-Cre with reporter will also infect some SPNs (Straub et al Neuron 2016).

We indeed observed some expression of Chr2/Arch3 in cells that were not PV or SOM cells. Nevertheless the percentage was very low since in PV-ChR2 or PV-Arch3 less than 10% of the YFP/GFP+ cells were not interneurons (Supplementary Fig. 1). For SOM-ChR2 and SOM-Arch3, this percentage was 10-15% (slightly less than Straub et al., 2016). The most important for the purpose of our study is that this percentage is the same in DMS and DLS for the different transgenic mice, to not induce a bias regarding the specific action of interneuron subtypes in the different territories. We indeed did not observe difference between the YFP+/GFP+ cells that were not SOM+ in DMS and DLS.

Finally, our additional experiments with targeted viral infection of striatal interneurons with AAVs vectors, showed that 100% of infected neurons were PV or SOM cells (in accordance with Straub et al., 2016). Since we observed similar results with viral infection and transgenic mice, we can rule out a confounding effect of non-specific expression of opsins in SPNs.

7) Optogenetic stimulation of SOM and PV expressing Chr2 would cause different number of spikes, even in relatively short (5 ms) and definitely in longer (several hundreds of ms) in the respective cell types. How was the output of the interneurons controlled in the activation experiments? Were differences in the responses of SOM and PV cells to the same stimulation observed? How many spikes were evoked by a 5 ms light pulse?

We agree with Reviewer#2 that we did not include a quantification of opto-activation of interneurons in the first version of the manuscript. We have performed this quantification and observed that there was no major difference in the activation of PV and SOM cells in DMS and DLS. Both in DMS and DLS, PV 5ms optical stimulation of PV cells induced 1AP in 100% of neurons. A 300ms optical stimulations induced 14.30 ± 0.68 APs in DLS and 14.85 ± 2.28 APs in DMS (n=8 PV cells). Concerning SOM cells, a

5 ms optical stimulation induced 1AP in 99% of neurons both in DMS and DLS. A 300 ms optical stimulations induced 13.25 ± 0.94 APs in DLS and 12.67 ± 1.20 APs in DMS (n=7 SOM cells).

We have now added this information in the legend of the new Supplementary Fig. 6.

8) Numerous typos and grammatical errors throughout the text. Please check thoroughly.

A few examples:

- p2 (Abstract): "...in spiking SPNs, GABAergic interneurons potently inhibit spiking SPNs (feed-forward inhibition) while in resting SPNs, they favoring cortical activity..."
- p5: "...Striatum acts as coincidence detectors..."
- p6: "...To investigate the cause of this differential effects..."
- p7 (143-146): decreased instead of increased? "...PV cells opto-activation increased the spiking probability of SPNs in DLS ($p=0.0059$, $n=9$) but not in DMS ($p=0.9816$, $n=9$) while SOM opto-activation increased spiking probability in DMS ($p=0.0002$, $n=7$) but not in DLS ($p=0.9114$, $n=6$) (Fig. 2h-i)..."
- p10: references the wrong figure (3 instead of 4).
- p12: "...GABA current show a strong depolarizing effect..."
- p16: "...but highe amplitude of evoked-responses..."
- p17: "...Indeed, NPY expressing interneuron (also expressing somatostatin) also strongly inhibits SPNs..."
- p17: "...In addition, it exists also a feedback inhibition..."
- p17: "...recent evidences show that GABAergic..."

We have now corrected all these mistakes.

Reviewer #3 (Remarks to the Author):

This is a very interesting manuscript by Fino et al studying subregion specific properties of GABA interneurons in the striatum. The authors provide evidence that PV interneurons are more effective in controlling MSN activity in the DLS (due to their higher density) whereas SOM interneurons are more effective in the DMS (possibly due to their higher excitability in the DMS). Both neurons subtypes provide feed forward inhibition for above threshold cortical input but have depolarizing effects at resting state thereby positively affecting cortical integration/summation.

This is an extensive and carefully performed slice physiology study. The authors add in vivo significance to their findings by measuring basal ganglia output (SNr) activity after inhibition of PV and SOM interneurons anesthetized animals.

The manuscript is well written and of interest for a broad audience. I have some technical points that would need to be addressed:

Figure 1: I couldn't find the number of animals in the experiments for figure 1. This is important to know because bias could have been induced by one or two animals. Also, although I believe that the recordings have been performed in anesthetized animals this hasn't been explicitly stated in the result section.

We have now explicitly indicated the numbers of animals for the *in vivo* experiments and mentioned that the experiments were performed in anesthetized animals.

Figure 1: It seems that the location of the recording probe is the same for DMS and DLS stimulations. Therefore differences in the response to interneuron inhibition when DMS and DLS are compared could be due to differential connectivity of to the specific recording site. For example DMS interneuron inhibition may just have missed the target. This needs to be clarified.

Because we used 4-shank probes with 200 μ m spacing between shanks, which spans a generous part of the mouse SNr, we did not think it necessary to adapt the recording location to the targeted territory. Nevertheless, we agree with the Reviewer#3 that the functional organization of the SNr might still impact the differential results seen between DMS and DLS territories -- though it should not impact differential results between PV and SOM opto-inhibition in the same territory. We have now addressed this issue in new Supplementary Fig. 2 and new Supplementary Fig. 12a:

We compared the effects of opto-inhibition of PV or SOM cells in DMS or DLS on spontaneous SNr activity with the recording location of the SNr units. First, we observed that individually significantly modulated units were scattered all over the range of the recording sites, including in the most medial and lateral ranges explored, in each of the 4 conditions tested (Supplementary Fig. 2a). We quantified this relationship by two means (1) we divided our data set in medial and lateral parts, and compared the proportion of individually significantly modulated units, and the median change in firing rate between the medial and lateral parts. We found no significant difference in either metrics, for the 4 conditions tested (Supplementary Fig. 2b); (2) we correlated the median change in firing rate with either the depth and laterality of the recorded unit, and found no significant correlation between these parameters in any of the 4 conditions tested (Supplementary Fig. 2 legend).

With the additional experiments using cortical stimulation, we could address this issue even more directly. Typical cortico-nigral responses include a combination of 3 main phases: (1) an early excitation thought to correspond to the hyperdirect cortico-STN-SNr pathway, (2) an inhibition thought to reflect the trans-striatal direct pathway (cortico-striato-SNr) and (3) a late excitation thought to correspond to the activation of the trans-striatal indirect pathway (cortico-striato-pallido-STN-SNr) (Fujimoto et al., 1992; Kolomiets et al., 2003; Sano et al., 2013). If the recorded site in the SNr is connected to the striatum region that receives the cortical input (DMS for CG2 stimulation), the SNr unit will display at least one of the trans-striatal phases. We compared the responses of SNr units to cortical stimulation in our 4 conditions (Supplementary Fig. 12a), and observed that (1) the overall distribution of response patterns was similar across conditions; (2) the proportion of each individual phase (in particular the trans-striatal inhibition and late excitation) was similar across conditions; (3) the proportion of full responses was similar across conditions (consistent with previous studies).

Our results are consistent with the partial convergence of cortico-nigral pathways upon SNr functional territories. Indeed, while the SNr does display a lamellar organization relative to striatal projections, convergence between pathways can occur, if only because part of their dendritic tree extend beyond the main striatal projection region of their soma (Mailly et al., 2001). Moreover, individual SNr neurons have been shown to respond to stimulation of very different cortical areas (e.g. auditory and motor, or motor and prefrontal) (Kolomiets et al., 2003). It is therefore not unlikely that associative and somatosensory cortex can converge on a wider part of the SNr, as seen in our results.

Therefore, the differences observed between the effects of PV and SOM opto-inhibition in DMS and DLS territories are not due to a bias in SNr anatomical targeting.

Figure 2: It looks like that PV::Arch3 inhibition has a similar effect size though with higher variability in DMS compared with DLS. A higher n would be needed for the PV-DMS condition to clarify this.

We did increase the n by doing similar experiments with viral injections instead of transgenic animals. We performed such experiments to make sure that the effects we observed were due to local striatal interneurons on not from PV or SOM cells projecting to the striatum. We observed similar effect with virally induced expression of opsins that

for the transgenic animals. More specifically for the spiking probability in PV-DMS conditions we recorded 9 SPNs and did not see any significant effect of the opto-inhibition of PV cells on SPN spiking in DMS ($p=0.8007$), contrarily to DLS ($p=0.0063$) (see Supplementary Fig. 5). Since we observed similar effect, we pooled both experiments and illustrated it in the attached figure. The total n is therefore 17 neurons in DMS and 19 neurons in DLS. We can confirm that there is a strong effect of PV cells opto-inhibition on SPN spiking probability in DLS and not in DMS.

Page 7 line 143 the authors write: "PV cells opto-activation increased the spike probability"; to my understanding this should read: "decreased the spiking probability"

It was indeed a mistake, we have now corrected it in the text.

Page 8: line 171 Fig. 2e should be Fig. 3c

We have now corrected all these mistakes.

Figure 3: The authors should provide some quantitative data to confirm that the optogenetic stimulation employed leads to a comparable activation of SOM and PV interneurons in DLS and DMS (the four conditions). For example, a 5 ms optical stimulus with stimulation intensity X induces 1 AP in Y% of neurons; 300 ms optical stimulation with stimulation intensity X induces Y+-Z APs in N % of neurons.

We agree with Reviewer#3 that we did not include a quantification of opto-activation of interneurons in the first version of the manuscript. We have performed this quantification and observed that there was no major difference in the activation of PV and SOM cells in DMS and DLS. Both in DMS and DLS, PV 5ms optical stimulation of PV cells induced 1AP in 100% of neurons. A 300ms optical stimulations induced 14.30 ± 0.68 APs in DLS and 14.85 ± 2.28 APs in DMS ($n=8$ PV cells). Concerning SOM cells, a 5 ms optical stimulation induced 1AP in 99% of neurons both in DMS and DLS. A 300 ms optical stimulations induced 13.25 ± 0.94 APs in DLS and 12.67 ± 1.20 APs in DMS ($n=7$ SOM cells).

We have now added this information in the legend of the new Supplementary Fig.6.

Figure 5C is described but not explicitly mentioned in the result section.

We have now added the Fig. 5c in the Results section.

"Interestingly, at hyperpolarized states, GABA mediates a depolarizing current (Fig. 5a-c). This is explained by the fact that SPNs membrane fluctuations widely crosses E_{Cl} threshold (physiological $E_{Cl} \sim -60$ mV^{16,25}) and GABA is inhibitory for $E_h > -60$ mV and depolarizing for $E_h < -60$ mV."

Reviewers' comments:

Reviewer #1 (Remarks to the Author):

The study has been improved greatly, especially due to the inclusion of viral injection-based protocols, examination of the triphasic SNr response to cortical stimulation, and more complete basic characterization of interneuron connectivity. The authors are to be commended for adding so much additional data to what was already an extensive study. While almost all the concerns raised in the initial review have been addressed, there is one remaining concern about the new in vivo data, as outlined below:

While the new in vivo experiment with cortical stimulation is well-designed and informative, the interpretation of the data in figure 7d is not as straightforward as the authors suggest. The decrease in area of the inhibitory phase is very small, and the single unit example shows no change in latency to onset or peak percent inhibition. This effect could be due to a decrease in the latency to onset of the late excitatory phase. Furthermore, the amplitude of the short-latency excitatory effect appears to be decreased (although it is difficult to determine what baseline was used to measure this component). This would seem to be an effect on the hyperdirect pathway which raises concerns about the use of breeding rather than viral expression to achieve Arch expression. The problem may stem from the choice of representative unit or the analysis itself, which is hard to tell from the figure as currently constituted. Would it be possible to present plots of the time course of the average response in all units with some sort of representation of the error? If this is not possible, perhaps a different representative unit should be chosen.

Reviewer #2 (Remarks to the Author):

The revised manuscript addressed many of the comments and the paper has improved, however there are still remaining issues that, in the view of this reviewer, diminish the impact of this study.

Most of the data is still based on the transgenic mice with ChR2 or Arch3 reporters, thus strongly confounding the interpretation of the results, as many different neuronal populations are labeled in addition to striatal SOM and PV interneurons. Axons from PV/SOM neurons in midbrain, cortex, GP can all affect the results, especially in vivo, where all of these pathways are intact and robust stimulation is used. The PV expressing cells in GP project to striatum (mainly interneurons) but also to arky pallidal neurons, which in turn project back to striatal projection neurons. Moreover, even though the respective contribution of PV and SOM to SNR discharge rate differs topographically, the degree of modification is small (single percents in the SOM effect) and in both DLS and DMS the PV are clearly more effective in modulation. These data do not fully support a "differential effect" of the interneurons in terms of their location. Rather, a relatively small (compared to the difference between PV and SOM in both areas) change in their relative impact between DMS and DLS is seen. Some slice experiments were now done using viral injections and presented as supplementary material. Even though they support some of the previous findings in crossed transgenic animals, it is still unclear how they fit with the in vivo data, which overall reduces the impact of the study.

The total lack of effect by stimulating arch-expressing terminals is surprising since the effects seen in Mahn et al are indeed very large and have been seen in other pathways (and groups) since. Even if the overall impact of photo inhibition would be to completely silence the interneurons and their terminals, wouldn't the authors expect a change in SPN membrane potential when silencing SOM interneurons, if only due to their tonic activity?

Figure 2: the spiking depends on various factors, especially in whole cell recordings, the holding potential. At what potential were they held? Why not do these recordings in cell-attached mode and record only spike responses? These results show that SOM and PV are different from each other in

both areas, rather than a pronounced topographical difference. The results in 2H and 2K are in sharp contrast to those implied from the in vivo recordings, showing higher (than SOM) impact of DMS PV cells on SNR firing. In the slice recordings it seems that PV manipulation does not affect the firing of SPNs.

Figure 3: why was the measure of change in instantaneous frequency used? What period was compared between the frequencies? What was the actual synaptic strength that caused these frequency changes?

Figure 5: DMS SOMs provide stronger than PV ipsp's onto SPNs. Again, not completely in line with the in vivo results. The reversal potential for GABA_A is determined by the pipette solution and it is therefore not surprising that both interneuron types evoke depolarizing responses at a membrane potential of -80 mV. The novelty of this is not clear, as well as the validity of -60 mV as the reversal of chloride under physiological conditions. It would be much more convincing if experiments were done in actual chloride concentration (perforated or cell attached recordings).

Reviewer #3 (Remarks to the Author):

The authors addressed all my concerns.

Please find below our answers (in blue) to the Reviewers comments. In summary, in the new version of the manuscript we have now included a new Figure 7 (including a better representative experiment) and a new Supplementary Table 2 recapitulating all the parameters measured on the different phases of *in vivo* evoked SNr response. In addition, we have included in the manuscript different clarifications based on the Reviewers comments that we highlighted in yellow for convenience.

Reviewers' comments:

Reviewer #1 (Remarks to the Author):

The study has been improved greatly, especially due to the inclusion of viral injection-based protocols, examination of the triphasic SNr response to cortical stimulation, and more complete basic characterization of interneuron connectivity. The authors are to be commended for adding so much additional data to what was already an extensive study. While almost all the concerns raised in the initial review have been addressed, there is one remaining concern about the new *in vivo* data, as outlined below:

While the new *in vivo* experiment with cortical stimulation is well-designed and informative, the interpretation of the data in figure 7d is not as straightforward as the authors suggest. The decrease in area of the inhibitory phase is very small, and the single unit example shows no change in latency to onset or peak percent inhibition. This effect could be due to a decrease in the latency to onset of the late excitatory phase. Furthermore, the amplitude of the short-latency excitatory effect appears to be decreased (although it is difficult to determine what baseline was used to measure this component). This would seem to be an effect on the hyperdirect pathway which raises concerns about the use of breeding rather than viral expression to achieve Arch expression. The problem may stem from the choice of representative unit or the analysis itself, which is hard to tell from the figure as currently constituted. Would it be possible to present plots of the time course of the average response in all units with some sort of representation of the error? If this is not possible, perhaps a different representative unit should be chosen.

We would like to thank Reviewer 1 for his positive comments about our revised manuscript.

Reviewer 1 has concerns about the representative experiment shown in Figure 7d. In fact, we agree that these traces may appear confusing as they may suggest an effect on the hyperdirect pathway. We did not appreciate this potential confusion while making the figures as it is often difficult to show representative data that recapitulate the different features of the analysis. However, we would like to stress out here that we provided quantitative data in the manuscript and Supplementary Figure 12b showing the absence of unspecific effect on the hyperdirect pathway.

We have now replaced the representative trace of the SOM-DMS condition by a different unit in the new Figure 7d, as suggested by Reviewer 1. We chose to show representative examples and not averaged traces since it would be meaningless to average the response over all neurons. Indeed, as shown in Supplementary Figure 12, not all neurons show the 3 phases of the cortico-nigral response. Moreover, the precise timing of each phase differs among neurons, so a group average would end up mixing at a given time the offset of a phase with the onset of the next. That is why it was only possible to quantify the phase parameters individually on each neuron PSTH. We therefore show a representative example

for the response, and the group data on the quantified parameters. In addition to the representative examples, we have now added a new Supplementary Table 2 that compiles all quantified parameters for the 3 phases in the four conditions.

Concerning the baseline, as previously indicated in the methods, the baseline used to measure the early phase was the same as for the other phases, i.e. the 50 ms before the stimulation. If the onset of a component (where the PSTH crosses the baseline) was masked by the stimulation artefact (since 2ms on each side of the artefact are excluded from the analysis), it was taken as the first “analysable” bin (i.e. the bin centered on 2.5 ms). This precision has now been added to the methods section: “If the onset of a component was masked by the stimulation artefact (since 2ms on each side of the artefact are excluded from the analysis), it was taken as the first “analysable” bin (i.e. the bin centered on 2.5 ms)”.

To be noted, as shown in the new Supplementary Table 2, the change in peak percent of the inhibition phase is indeed not significant for the inhibition phase (while the decrease in area is significant). This is expected, since inhibition amplitude is mathematically capped at -100%, and most cells display a plateau at -100%. It is thus expected that the representative example does not display this change. (The peak percent change is significantly decreased for the late excitation phase in PV-DLS condition, as seen in both the group data (Supplementary Figure 12 and Supplementary Table 2) and the representative example in Figure 7).

We agree with Reviewer1 that the decreased duration of the inhibition phase when inhibiting SOM interneurons in the DMS could be due to a decrease in latency of the late excitation phase. Indeed, both the offset of the inhibition phase and the onset of the late excitation phase are significantly decreased in this condition (Supplementary Table 2). So it could either be an actual decrease in the inhibition phase intensity (duration, area) that leads to a decreased latency of the late phase, or a change in the latency of the late excitation that causes the inhibition phase to be shortened. However, the duration, peak and the area of the late excitation phase are unchanged, arguing against a strengthening of the late excitation phase. This implies that for a SNr neuron, the outcome will still be less inhibition (and an earlier late excitation, but not a stronger one), whether the original cause lies in the change of the inhibition phase, of the late excitation phase, or of both. Because of this common outcome, we focused on the inhibition phase and did not go into details about all the parameters we had measured.

To this end, we have now added the new Supplementary Table 2 that includes all the parameters. In addition, we are now mentioning this point of interpretation in the discussion: “Though it would be tempting to jump to the conclusion that SOM/PV interneurons only affect each direct/indirect pathway, respectively, we think the interpretation should be more careful since the activation of the direct and indirect pathways are overlapping, and the phases that can be measured, while mainly corresponding to each pathway, are still the sum of this overlap.”

Reviewer #2 (Remarks to the Author):

The revised manuscript addressed many of the comments and the paper has improved, however there are still remaining issues that, in the view of this reviewer, diminish the impact of this study.

Most of the data is still based on the transgenic mice with ChR2 or Arch3 reporters, thus strongly confounding the interpretation of the results, as many different neuronal populations

are labeled in addition to striatal SOM and PV interneurons. Axons from PV/SOM neurons in midbrain, cortex, GP can all affect the results, especially *in vivo*, where all of these pathways are intact and robust stimulation is used. The PV expressing cells in GP project to striatum (mainly interneurons) but also to arkypallidal neurons, which in turn project back to striatal projection neurons. Moreover, even though the respective contribution of PV and SOM to SNr discharge rate differs topographically, the degree of modification is small (single percents in the SOM effect) and in both DLS and DMS the PV are clearly more effective in modulation. These data do not fully support a “differential effect” of the interneurons in terms of their location. Rather, a relatively small (compared to the difference between PV and SOM in both areas) change in their relative impact between DMS and DLS is seen. Some slice experiments were now done using viral injections and presented as supplementary material. Even though they support some of the previous findings in crossed transgenic animals, it is still unclear how they fit with the *in vivo* data, which overall reduces the impact of the study.

Reviewer 2 expresses concerns about the use of transgenic mice and about *in vivo* data from Figure 1 as originally stated in the first round of revisions. We are very surprised by these comments as we put a substantial amount of efforts to provide new experiments that fully address these important concerns.

First, we repeated all *ex vivo* experiments using viral injections in Cre mouse lines, as recommended by Reviewers 1 and 2, to rule out any unspecific effect from projecting interneurons (pallidostriatal, corticostriatal...) that may exist in transgenic mice. The results we obtained with viruses are completely in line with our previous data using transgenic mice (Supplementary Figures 4, 5, 7 and 11). If preserved terminals from cortex or GP in brain slices could have led to artefactual recruitment by LED and to unspecific effects, we would have seen some *ex vivo*, but there was none. It is therefore reasonable to use transgenic mice for *in vivo*. Moreover, using virus injections for *in vivo* would add a variability in the extent of the expression that would outweighs its benefits. Also, very importantly, we are not using the opto-activating ChR2 for the *in vivo* experiments but only the opto-inhibiting opsin Arch, which is less likely to affect striatal-projecting PV axons, since terminal-only opto-inhibition with Arch is weak (Mahn et al., 2016). And in addition, to ensure that we did not have any unspecific effects, we used short enough light pulses to avoid increased synaptic release (see also our answer to the following point of Reviewer2). Our new sets of data thus clearly argue against the existence of unspecific effects from other interneuron populations and validate the specificity of the response in transgenic mice.

Second, we put a lot of effort to strengthen our previous *in vivo* data (presenting SNr *in vivo* spontaneous activity in Figure 1) by designing a new set of experiments using *in vivo* cortical stimulations, as suggested by Reviewers 1 and 2. We indeed agreed that the initial concerns about the spontaneous activity were critical for our conclusions. This is the reason why we decided to perform additional experiments to provide direct *in vivo* evidence of the functional role of PV-DLS and SOM-DMS differential effect by using cortically-evoked activity. These new data, which are now presented in Figure 7, fully address Reviewer 2's concerns about Figure 1 as it clearly shows the *in vivo* relevance of the differential effect of striatal interneurons. We indeed observed that only SOM interneurons silencing in DMS and PV silencing in DLS modulated cortically-evoked SNr activity. This constitutes a functional validation *in vivo* of our observation in brain slices. Very surprisingly, although this point was a main concern in the first round of revision, Reviewer 2 does not mention the new Figure 7 and still questions the *in vivo* data interpretation solely based on Figure 1. We do not understand how Reviewer 2 could have missed the new Figure 7, which is now central to the manuscript.

The total lack of effect by stimulating arch-expressing terminals is surprising since the effects seen in Mahn et al are indeed very large and have been seen in other pathways (and groups) since. Even if the overall impact of photo inhibition would be to completely silence the interneurons and their terminals, wouldn't the authors expect a change in SPN membrane potential when silencing SOM interneurons, if only due to their tonic activity?

As explained in our first point-by-point letter, the length of light pulses we are using in our study is much shorter than the one presented in Mahn et al. Therefore we can exclude any unspecific effects since they are appearing for very long light pulses. "In their paper, Mahn et al. indeed described an opposite effect of Arch activation but this undesirable effect appears for very long light stimulations since they apply several minutes of light ON. Their results using shorter pulses (200 ms, figure 2 of the article) show that illumination of axonal terminals expressing Arch3 does slightly attenuate synaptic transmission but does not increase it. In our conditions, we analyzed the first 300 ms from maximum 1 sec long the light pulses, therefore avoiding any artefactual release."

In addition, we indeed observed that LED illumination has no effect on the membrane potential of SPNs (data presented in Supplementary Figure 3), excluding the possibility of Arch-triggered synaptic release in our conditions. Reviewer 2 is surprised by the absence of major effect of light on SPNs due to spontaneous activity of SOM interneurons but it seems rather logical to us. Indeed, based on our recordings, we observed that 3 to 4 SOM cells were activated/inhibited by the LED (Supplementary Fig. 6) and we reported that only half of the SOM interneurons display spontaneous activity (Figure 4). This means that only 1 to 2 spontaneously active SOM cells would contact the recorded SPN. In addition, the unitary synaptic weight measured in paired recordings are typically weak (median[interquartile range (IQR)]: 0.4[0.7] mV, Supplementary Fig. 6e) with a strong short-term depression for trains at 10Hz (Figure 5e). The spontaneous activity frequency of SOM cells is in average 6-11Hz and the pulse length used for the LED illumination is 1s done every 10 s. Altogether, we think all these observations are consistent with the absence of effect on the SPN RMP when opto-inhibiting spontaneous SOM interneurons. On the contrary, we observed a significant effect of SOM cells when activated by cortical inputs, both *ex vivo* and *in vivo*.

Figure 2: the spiking depends on various factors, especially in whole cell recordings, the holding potential. At what potential were they held? Why not do these recordings in cell-attached mode and record only spike responses? These results show that SOM and PV are different from each other in both areas, rather than a pronounced topographical difference. The results in 2H and 2K are in sharp contrast to those implied from the *in vivo* recordings, showing higher (than SOM) impact of DMS PV cells on SNR firing. In the slice recordings it seems that PV manipulation does not affect the firing of SPNs.

The holding potential of SPNs was their physiological resting membrane potential and was similar in all the conditions as indicated in the material and methods part "SPNs were held at their physiological membrane potential, in average -77.0 ± 0.3 mV ($n=140$) and there was no statistical difference in the holding membrane potentials between the different experimental conditions". We chose to record SPN activity at their physiological resting membrane potential to evaluate the actual physiological effects; it should be noted that the variability was really low (0.3mV) with no significant difference between conditions. In addition, we chose to do the experiments in whole-cell recordings to measure spiking probability but also the subthreshold I/O curves and EPSP summations in the very same neurons and study the effect of opto-modulation on all these factors.

As explained in the discussion, the stronger effect of PV compared to SOM in *in vivo* spontaneous activity (Figure 1) is likely to come from a stronger spontaneous activity of PV interneurons in urethane anaesthesia. “The stronger effect of silencing PV than SOM interneurons on SNr spontaneous activity in both territories could be due to a stronger spontaneous activity of PV cells, classically described as “fast-spiking” and identified *in vivo* by their higher firing rates^{11,35}.”

This caveat is why experiments using cortical stimulations were suggested by Reviewer 1 and Reviewer 2, to evoke activity in both PV and SOM neurons. We performed these experiments and included them in the revised version. The results obtained with such cortically-evoked activity (Figure 7) are in total agreement with Figure 2H and 2K, since they demonstrate a specificity of GABAergic circuits with stronger weight of SOM cells in DMS and PV cells in DLS.

Figure 3: why was the measure of change in instantaneous frequency used? What period was compared between the frequencies? What was the actual synaptic strength that caused these frequency changes?

We used the measure of instantaneous frequency change to highlight the effect of LED illumination on the spiking activity of SPN in the very same conditions, within less than a second time period. For this purpose, we triggered stable spike trains with 1 sec current injection in SPNs and measured either (1) the instantaneous frequency modulation by a 5 ms light pulse (Fig. 4b) by comparing the frequency between 2 spikes right before the 5ms LED pulse and at the time of the LED pulse or (2) the averaged frequency within 300ms before, 300ms during and 300ms after the 300ms LED pulse (Fig. 4c). We averaged 3 measures for each SPN and each measure was performed every 10 seconds.

We have now included this explanation in the Methods part: “To measure the effect of light on SPN spiking frequency, we triggered stable spike trains with 1sec current injection and measured either (1) the instantaneous frequency (Fig. 4b) by comparing the frequency between 2 spikes right before the 5ms LED pulse and at the time of the LED pulse or (2) the averaged frequency within 300ms before, 300ms during and 300ms after the 300ms LED pulse (Fig. 4c).”

The exact synaptic strength responsible for the effect of interneurons on the instantaneous frequency is hard to measure exactly but from our measurements we know that: (1) the inputs are coming from 3 to 4 PV or SOM interneurons (as we measured the connectivity by comparing the amplitude of IPSPs (with paired recordings) and light-induced IPSPs) with an averaged value of ~1-2mV for one AP (Supplementary Fig. 6) (2) a 5ms LED pulse triggers in average 1 AP and a 300ms optical stimulations is triggering trains of action potentials (average of 13 and 14 evoked APs for SOM and PV cells, Supplementary Fig. 6), so an average of 50Hz firing and, (3) there is a strong short-term depression for PV-SPN and SOM-SPN connections for a 50Hz trains. Altogether, we can evaluate the number of neurons, APs and synaptic weight responsible for the effects on SPN instantaneous frequency.

Figure 5: DMS SOMs provide stronger than PV ipsp onto SPNs. Again, not completely in line with the *in vivo* results. The reversal potential for GABA_A is determined by the pipette solution and it is therefore not surprising that both interneuron types evoke depolarizing responses at a membrane potential of -80 mV. The novelty of this is not clear, as well as the validity of -60 mV as the reversal of chloride under physiological conditions. It would be much more convincing if experiments were done in actual chloride concentration (perforated or cell attached recordings).

The fact that SOM provide stronger IPSPs than PV in DMS is totally in line with our new *in vivo* data presented in Figure 7. Indeed, as explained above, we observed in all our experiments, in brain slices and in our cortical stimulation-evoked *in vivo* experiments that SOM cells have a stronger weight in DMS and PV cells in DLS.

We did not understand exactly the questioning about the reversal of chloride. Indeed, both PV and SOM cells release GABA and act on GABA_A receptors on SPNs so will influence SPNs with a similar reversal of chloride. We, and others, have determined the physiological chloride reversal in SPNs (Dehorter et al., 2009, J Neurosci; Paillé et al., 2013, J Neurosci) at -60mV using methods avoiding any ionic perturbations. It is therefore relevant to measure the functional weight of GABAergic cells on SPNs using this physiological reversal. We explained this point in the material and methods part: “The reversal of chloride was set to be at the physiological value (~ -60 mV), based on previous measures performed in SPNs with methods avoiding any ionic perturbations^{18,25}.” Moreover, as mentioned above, we chose to do the experiments in whole-cell recordings with physiological chloride reversal to measure different parameters and their modulation by opto-activation and opto-inhibition in the very same neurons.

Reviewer #3 (Remarks to the Author):

The authors addressed all my concerns.

We would like to thank Reviewer 3 for his positive feedback about our revised manuscript.

REVIEWERS' COMMENTS:

Reviewer #1 (Remarks to the Author):

The authors have addressed all the comments in the previous review. While the effects on SNr firing in vivo (figure 7) are small and the interpretation of which phase is affected by interneuron inhibition is not always clear, the authors have added appropriate caveats about their interpretations.

With regard to figure 7, the authors should reiterate which statistic was used to compare the firing parameters either in the text or figure caption.

Reviewer #2 (Remarks to the Author):

Comments to authors

The authors have added a significant amount of experiments, supplementary material and text to the manuscript and it had indeed improved.

However, there is still not a full addressing of comments from the two previous reviews:

1) The experiments done in the crossed ChR2/Arch reporter mice are problematic due to the known PV and SOM cells and afferents to the striatum. In particular, optogenetic stimulation of SOM expressing corticostriatal neurons showed large GABAergic responses in SPNs in dorsal striatum. Selected references for SOM/PV afferents were presented in my previous review. Added to them now is the innervation of striatal interneurons by GP (Klug et al, elife 2018). This means that the global pv/som activation inhibition is likely to affect many other than the intended interneurons. The fact that viral injections and global transgenic mice give the same results is indeed surprising, however the experiments described in the supplementary materials are not as extensive and do not replicate the whole dataset currently presented in the main figures. It is the opinion of this reviewer that the viral experiments should be presented in the main text and not in supplementary material since their results are more valid than those obtained from transgenic reporter mice under these experimental conditions.

2) Many of the observations are not novel, including the feedforward inhibitory pathway of the interneurons, their electrophysiological properties, the depolarizing effect of GABA, and the gradient of PV interneuron distribution. The main observation regarding the regional functional differences is not offered an underlying mechanism and moreover, is not very strong in many of the presented data.

3) Another issue is the use of different electrical stimulation protocols for the DLS and DMS, they are deliberately not done in the same place or slice configuration. This should be clarified in the figure describing the slice experiments, explicitly stating that the cortical region stimulated is different for the different conditions. Moreover, a general problem with electrical stimulation is that it does not necessarily include only cortical fibers but can also activate thalamic inputs. The effect on the circuitry of stimulation in S1 compared to cingulate cortex is very different since the cortical projections have different neuronal targets in the striatum, in particular for interneurons (See Klug et al 2018, Monteiro et al 2018).

4) There are still discrepancies between the in vivo and in vitro data, apparent even in the first two figures. While figure 1 shows light modulation of SNR firing under all conditions (DLS,DMS, SOM, PV), the in vitro data of electrical stimulation and optogenetic inhibition shows biased effects, with no modulation under half the conditions. Moreover, in figure 1 e,f the PV show higher modulation in both areas. The experiments in figure 7 suggest that modulation of direct and indirect pathway responses (phase 2 and 3 in SNR multiphasic

responses) was done only in one condition each: SOM/DMS for the direct pathway and PV/DLS for the indirect pathway. Again, what this means is unclear, and if the conclusion should be that PV/DLS is more strongly connected to indirect pathway SPNs, this is not supported by the literature (See Gittis et al, Neuron 2011, J.neuroscience 2010).

5) The core of the manuscript is inhibition from PV/SOM interneurons onto SPNs, however not a single example of such direct connections is presented. The main claim that "SOM cells have a stronger inhibitory weight in DMS" (line 191) is not supported in the data, and at least should be shown by examples of paired recordings between interneurons to SPNs showing that this is the case.

6) The mechanism for any regional differences is not provided or hypothesized. There are several ideas discussed in the discussion section however no clear conclusion. Each condition in the experiment is done in a different mouse line and/or stimulation location, making it very hard to compare the effects. There are no direct comparisons supported by simultaneous recordings or even in the same slice of dorsolateral vs dorsomedial cells. The discussion contains numerous speculations regarding the organization of the striatal circuitry that could explain the regional differences but no clear mechanism is presented as the one underlying the observations.

Minor comments:

- The readout in SNR spiking activity is not a direct readout of SPN activity, since it reflects opposite modulation by direct and indirect pathway SPNs. What does the SNR modulation actually mean in terms of the effect on both SPN types? The authors put forward an idea that SOM and PV separately regulate the direct and indirect pathways but there is no evidence for such a differentiation. It still remains unclear what is the meaning of SNR modulation in terms of the striatal circuitry, given that both interneuron types target both the direct and indirect pathway SPNs.

- A recent paper by Monteiro et al (2018, PMID: 29808928 DOI: 10.1113/JP275936) shows differences between PV cells in DLS and DMS in PV expression, excitability, and input from different cortical areas, both of which have bearing to the current results and should be discussed.

- The difference in targeting of perisomatic regions by PV and dendrites by SOM is an important issue in any analysis of interneurons effect on SPNs. Was there any difference in this pattern between DLS and DMS?

- In figure 4, the baseline potential value should be shown for the traces, especially due to the discrepancy between 4c and 4g in baseline firing of SOM cells. Were the examples in 4c hyperpolarized to prevent spontaneous firing?

- Please read through again for typos and grammar. Typos examples:

Lines 243,492,553: two dots in a row ("..")

Line 481: "Therefore cortical inputs is recruiting recently revealed complex..."

Line 466: interneuron instead of interneurons

REVIEWERS' COMMENTS:

Reviewer #1 (Remarks to the Author):

The authors have addressed all the comments in the previous review. While the effects on SNr firing in vivo (figure 7) are small and the interpretation of which phase is affected by interneuron inhibition is not always clear, the authors have added appropriate caveats about their interpretations. With regard to figure 7, the authors should reiterate which statistic was used to compare the firing parameters either in the text or figure caption.

We would like to thank Reviewer 1 for his positive comments about our study.

The statistical test used in Figure 7 (new Figure 8) was a paired t-test. We have now added this information in the Figure caption.

Reviewer #2 (Remarks to the Author):

Comments to authors

The authors have added a significant amount of experiments, supplementary material and text to the manuscript and it had indeed improved.

However, there is still not a full addressing of comments from the two previous reviews:

1) The experiments done in the crossed Chr2/Arch reporter mice are problematic due to the known PV and SOM cells and afferents to the striatum. In particular, optogenetic stimulation of SOM expressing corticostriatal neurons showed large GABAergic responses in SPNs in dorsal striatum. Selected references for SOM/PV afferents were presented in my previous review. Added to them now is the innervation of striatal interneurons by GP (Klug et al, elife 2018). This means that the global pv/som activation inhibition is likely to affect many other than the intended interneurons. The fact that viral injections and global transgenic mice give the same results is indeed surprising, however the experiments described in the supplementary materials are not as extensive and do not replicate the whole dataset currently presented in the main figures. It is the opinion of this reviewer that the viral experiments should be presented in the main text and not in supplementary material since their results are more valid than those obtained from transgenic reporter mice under these experimental conditions.

To overcome the artefactual activation problem raised by Reviewers 1 and 2 in the first round of revisions, we reproduced the original optogenetics experiments using viral expression of the opsins instead of the transgenic mice. Similar effects were observed with the two approaches, demonstrating that the territory-specificity is indeed due to local GABAergic circuits and not to external sources that may project to the striatum.

As suggested by Reviewer 2, the virus approach and its rationale is now presented explicitly in the Main text (pages 7, 8) and in the new Figure 3.

"The viral approach allowed us to restrict the expression of Arch3 to only striatal interneurons either in DMS and DLS (Supplementary Fig. 4), to exclude any eventual external sources of PV or SOM inputs (from cortex or GP for example) and to study only the effect of local microcircuits."

"Interestingly, we obtained similar results with transgenic mice or virally expressed opsins, which shows that only local PV and SOM striatal interneurons have a role in the effects we observed."

However, in order to avoid overloaded figures with twice the same results, the other sets of experiments that have been replicated with viruses are shown as Supplementary Figures

(Supplementary Figures 7 and 11) but they are explicitly mentioned in the Main text (pages 9 and 15).

“Using the viral infection strategy, we observed similar results, indicating that only local GABAergic circuits are responsible for the inhibitory weight specificity (Supplementary Fig. 7).”

“In addition, similar results were obtained with expression of Arch in SOM and PV cells restricted locally to DMS or DLS using viral strategy (Supplementary Fig. 11).”

In addition, we agree with Reviewer 2 that the study by Klug et al. 2018 is an interesting recent article that is relevant to this study. We therefore included this reference in the main text (page 7).

2) Many of the observations are not novel, including the feedforward inhibitory pathway of the interneurons, their electrophysiological properties, the depolarizing effect of GABA, and the gradient of PV interneuron distribution. The main observation regarding the regional functional differences is not offered an underlying mechanism and moreover, is not very strong in many of the presented data.

This study reports a never before described territoriality of GABAergic circuit functions in the DMS and DLS as well as the underlying functions associated to this dichotomy. To do so, we used well described GABAergic interneuron properties such as feedforward inhibition as functional readouts to provide evidence for the relevance of territory-specificity of SOM and PV cells. These previous studies are acknowledged in the manuscript and are used as a basis for the design of our experiments. Here, using different and complementary approaches, we systematically observed territory specificity which validate the robustness of our conclusions. In addition, we found differences in neuronal distribution, intrinsic properties and the resulting local global connectivity along the striatum which explain the differential effect of PV and SOM cells between the DMS and the DLS. We believe that these specific characteristics constitute the mechanism underlying the territory specificity of local GABAergic circuits in the striatum we report here.

We now emphasize this important aspect in the Main text by discussing the putative mechanism as well as the role of the territory specificity for the global functioning of the basal ganglia (page 19)

“The differences in the characteristics of PV and SOM populations between DMS and DLS would give rise to the territory-specificity. Indeed, even though the unitary connections from PV to SPN or from SOM to SPN are similar in both territories, their global action as populations is higher for PV cells in DMS due to their density and to SOM cells in DMS due to their intrinsic properties. In addition, cross-comparison shows that unitary PV-SPN connections are stronger than SOM-SPN connections in DLS. Altogether, we propose that these properties are underlying the functional dichotomy of PV and SOM cells in DMS and DLS.”

(and pages 22-23):

“Interestingly, we describe here that each interneuron subtype has a similar impact with a territory-specificity, which translates into a differential effect on the downstream SNr. On cortically-evoked activity, PV cells exhibit a strong weight in DLS while SOM cells control more efficiently DMS activity. We propose that potential underlying mechanisms to explain such specificity could come from the differences in the number of cells, their electrophysiological properties and the resulting local global connectivity. Therefore, the specificity stands in the fact that each GABAergic microcircuit, with its own intrinsic characteristics, has a specific role in DLS and DMS. The DLS is responsible for sensorimotor integration leading to habit formation. Sensory information requires to be quickly and reliably integrated and processed to produce a behavior adapted to the environmental stimuli. PV interneurons are reliably activated by the cortical activity, have fast-spiking characteristics and they modulate SPNs for any level of activity, which means that they tightly control sensory inputs of various amplitudes. Therefore the intrinsic properties of PV cells, the fact that they are much more numerous in DLS^{33,34}, with denser arborizations³⁷ and their resulting action on SPNs are particularly

adapted to the temporal precision needed to control sensorimotor information transmission. The DMS is involved in associative functions, receiving mainly inputs from the frontal parts of the cortex⁵⁶. Frontal cortex displays a lot of recurrent activity in the networks, particularly during working memory tasks^{57,58}. Fronto-corticostriatal inputs lead to recurrent activity in striatum that could be modulated by a more global inhibition, less precise in time but efficient to modulate network activity level. SOM cells are less dependent on cortical inputs to discharge and not time-locked. In addition, they control more efficiently the first steps of build-up activity (for a small cortical activity) but then tend to lose their efficiency with increasing SPN activity. SOM cells would thus have the ability to drive the GABAergic modulation of associative integration. Therefore, the specificity of GABAergic microcircuits might be an active part of the different integration processes involved in the territory-specific striatal functions. We would like to point out that we focused here on feed-forward inhibition mediated by cortical inputs but future studies should extend this work to another major striatal input coming from the thalamus.”

3) Another issue is the use of different electrical stimulation protocols for the DLS and DMS, they are deliberately not done in the same place or slice configuration. This should be clarified in the figure describing the slice experiments, explicitly stating that the cortical region stimulated is different for the different conditions. Moreover, a general problem with electrical stimulation is that it does not necessarily include only cortical fibers but can also activate thalamic inputs. The effect on the circuitry of stimulation in S1 compared to cingulate cortex is very different since the cortical projections have different neuronal targets in the striatum, in particular for interneurons (See Klug et al 2018, Monteiro et al 2018).

This study compares GABAergic local microcircuits in two distinct and distant striatal regions that receive inputs from different cortical areas. In this matter, we used two distinct slice models that preserve the respective corticostriatal connections of DMS (CG2 inputs) and DLS (S2 inputs). Indeed, as shown in many studies (McGeorge et al., 1989, Alexander et al., 1990, Hintiryan et al., 2016, Hunnicutt et al., 2016, and also the recent report by Klug et al., 2018), these inputs are specific to each territory but with the same density of pyramidal cells in CG2 projecting to DMS and S2 projecting to DLS (Klug et al., 2018). The difference in brain slices, which is intentional and necessary to preserve corticostriatal connections and stimulated cortical areas, was indicated in the Methods section. In addition, this strategy allows us to avoid stimulating unspecific afferents such as thalamic afferents since (i) they are not preserved in our corticostriatal slices (unpublished observations) and (ii) we stimulate specifically cortical inputs from layer 5 pyramidal cells directly in the cortex, in both DMS and DLS slices, with the same stimulation protocols.

To better clarify this strategy, we have added a scheme explicitly indicating the two different sites of cortical stimulation for DMS and DLS in new Figure 2d. To avoid overloading Figures 3 and 7, we have represented only one cortical afferent. In Figure 8, a scheme was already presented to clearly show on the scheme that the stimulations were done in two different cortical areas.

In addition, we have now included this precision in the Methods part (pages 29-30):

“Connections between the somatosensory cortex (layer 5) and the striatum are preserved in a horizontal plane. For somatosensory territory study, horizontal brain slices containing the somatosensory cortex and the corresponding corticostriatal projection field were prepared according to the methods previously described⁷⁰. Concerning the associative territory study, we set up a new corticostriatal slice (para-sagittal with a 30° angle) preserving the connections between the cingulate cortex and the striatum, based on 3D anatomical reconstruction of such connections⁵⁶. We chose to use these two different slices to be able to preserve and stimulate specific cortical inputs to either DMS or DLS. In addition both slices orientation do not preserve thalamic inputs, allowing a specific activation of only corticostriatal inputs from layer 5 pyramidal cells.”

We also provide details about this point in the Results section (page 6):

“To this aim, we characterized the effect of striatal GABAergic circuits locally in the striatum, by performing ex vivo experiments using brain slices preserving layer 5 cortical connections from cingulate cortex to DMS or from somatosensory cortex to DLS (to specifically stimulate cortical inputs of each territory we used slices preserving corticostriatal inputs to either DMS or DLS, see Methods).”

4) There are still discrepancies between the in vivo and in vitro data, apparent even in the first two figures. While figure 1 shows light modulation of SNr firing under all conditions (DLS, DMS, SOM, PV), the in vitro data of electrical stimulation and optogenetic inhibition shows biased effects, with no modulation under half the conditions. Moreover, in figure 1 e,f the PV show higher modulation in both areas. The experiments in figure 7 suggest that modulation of direct and indirect pathway responses (phase 2 and 3 in SNr multiphasic responses) was done only in one condition each: SOM/DMS for the direct pathway and PV/DLS for the indirect pathway. Again, what this means is unclear, and if the conclusion should be that PV/DLS is more strongly connected to indirect pathway SPNs, this is not supported by the literature (See Gittis et al, Neuron 2011, J.neuroscience 2010).

The results presented in Figure 1 and the ones presented in Figures 2 and 7 are not contradictory but they represent different kind of activities: effect of the interneurons on spontaneous activity for Figure 1 and on cortically-evoked activity in Figures 2 and 7. We do agree that the cautious interpretation of the differences between these two types of experiments have to be explicitly mentioned in the Discussion section, which is now improved in the current version.

The *in vivo* results presented in Figure 1 show the effect of modulation of striatal interneurons of spontaneous SNr frequency discharge. We observed that nearly all individually significantly modulated neurons were activated, suggesting that in all conditions, the net effect of opto-inhibition is a shift of the balance between the direct and the indirect pathways towards a stronger relative activity of the indirect pathway. This would suggest that opto-inhibition of striatal interneurons disinhibits more strongly indirect pathway neurons (iSPNs) than direct pathway neurons (dSPNs). PV and SOM interneurons connect dSPNs and iSPNs rather symmetrically (Planert et al., 2010, Straub et al., 2016). The marginal asymmetry described in Gittis et al., 2010 and 2011, with a preferential connection to dSPNs, would no more explain our results, since it would lead to a stronger disinhibition of the inhibitory direct pathway. Therefore, we do agree with Reviewer 2 that it is unlikely that our result on SNr activity is due to an asymmetrical connection of striatal interneurons on SPNs. However, iSPNs are known to be more excitable than dSPNs (Kreitzer and Malenka, 2007, Kravitz et al., 2010, Planert et al., 2013). Therefore, while the effect of opto-inhibition of striatal interneurons might be similar on dSPNs and iSPNs, it may translate into iSPNs reaching spiking threshold faster than dSPNs, under spontaneous anesthetized network state. This should result in a stronger indirect pathway activity relative to the direct pathway, and thus a net activation of SNr units.

In addition, opto-inhibiting of SOM interneurons leads to a smaller effect on SNr activity than opto-inhibiting PV interneurons in both DMS and DLS. This could be explained by the fact that in our anesthetized *in vivo* experiments on spontaneous SNr activity (Figure 1), it is likely that PV interneurons have a stronger spontaneous activity than SOM interneurons. Indeed, PV interneurons are classically described as “fast-spiking” *in vivo* and typically identified by a short waveform duration and a higher activity compared to MSNs (e.g. in Berke et al., 2004; Mallet et al., 2005); while little data is available on identified SOM interneurons, it is likely that their spontaneous activity would be weaker than “fast-spiking” interneurons. Therefore, it is consistent that the opto-inhibition of more active population (PV) has a stronger effect than the opto-inhibition of a less active population (SOM).

To investigate the role of striatal interneurons on SNr activity without the bias induced by their likely different spontaneous activity, we compared the response of SNr units to cortical stimulations with and without opto-inhibition of PV or SOM interneurons in DMS or DLS (Figure 7). We observed

that striatal interneurons had specific effects on the “triphasic” response, depending on the territory: opto-inhibition of SOM-DMS led to a decrease of the inhibition phase, and opto-inhibition of PV-DLS led to a decrease of the late excitation phase. Though it would be tempting to conclude that SOM/PV interneurons only affect each direct/indirect pathway, respectively, we do agree that the interpretation should be more careful. Indeed the activation of the direct and indirect pathways are overlapping, and the phases that can be measured, while mainly corresponding to each pathway, are still the sum of this overlap. Additional explanations are related to specific properties of iSPNs and dSPNs and specificities in the cortical inputs innervating DMS and DLS. As mentioned above, iSPNs are more excitable and are recruited faster with shorter responses than dSPNs, which maintain a longer activation in response to cortical stimulation (Flores-Barrera et al., 2010). The latency of interneurons activation is shorter for PV cells (new Figure 2), which would be likely to modulate more efficiently the first recruited iSPNs while SOM cells would control more efficiently dSPNs, activated after and longer. In addition, our observation is also coherent with different organization of cortical inputs described with rabies viruses. Inputs from cingulate cortex contact more dSPNs in DMS while iSPNs are more connected by somatosensory inputs in DLS (Wall et al., 2013).

These explanations are included in the Discussion part to acknowledge the limitations of our experiments and our interpretations of the differences between the effects on spontaneous or evoked activity (pages 20-21):

“Our recordings of SNr spontaneous activity show an activation of SNr upon opto-inhibition of both interneurons in both territories, meaning that the net effect of SOM and PV cells on SNr spontaneous activity is inhibitory. This suggests that interneurons modulate the balance between the direct (inhibitory on SNr) and indirect (disinhibitory) pathways, towards a relative activation of the direct pathway, and/or inhibition of the indirect pathway, yet the striatal mechanism leading to this result could be multiple. Recent studies show that GABAergic interneurons contact both direct (dSPNs) and indirect pathway SPNs (iSPNs)^{39,40,42}, meaning that the local interneurons might control both SPN subtypes in a similar way. Since iSPNs are more excitable than dSPNs⁴⁹, their disinhibition could lead to a stronger increase in firing rate in iSPNs than dSPNs, thus explaining the observed overall increase of firing rate in SNr after silencing striatal interneurons. However, considering the dual depolarizing/hyperpolarizing effect of PV and SOM described here, the opposite hypothesis cannot be excluded: silencing striatal interneurons could lead to a decreased summation of spontaneous cortical inputs, which, if stronger in dSPNs, would also shift the net balance towards an activation of SNr. Other members of striatal microcircuits are also likely involved in this effect, and could contribute to either a decreased activity of the direct pathway, or an increased activity of the indirect pathway. The stronger effect of silencing PV than SOM interneurons on SNr spontaneous activity in both territories could be due to a stronger spontaneous activity of PV cells, classically described as “fast-spiking” and identified in vivo by their higher firing rates^{11,36}. When stimulating cortical afferents, it is possible to visualize within the SNr different phases of response corresponding to hyper-direct, direct and indirect pathways. Similarly to local striatal control, we observed a specific modulation of cortically-evoked SNr activity from SOM-DMS and PV-DLS. Interestingly, SOM-DMS seem to modulate the inhibition resulting from the direct pathway and PV-DLS the late excitation related to indirect pathway recruitment. Though it would be tempting to jump to the conclusion that SOM/PV interneurons only affect each direct/indirect pathway, respectively, we think the interpretation should be more careful since the activation of the direct and indirect pathways are overlapping, and the phases that can be measured, while mainly corresponding to each pathway, are still the sum of this overlap. Nevertheless, iSPNs are more excitable and are recruited faster with shorter responses than dSPNs, which maintain a longer activation in response to cortical stimulation⁵⁰. The latency of interneuron activation is shorter for PV cells (Fig. 2 and coherent with kinetics described in⁴²), which would be likely to modulate more efficiently the first recruited iSPNs while SOM cells would control more efficiently dSPNs, activated later. In addition, our observation is also coherent with different organization of cortical inputs from cingulate cortex, which contact more dSPNs in DMS, and in DLS where iSPNs are more connected by somatosensory inputs⁵¹. In vivo

recordings also show that the interplay of inhibitory and excitatory effects of GABA is complex on the resulting effect on output structures such as SNr. Both SOM-DMS and PV-DLS opto-inhibition lead to a decrease of a striato-SNr pathway, meaning that the net effect of striatal interneurons would be excitatory locally on SPNs. A recent study also describes an overall decrease of SPN activity after silencing PV cells in vivo due to a disinhibitory loop from PV cells to NPY (overlapping with SOM cells) interneurons⁵². Another disinhibitory mechanism has been described between TH-positive interneurons and SOM cells⁵³. A silencing of SOM might also lead to a decreased activity of SPNs, since SOM interneurons (but not PV) inhibit cholinergic interneurons⁴², that in turn inhibit SPNs via their activation of NPY-NGF interneurons⁵⁴. Therefore cortical inputs is recruiting complex and various local striatal micro-circuits whom interplay will lead to complex effect combining inhibition and disinhibition. Local disinhibitory role of interneurons (such as VIP or SOM interneurons in cortex or amygdala) has been recently highlighted in several brain structures as an important mechanism to regulate the input/output flow of information⁵⁵.”

5) The core of the manuscript is inhibition from PV/SOM interneurons onto SPNs, however not a single example of such direct connections is presented. The main claim that “SOM cells have a stronger inhibitory weight in DMS” (line 191) is not supported in the data, and at least should be shown by examples of paired recordings between interneurons to SPNs showing that this is the case.

This important point was raised by Reviewers 1 and 2 in the first round of revisions. We therefore responded to this critical point by providing a quantification of unitary connections between pairs of PV-SPN and SOM-SPN in the answers in our point by point letter to the Reviewers.

We reported that, although no significant difference in the weight of unitary connections in DMS and DLS could be observed for each interneuron populations, there are differences in the amplitude of light-evoked IPSPs. These data indicate that the differential weight on SPNs is rather due to differences in the number of recruited interneurons. This effect can be explained by the higher number of PV neurons in the DLS and by the increased excitability of SOM interneurons in the DMS (therefore easier to recruit). In addition, crossed comparisons show that PV-SPN connections are stronger than SOM-SPN connections in DLS.

In consequence, these differences in PV distribution and in SOM intrinsic properties lead to selective connectivity of both populations to SPNs in DMS and DLS, which support the territory specificity of GABAergic microcircuits within the striatum.

These data were not originally presented in the manuscript but we do agree with Reviewer 2 that this information was missing. Therefore, we now added the quantification of unitary connections from PV and SOM cells to SPNs, both in DMS and DLS, and also compared it with light-induced IPSPs in the new Figure 5.

These experiments and their interpretation are presented in the Results section (page 11):

“Another explanation could be that the unitary synaptic weight of PV-SPN and SOM-SPN connections are not the same in DMS and DLS. To go deeper in the origin of the differential effect PV and SOM cells, we thus performed paired patch-clamp recordings to measure the unitary synaptic weight of PV-SPN and SOM-SPN connections. We observed that unitary IPSP amplitudes induced with a single presynaptic AP were not significantly different between PV and SOM in DMS but significantly higher for PV-SPN connections in DLS ($p=0.0094$) (Fig. 5i and 5j). The comparison with light-induced IPSPs indicate that about 3 to 4 interneurons are recruited by the opto-activation in our conditions for SOM-DLS and PV-DLS, 1 to 2 interneurons for PV-DMS and 8 to 10 interneurons for SOM-DMS. In addition, when comparing the different populations, there is a significantly higher amplitude for SOM cell opto-activation in DMS and PV cell opto-activation in DLS ($p=0.0158$ for DMS and $p=0.0019$ for DLS). This data show that, for each GABAergic population, there is no significant difference in the weight of unitary connections in DMS and DLS. The differences of global inhibitory weight would thus

result from the recruitment of more PV cells in DLS (due to their higher density in DLS) and more SOM cells in DMS (due to their higher excitability in DMS)."

6) The mechanism for any regional differences is not provided or hypothesized. There are several ideas discussed in the discussion section however no clear conclusion. Each condition in the experiment is done in a different mouse line and/or stimulation location, making it very hard to compare the effects. There are no direct comparisons supported by simultaneous recordings or even in the same slice of dorsolateral vs dorsomedial cells. The discussion contains numerous speculations regarding the organization of the striatal circuitry that could explain the regional differences but no clear mechanism is presented as the one underlying the observations.

We understand Reviewer 2's concern about the use of different slice models. However we would like to stress out here that the suggested experiments are technically impossible since DMS and DLS are two distinct regions of the striatum that are distant and for which corticostriatal connections are organized in different planes. Rather than testing the two regions on a unique slice without preserving proper inputs, we believe that keeping the specific cortical inputs orientation to each territory is critical to avoid aspecific activations (for example from thalamus or recruitment of local striatal microcircuits). This is the reason why we put efforts on the orientation of brain slices that preserve physiologically relevant corticostriatal inputs to each of the territory.

As indicated in point 3, we now better describe this approach in the Main text (page 6):

"To this aim, we characterized the effect of striatal GABAergic circuits locally in the striatum, by performing ex vivo experiments using brain slices preserving layer 5 cortical connections from cingulate cortex to DMS or from somatosensory cortex to DLS (to specifically stimulate cortical inputs of each territory we used slices preserving inputs to either DMS or DLS, see Methods)."

It should be noted that a set of *in vivo* experiments presented in Figure 1 were performed within the same mouse in which interneurons activity was modulated both in DMS and in DLS while recording SNr spontaneous activity (Figure 1). The fact that the PV modulation in DMS and DLS systematically led to different effect in SNr within the same mice is a direct evidence for the territory-specificity.

The core of our manuscript is then to find explanations for such specificity. Using different and complementary approaches, we systematically observed territory specificity which validate the robustness of our conclusions. In addition, we found differences in neuronal distribution, intrinsic properties and the resulting local global connectivity along the striatum which explain the differential effect of PV and SOM cells between the DMS and the DLS. We believe that these specific characteristics constitute the mechanism underlying the territory specificity of local GABAergic circuits in the striatum we report for the first time in this study. As detailed in point 2, we have now developed the part about the eventual mechanisms and role of such specificity in the Discussion section (pages 19 and 22-23).

Minor comments:

- The readout in SNR spiking activity is not a direct readout of SPN activity, since it reflects opposite modulation by direct and indirect pathway SPNs. What does the SNR modulation actually mean in terms of the effect on both SPN types? The authors put forward an idea that SOM and PV separately regulate the direct and indirect pathways but there is no evidence for such a differentiation. It still remains unclear what is the meaning of SNR modulation in terms of the striatal circuitry, given that both interneuron types target both the direct and indirect pathway SPNs.

As mentioned in the point 4, we do not claim that SOM and PV separately regulate the direct or indirect pathway. Indeed, even though we observed a major effect of SOM on direct pathway and PV

cells on indirect pathway, we remain cautious about the interpretation. These precautions are detailed in the Discussion section (please see details in the point 4).

- A recent paper by Monteiro et al (2018, PMID: 29808928 DOI: 10.1113/JP275936) shows differences between PV cells in DLS and DMS in PV expression, excitability, and input from different cortical areas, both of which have bearing to the current results and should be discussed.

We agree with Reviewer 2 that this recent paper is important in our context since they describe a slightly different picture concerning the intrinsic properties of PV cells in DMS vs DLS. This difference could be explained by the experimental conditions such as composition of aCSF or intracellular solution (leading to a significantly different reversal of chloride for example) and also the slice orientation (they use coronal slices for both territories).

We now discuss this study in the Discussion section (page 18):

"The observed distribution in mice is consistent with previous studies showing a rostral-caudal gradient of PV cells in rats, monkeys and humans³²⁻³⁵. In addition, we observed here that PV cells have similar membrane and spiking properties in DMS and DLS; this observation being different from a recent study showing higher excitability of PV cells in DMS³⁶ (probably due to different experimental conditions such as composition of extra- and intracellular solutions and slice orientation)."

- The difference in targeting of perisomatic regions by PV and dendrites by SOM is an important issue in any analysis of interneurons effect on SPNs. Was there any difference in this pattern between DLS and DMS?

This is an interesting point that requires the use of specific techniques such as two-photon glutamate uncaging as used in Straub et al. (Straub et al., 2016). Indeed, in their study, the authors report preferential targeting of perisomatic region by PV cells and dendritic region by SOM cells. Nevertheless, even though PV cells contact SPN closer to the soma than SOM cells, their contacts are both located within the first 250µm of the SPN dendritic arborization. Therefore, striatal PV and SOM cells are both placed in a strategic position to modulate inputs to SPNs.

This interesting point is discussed in the Discussion section (page 19):

"The differential localization of the synapses from SOM and PV cells on the SPN dendrites could contribute to the specific modulation of cortical inputs. Nevertheless, even though PV cells contact SPN closer to the soma than SOM cells, their contacts are both located within the first 250µm of the SPN dendritic arborization⁴¹, placing both of them in a strategic position to modulate the integration of glutamatergic inputs."

- In figure 4, the baseline potential value should be shown for the traces, especially due to the discrepancy between 4c and 4g in baseline firing of SOM cells. Were the examples in 4c hyperpolarized to prevent spontaneous firing?

According to Reviewer 2 comment, the holding potential of the cells in the different experimental configurations is now shown in new Figure 5.

It should be noted that examples in Figure 4c (now Figure 5c) show SOM cells with no spontaneous activity as only half of SOM population display spontaneous activity. Therefore these examples correspond to non-spontaneously active SOM cells and were therefore not hyperpolarized.

- Please read through again for typos and grammar. Typos examples:
Lines 243,492,553: two dots in a row ("..")

Line 481: "Therefore cortical inputs is recruiting recently revealed complex..."

Line 466: interneuron instead of interneurons

These errors have now been corrected.